

# Influx of African biomass burning aerosol during the Amazonian dry season through layered transatlantic transport of black carbon-rich smoke

Bruna A. Holanda[1,2], Mira L. Pöhlker[1,*], Jorge Saturno[2,a], Matthias Sörgel[2,3], Jeannine Ditas[4,1],

Florian Ditas[1,2,b], Qiaoqiao Wang[2,c], Tobias Donth[5], Paulo Artaxo[6], Henrique M. J. Barbosa[6], Ramon Braga[1], Joel Brito[6,d], Yafang Cheng[1], Maximilian Dollner[5,6], Marco Aurélio Franco[6], Johannes Kaiser[3,8], Thomas Klimach[1], Christoph Knote[7], Ovid O. Krüger[1], Daniel Fütterer[8], Jošt V. Lavrič[9], Nan Ma[4,1], Luiz A. T. Machado[10], Jing Ming[1,2], Fernando Morais[5], Hauke Paulsen[1], Daniel Sauer[8], Hans Schlager[8], Hang Su[1], Bernadett Weinzierl[8,11], Adrian Walser[5,8], David Walter[1,9],

Manfred Wendisch[5], Helmut Ziereis[8], Martin Zöger[8], Ulrich Pöschl[1], Meinrat O. Andreae[2,12], and Christopher Pöhlker[1,2,*]

[1] *Multiphase Chemistry Department, Max Planck Institute for Chemistry, 55128 Mainz, Germany*

[2] *Biogeochemistry Department, Max Planck Institute for Chemistry, 55128 Mainz, Germany*

[3] *Atmospheric Chemistry Department, Max Planck Institute for Chemistry, 55128 Mainz, Germany*

[4] *Center for Air Pollution and Climate Change Research (APCC), Institute for Environmental and Climate Research (ECI), Jinan University, Guangzhou, 511443, China*

[5] *Leipzig Institute for Meteorology, Leipzig University, Leipzig, Germany*

[6] *Institute of Physics, University of São Paulo, São Paulo 05508-900, Brazil*

[7] *Meteorological Institute, Ludwig Maximilians University, Munich, Germany*

[8] *German Aerospace Center (DLR), Institute for Atmospheric Physics & Flight experiments, 82234 Oberpfaffenhofen, Germany*

[9] *Max Planck Institute for Biogeochemistry, 07701 Jena, Germany*

[10] *National Institute for Space Research (INPE), São José Dos Campos, Brazil*

[11] *University of Vienna, Aerosol Physics and Environmental Physics, 1090 Wien, Austria*

[12] *Scripps Institution of Oceanography, University of California San Diego, La Jolla, California 92093, USA*

[a] *now at: Physikalisch-Technische Bundesanstalt, 38116 Braunschweig, Germany*

[b] *now at: Hessisches Landesamt für Naturschutz, Umwelt und Geologie, 65203 Wiesbaden, Germany*

[c] *now at: Institute for Environmental and Climate Research, Jinan University, China.*

[d] *now at: IMT Lille Douai, Univ. Lille, SAGE, 59000 Lille, France*

*Correspondence to:* M. Pöhlker (m.pohlker@mpic.de) and C. Pöhlker (c.pohlker@mpic.de)



**Abstract**

Black carbon (BC) aerosols are influencing the Earth's atmosphere and climate, but their microphysical properties, spatiotemporal distribution and long-range transport are not well constrained. This study analyzes the transatlantic transport of BC-rich African biomass burning (BB) pollution into the Amazon Basin, based on airborne observations of aerosol particles and trace gases in and off the Brazilian coast during the ACRIDICON-CHUVA campaign in September 2014, combining in-situ measurements on the re-

search aircraft HALO with satellite remote-sensing and numerical model results.

During flight AC19 over land and ocean at the Brazilian coastline in the northeast of the Amazon Basin, we observed a BC-rich atmospheric layer at ~3.5 km altitude with a vertical extension of ~0.3 km. Backward trajectory analyses suggest that fires in African grasslands, savannas, and shrublands were the main source of this pollution layer, and that the observed BB smoke had undergone more than 10 days of

atmospheric transport and aging. The BC mass concentrations in the layer ranged from 0.5 to 2 µg m$^{-3}$, and the BC particle number fraction of ~40 % was about 8 times higher than observed in a fresh Amazonian BB plume, representing the highest value ever observed in the region. Upon entering the Amazon Basin, the layer started to broaden and to subside, due to convective mixing and entrainment of the BB aerosol into the boundary layer. Satellite observations show that the transatlantic transport of pollution layers is a

frequently occurring process, seasonally peaking in August/September.

By analyzing the aircraft observations within the broader context of the long-term data from the Amazon Tall Tower Observatory (ATTO), we found that the transatlantic transport of African BB smoke layers has a strong impact on the north-central Amazonian aerosol population during the BB-influenced season (July to November). Specifically, the early BB season in this part of the Amazon appears to be domi-

nated by African smoke, whereas the later BB season appears to be dominated by South American fires. This dichotomy is reflected in pronounced changes of aerosol optical properties such as the single scattering albedo (increasing from 0.85 in August to 0.90 in November) and the BC-to-CO enhancement ratio (decreasing from 7.4 to 4.4 ng m$^{-3}$ ppb$^{-1}$). Our results suggest that, despite the high amount of BC particles, the African BB aerosol act as efficient cloud condensation nuclei (CCN) with potentially important impli-

cations for aerosol-cloud interactions and the hydrological cycle in the Amazon Basin.



## 1. Introduction

Biomass burning (BB) in the African and South American tropics and subtropics represents a globally significant source of atmospheric aerosol particles and trace gases (Andreae, 1991; Andreae et al., 1988; Barbosa et al., 1999; Ichoku and Ellison, 2014; Kaiser et al., 2012; Reddington et al., 2016; van der Werf et al., 2017). A major constituent of BB smoke is black carbon (BC), which is of major climate relevance through its roles in the so-called direct, semi-direct, and indirect aerosol effects (Bond et al., 2013; Stocker et al., 2013). In BB plumes, BC particles are co-emitted along with organic aerosol (OA) and inorganic salts in proportions that depend on the fuel type and fire phase (Allen and Miguel, 1995; Andreae, 2019; Andreae and Merlet, 2001; Jen et al., 2019; Levin et al., 2010; Reid et al., 2005). During their typical atmospheric lifetime of several days, BC particles undergo atmospheric aging, creating internally mixed BC aerosols via the condensation of low and semi-volatile compounds, coagulation, and cloud processing (Bond et al., 2013; Cubison et al., 2011; Konovalov et al., 2017, 2019; Schwarz et al., 2008; Willis et al., 2016). The formation of non-absorbing coatings on the BC cores changes the particle optical, chemical, and physical properties, with implications for their atmospheric cycling and lifetime (e.g., Zhang et al., 2015; Pokhrel et al., 2017; Ditas et al., 2018).

In terms of the *direct* aerosol effect, BC particles efficiently absorb shortwave (solar) radiation and, thus, affect the Earth's radiative energy budget (Boucher et al., 2016). Recent studies have classified BC as the second largest contributor to global warming and estimated its direct radiative forcing as high as +1.1 W m$^{-2}$, with 90 % uncertainty bounds spanning from +0.17 to +2.1 W m$^{-2}$ (Bond et al., 2013 and references therein). These large uncertainties arise from an insufficient understanding of the BC microphysical properties and its spatiotemporal distribution in the atmosphere (Boucher et al, 2013, Andreae and Ramanathan, 2013). With respect to the *indirect* aerosol effect, the numerous BB-emitted BC particles potentially act as cloud condensation nuclei (CCN) and, thus, influence cloud microphysical (e.g., cloud droplet sizes and number concentrations) and optical (e.g., cloud reflectivity) properties (Andreae, 2004; Andreae and Rosenfeld, 2008; Jiang et al., 2016; Kuhn et al., 2010; Pöhlker et al., 2016). While freshly emitted BC particles tend to be hydrophobic, internal mixing with soluble and/or wettable compounds increases their hygroscopicity and, thus, enhances their ability to act as CCN (Laborde et al., 2013; Liu et al., 2017; Tritscher et al., 2011). The semi-direct effects of BC particles refer to absorption of solar radiation and a related local heating in the atmosphere, perturbing the atmospheric temperature structure (e.g., by strengthening temperature inversions) as well as altering cloud cover and albedo (Brioude et al., 2009; Koch and Del Genio, 2010; Zuidema et al., 2016). The sign of the radiative forcing from *semi-direct* effects depends on the aerosol optical and cloud microphysical properties, as well as on the relative altitudes of the clouds versus BC layers (Feingold, 2005; Johnson et al., 2004; Keil and Haywood, 2003; Wang et al., 2019).



The Amazonian atmosphere is characterized by a pronounced seasonality in aerosol concentrations (e.g., BC and CCN) and in other aerosol properties (e.g., single scattering albedo) (Roberts et al., 2001; Roberts, 2003; Martin et al., 2010; Artaxo et al., 2013; Rizzo et al., 2013; Andreae et al., 2015; Pöhlker et al., 2016; Saturno et al., 2018b). This makes the Amazon Basin an ideal environment to study atmospheric and biogeochemical processes as a function of the highly variable aerosol population. During the wet sea-

son (February to May), trace gas and aerosol emissions from the regional biosphere predominantly regulate atmospheric cycling, precipitation patterns, and regional climate (Pöhlker et al., 2012; Pöschl et al., 2010). Average wet season black carbon (BC) mass concentrations, $M_{BC}$, are ~0.07 µg m$^{-3}$ and $M_{BC}$ approaches zero during pristine episodes (Andreae and Gelencsér, 2006; Pöhlker et al., 2018). In contrast, the dry season (August to November) is characterized by intense and persistent BB emissions, changing

substantially the atmospheric composition and cycling (Artaxo et al., 2013; Rizzo et al., 2013). In central Amazonia, average dry season BC mass concentrations are ~0.4 µg m$^{-3}$ with peaks reaching ~0.9 µg m$^{-3}$ (Pöhlker et al., 2018; Saturno et al., 2018b). In the southern hot-spot regions of agriculture-related burning, Artaxo et al. (2013) reported a dry season mean of $M_{BC} \approx 2.8$ µg m$^{-3}$.

The Amazonian trace gases and aerosol particles originate from regional, continental, and trans-conti-

nental sources. Various observational results underline that the long-range transport (LRT) of (long-lived) species from Africa plays a major role for the Amazonian atmospheric composition. For instance, the plume-wise LRT of African dust and smoke during the Amazonian wet season has been well documented (Talbot et al., 1990; Swap et al., 1992; Ansmann et al., 2009; Baars et al., 2011; Wang et al., 2016; Moran-Zuloaga et al., 2018). Moreover, the arrival of a plume of remarkably sulfur-rich aerosol particles in the

central Amazon in September 2014 has been traced back to the LRT of volcanogenic emissions from eastern Congo (Saturno et al., 2018a). For general illustration, animations[1] of the Goddard Earth Observing Model (Version 5, GEOS-5) show that aerosol particles are transported efficiently from Africa to South America and to a lesser extent from South America to Africa (Colarco et al., 2010; Yasunari et al., 2011).

In the light of these examples of effective transatlantic transport of African aerosol particles, it is plau-

sible to assume that the LRT of smoke from the intense African BB events also plays a substantial role during the Amazonian dry season. In fact, the LRT in defined layers with enriched pollution has been debated as possible transport mechanism. The earliest observations of such pollution layers in the free troposphere over the Brazilian coast can be found in ozone ($O_3$) soundings made from Natal, on the east coast of Brazil (5.8º S, 35.2º W), where $O_3$ mixing ratios of ~70 ppb were measured with a maximum in Sep-

tember (Kirchhoff et al., 1983; Logan and Kirchhoff, 1986). These measurements were continued over a ten-year period (1978-1988), confirming the climatological presence of a tropospheric $O_3$ maximum over

---

[1] https://climate.nasa.gov/climate_resources/146/video-simulated-clouds-and-aerosols/ (last access 04 Jul 2019)
https://gmao.gsfc.nasa.gov/research/aerosol/modeling/nr1_movie/ (last access 04 Jul 2019)



the Brazilian coast, centered at the 500 hPa pressure level and peaking in the September-October period (Kirchhoff et al., 1991). The first comprehensive airborne measurements of these pollution layers off the South American coast (near Natal, Brazil), made in 1989, were attributed to LRT of African BB emissions

(Andreae et al., 1994). Additional aircraft campaigns in the regions of southern Africa, the tropical South Atlantic, and the Amazon Basin have found pollution layers in the free troposphere with similar characteristics (e.g., Andreae et al., 1988; Diab et al., 1996; Thompson et al., 1996; Bozem et al., 2014; Marenco et al., 2016). Recent studies in the central basin, at the Amazon Tall Tower Observatory (ATTO), and in northeastern edge of the Amazon discovered indications for a significant abundance of African smoke dur-

ing the Amazonian dry season (Barkley et al., 2019; Pöhlker et al., 2019, 2018; Saturno et al., 2018b; Wang et al., 2016). However, robust quantitative data from observations and/or models (e.g., African BC and CCN fractions in the basin) have remained sparse.

This study focusses on the transatlantic transport of African pollution towards and into the Amazon Basin by combining *in situ* aircraft observations, modeling results, and remote sensing data. The core of

this work are aircraft observations during the ACRIDICON-CHUVA campaign over Amazonia in September 2014 (Wendisch et al., 2016) within a defined African pollution layer upon its arrival at the South American coast. Here, we characterize the layered transatlantic transport of African pollution by means of (i) characteristic aerosol and trace gas profiles, (ii) backward trajectories and African BB source regions, (iii) the seasonality of the pollution transport, (iv) the horizontal and vertical extent of the transported lay-

ers, and (v) the convective mixing and smoke entrainment from the layers into the planetary boundary layer as they are transported from the ocean over the South American continent. Finally, we integrate the key results of this study into the broader picture of the long-term aerosol observations at the central Amazonian ATTO site to estimate the relevance of African pollution for the aerosol life cycle in the dry season.


## 2. Materials and methods

### 2.1. The ACRIDICON-CHUVA campaign

The data presented here were obtained during the ACRIDICON-CHUVA aircraft campaign, which took place over the Amazon Basin during the 2014 dry season, from 6 September to 1 October (Machado et al.,

2017; Wendisch et al., 2016). ACRIDICON stands for "Aerosol, Cloud, Precipitation, and Radiation Interactions and Dynamics of Convective Cloud Systems" and CHUVA is the acronym for "Cloud Processes of the Main Precipitation Systems in Brazil: A Contribution to Cloud Resolving Modeling and to the Global Precipitation Measurement". The main objective of this campaign was to better understand the interactions between aerosol particles, deep convective clouds, and atmospheric radiation. For this purpose,

a comprehensive set of instruments for airborne observations of aerosol physical and chemical properties, trace gases, radiation, and cloud properties was utilized. The measurements were conducted onboard the





German HALO (High Altitude and LOng range Research Aircraft), operated by the German Aerospace Center (DLR), which can reach altitudes up to 15 km with flight durations up to ten hours. The research flights covered a wide geographic area of the Amazon Basin, probing different pollution states by means

of highly resolved atmospheric profiles. Detailed information on the ACRIDICON-CHUVA campaign and instrumentation installed on HALO can be found in Wendisch et al. (2016).

## 2.2.    Airborne measurements of aerosol, trace gas and meteorology during ACRIDICON-CHUVA

Navigation and basic meteorological data (e.g., air pressure, temperature, humidity, and water vapor mix-

ing ratio) were obtained from the BAsic HALO Measurement And Sensor System (BAHAMAS) at 1 s time resolution. BAHAMAS acquires data from air flow and thermodynamic sensors as well as from the aircraft avionics and a high precision inertial reference system to derive basic meteorological parameters like pressure, temperature, and the 3-D wind vector, as well as aircraft position and attitude. Water vapor mixing ratio and further derived humidity parameters are measured by SHARC (Sophisticated Hygrometer

for Atmospheric ResearCh) based on direct absorption measurement by a tunable diode laser (TDL) system. Typical absolute accuracy of the basic meteorological data is 0.5 K for temperature, 0.3 hPa for pressure, 0.4-0.6 m s$^{-1}$ for wind and 5% (+1 ppm) for water vapor mixing ratio. All aerosol concentration data were normalized to standard temperature and pressure (STP, $T_0$ = 273.15 K, $p_0$ = 1013.25 hPa). Most of the aerosol sampling was conducted through the HALO aerosol submicrometer inlet (HASI), which pro-

vides up to 30 l min$^{-1}$ sample air flow divided over four sample lines. The air stream sampled on top of the fuselage is aligned with the inlet using a front shroud and decelerated by a factor of approximately 15, providing near-isokinetic sampling to the aerosol instruments mounted inside the aircraft cabin (Andreae et al., 2018).

     The characterization of refractory black carbon (rBC) particles at high time resolution was conducted

using an eight-channel single particle soot photometer (SP2, Droplet Measurement Techniques, Longmont, CO, USA) (Stephens et al., 2003; Schwarz et al., 2006). The instrument measures the time-dependent scattering and incandescence signals produced by single aerosol particles when crossing a Gaussian-shaped laser beam (Nd:YAG; wavelength $\lambda$ = 1064 nm) (Schwarz et al., 2006). The avalanche photo-diode (APD) detectors measure at high and low gain stages the aerosol particle light scattering and incandes-

cence in two wavelength ranges ($\lambda$ = 350 − 800 nm and $\lambda$ = 630 − 880 nm). All particles scatter the laser light with an intensity that is proportional to their optical size, from which the optical diameter ($D_o$) is determined. The instrument detects purely scattering particles in the size range of 200 nm < $D_o$ < 400 nm. Particles containing sufficient mass of rBC absorb the laser light and are heated to their vaporization temperature (~4000 °C), emitting incandescence light. The peak intensity of the incandescent signal is linearly

proportional to the mass of rBC in the particle, which is determined after applying a calibration factor (Laborde et al., 2013). Assuming a void-free density of 1.8 g cm$^{-3}$, the mass-equivalent diameter ($D_{MEV}$) of



rBC cores is calculated from the measured rBC mass (Laborde et al., 2013). The SP2 measurements are sensitive to rBC cores in the nominal size range of 70 nm $< D_{MEV} <$ 500 nm. The SP2 incandescence signal was calibrated at the beginning, during, and at the end of the campaign, using size-selected fullerene

soot particles. The scattering signal was calibrated using spherical polystyrene latex spheres (208, 244, and 288 nm) and ammonium sulfate particles with diameters selected by a differential mobility analyzer (DMA, Grimm Aerosol Technik, Ainring, Germany). The results of all calibrations agreed within their uncertainty ranges, confirming good instrument stability throughout the campaign.

The concentration of condensation nuclei, $N_{CN}$, was measured using two butanol-based condensation

particle counters (CPC, Grimm Aerosol Technik) with different nominal lower cutoff particle diameters (10 nm and 4 nm, respectively). Due to losses in the inlet lines, the effective cut-off diameters were 10 nm at lower atmospheric levels and 20 nm in the upper troposphere (UT). Accordingly, total aerosol concentrations will be represented by $N_{CN,20}$. An additional CPC ($D_p >$ 10 nm) was connected to a thermodenuder, which heats a segment of the sample line to 250 °C. The thermodenuder is used to evaporate the volatile

aerosol constituents, such as organics and ammonium sulfate salts, allowing to quantify the non-volatile (or refractory) particles (e.g., mineral dust, black carbon, sea salt) (Clarke, 1991; Weinzierl et al., 2011). In addition, the number size distributions of aerosol particles in the size range of $D_p$ = 90 - 600 nm were obtained from an ultra-high sensitivity aerosol spectrometer (UHSAS; Droplet Measurement Technologies, Longmont, CO, USA) (Cai et al., 2008). In this paper, we refer to the total number concentration measured

by the UHSAS as the accumulation mode number concentration, $N_{acc}$. The ultrafine fraction is obtained as the difference between the CPC particle counts, $N_{CN,20}$, and the $N_{acc}$ obtained by the UHSAS. Likewise, the volatile fraction is obtained from the difference between aerosol counts measured by the two CPCs ($D_p >$ 10 nm) with and without a thermodenuder.

The CCN concentrations, $N_{CCN}$, were measured with a two-column CCN counter (CCNC, model

CCN-200, DMT, Longmont, CO, USA) (Krüger et al., 2014; Roberts and Nenes, 2005; Rose et al., 2008). In this study, we used only the measurements at constant supersaturation ($S$ = 0.52 ± 0.05 %). The activated fraction, $f_{CCN}$, was calculated by dividing $N_{CCN}$ over $N_{CN,20}$.

A dual-cell ultraviolet (UV) absorption detector (TE49C, Thermo Scientific) operating at a wavelength of $\lambda$ = 254 nm was used to measure $O_3$ with precision of 2 % or 1 ppb. The CO was detected with a

fast-response fluorescence instrument (AL5002, Aerolaser, Garmisch, Germany) (Gerbig et al., 1999). NO and total reactive nitrogen, $NO_y$ were measured by a modified dual-channel chemiluminescence detector (CLD-SR, Ecophysics) in connection with a gold converter (Baehr, 2003; Ziereis et al., 2000). The BC enhancement ratio relative to CO ($EnR_{BC}$ = $\Delta M_{rBC}/\Delta CO$, where $\Delta$ is the difference between the concentration of the species in the plume and in the background atmosphere) was obtained by applying a bivariate fit to

the rBC and CO correlation within different atmospheric layers. The 5[th] percentiles of the data were used as background values. More details on the measurement techniques can be found in Andreae et al. (2018).


### 2.3. Ground-based aerosol and trace gas measurements at ATTO

The ATTO site was established in 2010/2011 as a research platform for in-depth and long-term measure-
ments of aerosol particles and trace gases as well as meteorological and ecological parameters in the cen-
tral Amazon rain forest (Andreae et al., 2015). The research site is located 150 km northeast of Manaus, in
a region characterized by periodic pristine atmospheric conditions during parts of the wet season versus
strong BB pollution during the dry season (Pöhlker et al., 2016, 2018; Saturno et al., 2018b). The present
study includes ATTO data of the aerosol absorption coefficient at $\lambda = 637$ nm, $\sigma_{ap}$, using the Multiangle
Absorption Photometer (MAAP, model 5012, Thermo Electron Group, Waltham, USA) and the aerosol
scattering coefficients, $\sigma_{sp}$, using a nephelometer (model Aurora 3000, Ecotech Pty Ltd., Knoxfield, Aus-
tralia), respectively. The $M_{BCe}$ was calculated using a mass absorption cross section (MAC) of 12.3 m$^2$ g$^{-1}$
for the dry season, as obtained by Saturno et al. (2018b). The single scattering albedo (SSA), which char-
acterizes the absorption properties of an aerosol population, is defined as scattering divided by total ex-
tinction (absorption + scattering). All data were normalized to standard temperature and pressure (STP, $T_0$
= 273.15 K, $p_0$ =1013.25 hPa). The CCN concentrations at a supersaturation of 0.5 %, $N_{CCN}$ ($S = 0.5$ %),
were calculated using long-term scanning mobility particle sizer (SMPS) data and the κ-Köhler parametri-
zation as described in Pöhlker et al. (2016). For more details about the aerosol optical properties character-
ization and CCN observations we refer to Saturno et al. (2018b) and Pöhlker et al. (2016, 2018), respec-
tively. Further details on CO measurements conducted at ATTO site can be found in Winderlich et al.
(2010) and Andreae et al. (2015). Daily $EnR_{BC}$ was calculated by applying a bivariate fit to 30-min aver-
ages of rBC and CO correlation.

### 2.4. Satellite and ground-based remote sensing

The Cloud-Aerosol Lidar with Orthogonal Polarization (CALIOP) lidar system, onboard the Cloud-Aero-
sol Lidar and Infrared Pathfinder Satellite Observations (CALIPSO) satellite, measures the intensity and
orthogonally polarized components of the backscatter signal at two wavelengths ($\lambda = 532$ nm and $\lambda = 1064$
nm) with 30 m vertical resolution up to 30 km altitude (Winker et al., 2009). In this study, we used the LI-
DAR Level 2 Version 3 Aerosol Profile product with 5 km horizontal resolution, which includes the verti-
cally-resolved extinction coefficients. The CALIPSO models further classify the detected aerosol layers
into subclasses: polluted continental, biomass burning (smoke), desert dust, polluted dust, clean continen-
tal, and marine aerosol, using the observed physical and optical properties (Omar et al., 2009).

Further aerosol particle and trace gas satellite products used in this study were obtained from the Gio-
vanni online data system (https://giovanni.gsfc.nasa.gov/giovanni/, last access on 13 June 2019). The At-
mospheric Infrared Sounder (AIRS) onboard the NASA Aqua satellite provides measurements of aerosol
particle and trace gas properties (e.g., $CO_2$, CO, $CH_4$, $O_3$, $SO_2$) at several atmospheric levels. In this study,





we used the CO measurements between the 400 and 600 hPa pressure levels. The Moderate Resolution Imaging Spectroradiometer (MODIS) aerosol products from the NASA Terra and Aqua satellites provide spectral aerosol optical depth (AOD) at several wavelengths, ranging from $\lambda$ = 0.41 - 15 µm (Remer et al.,

2005). In this study, we used daily averages of the 550 nm AOD (level 3) with original grid resolution of 1º x 1º.

The AErosol RObotic NETwork (AERONET, https://aeronet.gsfc.nasa.gov/, last access 12 Mar 2019) is a global network of ground-based sun photometers that measure solar radiance at several wavelengths, typically 340, 380, 440, 500, 670, 870, 940, and 1020 nm (Holben et al., 1998). Aerosol properties such as

AOD and its wavelength dependence, the extinction Ångström exponent (EAE), are obtained from direct sun measurements. We use level 2.0 data (quality assured) of direct products in Ascension Island (7.976º S, 14.415º W), an AERONET sampling station in operation since 1996, to study the seasonality of aerosol concentrations in the middle of the South Atlantic Ocean.

**2.5.   Direct Radiative forcing at the top of the atmosphere**

In this study, we calculate the direct radiative forcing at the top of the atmosphere (DRF-TOA) by aerosol particles in the BB layer in the region of the South Atlantic Ocean. The DRF-TOA was calculated using the library for radiative transfer (LibRadtran) (Emde et al., 2016) with the uvspec tool. The Discrete Ordinate Radiative Transfer solver (DISORT) 2 was chosen to solve the radiative transfer equation (Evans,

1998; Tsay et al., 2000). The setup for the atmosphere was based on the standard tropical profile (Anderson et al., 1986), which was modified with measurement data. The vertical profiles of mean aerosol extinction coefficient were calculated based on multiyear (2012-2018) CALIPSO retrievals. A wavelength range from 300 to 4000 nm was considered. The extraterrestrial spectrum was used as described in Gueymard (2004). The *ocean* was set as underlying surface. With respect to the aerosol optical properties,

the AOD of the plume was calculated by integrating the mean extinction coefficient over the altitude band of the pollution layer (1 – 5 km). A SSA of 0.84 was assumed for the smoke layer based on Zuidema et al., (2016) and the section 3.5. of the present study. An asymmetry parameter of 0.7 was used based on the typical BC value presented in Cheng et al. (2014). With this, we obtained the mean daily value for the DRF-TOA along different longitudes.





### 2.6. Backward trajectory modelling and fire intensities

The NOAA hybrid single-particle Lagrangian integrated trajectory (HYSPLIT) model (Stein et al., 2015) was used to obtain systematic and multiyear sets of backward trajectories (BTs) for the ATTO site as outlined in detail in Pöhlker et al. (2019). The time series of cumulative fire intensity along the BTs was calculated based on (i) an ensemble of three-day HYSPLIT BTs, started every hour in the time frame between 01 January 2013 and 31 December 2018, at a starting height of 1000 m, and (ii) daily georeferenced fire intensity maps (unit W m$^{-2}$) from the Global Fire Assimilation System (GFAS). Details on the BT data set can be found in Pöhlker et al. (2019). The GFAS fire intensity maps were obtained as NetCDF3 files with a spatial resolution of 0.1° latitude by 0.1° longitude (0.1° equal roughly 11 km). The fire intensity calculation was conducted as follows: Each two consecutive points of the trajectories build one linear segment of a trajectory. For obtaining the time series of the cumulative fire along BTs (CF$_{BT}$), the twenty-four trajectories for each day are mapped to the raster of fire intensities of the corresponding days (see example in Fig. S1). For each trajectory, the collection of grid cells that the trajectory passes through is computed. This is done by finding all locations along the trajectory for which either the latitude or longitude coordinate is an integer multiple of 0.1°. As the trajectories pass through grid cells at different positions and with different angles, the influence of the grid cells is not equal. To account for the strength of influence of each grid cell,  the length of the trajectory path within the cell is calculated for each of the passed grid cells and multiplied by the fire intensity corresponding to the grid cell. These fire intensities, divided by the length of the trajectory path, are summed up for each trajectory segment separately and then divided by the length of the corresponding trajectory segment. This corresponds to a second normalization of the fire intensity to the speed of the air parcel along the segment. Finally, the fire intensities are summed over the whole trajectory length to represent a cumulative fire intensity along every individual trajectory.

### 2.7. GIS data products and analysis

The analysis of geographic information system (GIS) data sets was conducted with the QGIS software package ('Las Palmas' version 2.18.2, QGIS development team). The GIS data sets were handled using the coordinate reference system WGS84 (world geodetic system from 1984). The following GIS data sets were used in this study: (i) maps of global water bodies obtained from the European Space Agency (ESA) (https://www.esa-landcover-cci.org/?q=node/162, last access 04 Jul 2019), (ii) wind fields from the Modern-Era Retrospective analysis for Research and Applications Version 2 model (Merra-2, https://gmao.gsfc.nasa.gov/reanalysis/MERRA-2/, last access 04 Jul 2019) obtained through the Giovanni online data system, (iii) land cover maps obtained from ESA (http://maps.elie.ucl.ac.be/CCI/viewer/index.php, last access 04 Jul 2019), and (iv) a map of global biomes according to  Olson et al. (2001). For further details, we refer to Pöhlker et al. (2019).

### 3. Results and discussion

330    The ACRIDICON-CHUVA flight AC19 on 30 September 2014 probed the vertical atmospheric structure
over the northeastern (NE) region of the Amazon Basin. Its flight track followed the direction of the Amazon River from Manaus towards the coast and included cloud profiling maneuvers over the Atlantic Ocean
(Fig. 1). A remarkable observation during AC19 was the strong stratification of the troposphere over the
ocean with vertically well-defined and horizontally extended layers, with varying degrees of pollution (Table 1). Specifically, we distinguished an upper and a lower pollution layer (UPL and LPL) with a horizontal clean air mass layer (CL) in between. Figure 2 shows that the layers were readily recognized visually
from the aircraft's cockpit. The UPL is characterized by maxima in accumulation mode particle and BC
concentrations as well as peaks in CO and $O_3$ mole fractions. The LPL is characterized by a maximum in
particle number concentration in the Aitken and nucleation mode size range along with relatively lower
340    abundance of the combustion tracers BC and CO. In this study, we present the tropospheric stratification
for the lowest 5 km of the atmosphere, focusing primarily on aerosol and trace gas properties within the
UPL in contrast to the properties of the CL and LPL, which are included in the discussion for comparison.
Aerosol properties in the UT during ACRIDICON-CHUVA have been characterized in previous studies
by Andreae et al. (2018) and Schulz et al. (2018). Upon ascent and descent, the UPL was probed six times
345    at offshore locations[2], right before it reached the South American continent, and two times onshore ~200-
400 km from the coast line (blue squares in Fig. 1). The eight UPL penetrations were several hundred kilometers apart from each other, underlining the large horizontal extent of the layer. Later on, an active fire
plume was observed during AC19 northwest of Belem on the route back to Manaus airport (green square
in Fig. 1, photo of plume in Fig. S2). This plume was intercepted at ~ 1 km above the fire and is expected
to be only a few minutes old.  In the next sections, the aerosol properties in this local fresh BB plume are
contrasted with the UPL aerosol.

### 3.1. Offshore aerosol particle and trace gas profiles

The pronounced tropospheric stratification observed over the Atlantic Ocean near the NE margin of the
Amazon Basin is illustrated by selected meteorological, trace gas, and aerosol profiles in Fig. 3. In Fig. 3a,
the profiles of water vapor mass mixing ratio, $q$, and potential temperature, $\theta$, show rather small interquartile ranges, indicating comparable $q$ and $\theta$ conditions along the flight track where profiling maneuvers
were conducted. In relation to $q$ and $\theta$, a well-defined layering – particularly the UPL – clearly emerges in
the aerosol particle and trace gas properties (Fig. 3). Generally, the profile of $\theta$ indicates rather stable conditions along the entire profile with the UPL being centered at ~3.5 km altitude (Babu et al., 2011). For

---

[2] Note that two penetrations of the layer over the extent Amazon River delta were counted as offshore here.


comparison, radiosonde profiles at Belem airport (Figure S3) clearly show similar tropospheric stratification: the first layer (top around 1000 m) is associated with the boundary layer processes; the second (top around 3200 m) is related to the shallow clouds top and the third one (around 5000 m) is the large scale inversion. The stable conditions presumably prevented the pollution from being mixed downwards and

further suggest that the UPL is decoupled from the air masses above and below, facilitating an efficient horizontal transport pathway for the pollutants. Moreover, the distinct properties of UPL, CL, and LPL as outlined below, suggest that the corresponding air masses originated from different sources and/or processes and probably reflect different atmospheric aging times (see also Sect. 3.2). For example, shallow convection (or Scu) can increase aerosol at the top of clouds through detrainment.

370        In terms of aerosol properties, the UPL is characterized by a relative maximum in total number concentrations, $N_{CN,20} = 970 \pm 260$ cm$^{-3}$ (mean ± std), as shown in Fig. 3b. Aerosol particles in the accumulation mode dominate the UPL aerosol, as $N_{acc} = 850 \pm 330$ cm$^{-3}$ accounts for most of $N_{CN,20}$ (~85 %). This corresponds to a significant drop in the ultrafine particle fraction with $f_{fine} \approx 15$ % within the UPL (Fig. 3b). The aerosols in the UPL are further characterized by a low fraction of volatile particles, $f_{vol}$, as shown

in Fig. 3c. In the atmospheric column, $f_{vol}$ reaches its minimum of $16 \pm 9$ % within the UPL and generally shows a similar profile as $f_{fine}$, indicating a rather aged plume. Accordingly, the CCN concentrations at $S = 0.5$ %, $N_{CCN}(S = 0.5$ %), show a maximum within the UPL with $N_{CCN}(S = 0.5$ %) $= 560 \pm 180$ cm$^{-3}$ as well as high activated fraction, $f_{CCN}(S = 0.5$ %) $= 60 \pm 6$ % (Fig. 3d). In comparison to typical conditions at the ATTO site in central Amazonia, $N_{acc}$ within the UPL is lower than under strongly BB-influenced ($N_{acc,BB} \approx$

3400 cm$^{-3}$) and average dry season conditions ($N_{acc,dry} \approx 1300$ cm$^{-3}$) at ATTO, however, substantially larger than under average wet season ($N_{acc,wet} \approx 150$ cm$^{-3}$) and pristine rain forest conditions ($N_{acc,PR} \approx 90$ cm$^{-3}$) (Pöhlker et al., 2016, 2018). Remarkably, rBC particles represent a dominant species of the UPL aerosol population in terms of number concentration with $N_{rBC} = 280 \pm 110$ cm$^{-3}$, corresponding to an rBC number fraction of $f_{rBC} \approx 40$ % relative to $N_{CN,20}$ (Fig. 3e and Fig.4). The $f_{rBC}$ in the UPL belongs to the highest val-

ues encountered during flight AC19 and is much higher than $f_{rBC} \approx 5$ % in the probed fresh BB plume (Fig. 4). The high $f_{rBC}$ further agrees with the pronounced brownish color of the visually observable layer in Fig. 2. For comparison, rBC fractions of $0 - 15$ % relative to $N_{CN,20}$ were observed in megacity pollution (Laborde et al., 2013) and $f_{rBC} \approx 6$ % in wildfires plumes injected into the lowermost stratosphere in the northern hemisphere (Ditas et al., 2018). The rBC mass concentrations within the UPL, $M_{rBC} = 1.0 \pm 0.4$

µg m$^{-3}$ (ranged from 0.5 to 2 µg m$^{-3}$), reach up to the highest BC levels observed at ATTO ($M_{BCe}$ up to 2.5 µg m$^{-3}$) (Pöhlker et al., 2018). These findings, in combination with its large geographic extent, suggests that the UPL may represent an aerosol reservoir of particular significance for the Amazonian aerosol cycling and radiative budget.

        Regarding trace gases within the UPL, Fig. 3f – h shows absolute maxima for the mole fractions of

carbon monoxide ($c_{CO}$), ozone ($c_{O3}$), and total reactive nitrogen ($c_{NOy}$) as well as a secondary maximum for





nitrogen monoxide ($c_{NO}$). The elevated $c_{CO} = 150 \pm 30$ ppb along with the high $M_{rBC}$ indicates that the UPL air masses originated from BB emissions. Moreover, the ratio between these two co-emitted species can be used as tracer for the origin and age of BB plumes (Darbyshire et al., 2019; Guyon et al., 2005; Saturno et al., 2018b). The aged UPL is characterized by a higher rBC enhancement ratio, $EnR_{BC} = 12.4 \pm 0.2$ ng m⁻³

ppb⁻¹ compared to fresh Amazonian BB, with $EnR_{BC}$ of $7.8 \pm 0.1$ ng/m⁻³ ppb⁻¹ (Fig. S4). Recent aircraft measurements of African BB pollution over Ascension Island have found similar $EnR_{BC} = 11 - 17$ ng m⁻³ ppb⁻¹ in the free troposphere (Wu et al., 2019). The ozone as secondary pollutant – generated photochemically in BB plumes during atmospheric transport – also presents a maximum within the UPL ($c_{O3} = 56 \pm 9$ ppb) and appears to be anti-correlated with NO ($c_{NO} = 0.10 \pm 0.02$ ppb). In fact, the main pathway of the

$O_3$ formation is catalytically driven by nitrogen oxides ($NO_x = NO + NO_2$) in the presence of solar radiation (Andreae et al., 1994; Baylon et al., 2015; Jaffe and Wigder, 2012; Logan, 1983; Mauzerall et al., 1998; Val Martín et al., 2006). Therefore, the fact that $O_3$ and $NO_y$ ($c_{NOy} = 2.5 \pm 0.8$ ppb) are strongly enhanced in the pollution layers, reflects the photochemical age of the plume. Overall, the trace gas mole fractions within the UPL are consistent with previous aircraft measurements. Over the Atlantic, off the city

of Natal, Brazil, Andreae et al. (1994) found similar pollution layers with $c_{O3}$ and $c_{CO}$ up to 90 and 210 ppb, respectively. The mean mole fraction of $NO_y$ in these plumes was extremely high: $4.4 \pm 3.1$ ppb, with enhancement ratios, $EnR_{NOy}$, in the range 0.018 to 0.108. The $EnR_{NOy}$ in the UPL (0.019) lies at the lower part of this range. Over Ascension Island, $c_{O3}$ can be as high as 80 ppb in the lower troposphere (Thompson et al., 1996).

415        Below the UPL, the atmospheric vertical profile off the Brazilian coast shows a second maximum in aerosol concentrations in what we call the LPL ($N_{CN,20} = 1300 \pm 200$ cm⁻³; $N_{acc} = 650 \pm 140$ cm⁻³) at altitudes between ~2.3 to 3.0 km (Fig. 3). The properties of the UPL and LPL, however, are remarkably different. The LPL shows rather lower concentrations of rBC, ($M_{rBC} = 0.36 \pm 0.11$ µg m⁻³ and $N_{rBC} = 110 \pm 20$ cm⁻³), CO ($c_{CO} = 105 \pm 5$ ppb) and $O_3$ ($c_{O3} = 45 \pm 2$ ppb), which decreases with decreasing altitude.

$NO_y$ actually reaches the highest concentrations in this layer, with values up to 3.0 ppb. We assume that the pyrogenic species found in the LPL are also advected from Africa, however, possible influences from urban emissions in Africa and/or South America, for example, should not be neglected. This possibility is supported by the relatively high sulfate content of the aerosol in this layer, which at an average value of 1.38 µg m⁻³ accounts for 31% of total aerosol mass concentration. Sulfur-rich anthropogenic emissions

from fossil-fuel combustion may have become mixed with BB emissions by cloud-venting over the Gulf of Guinea region (Dajuma et al., 2019).

        One interesting aspect of the LPL is that the ultrafine fraction accounts for about half of aerosol number concentration. Likewise, a higher $f_{vol}$ was observed. One possible explanation for that is that new particle formation occurs in the detrainment regions around the shallow cumulus, which brings air masses from



the marine boundary layer (MBL), containing dimethyl sulfide (DMS) and $SO_2$, into the LPL. This phenomenon has been previously reported by several authors (Hegg et al., 1990; Kerminen et al., 2018; Perry and Hobbs, 1994). Direct convective transport of ultrafine particles from the MBL into the LPL is unlikely as an important source of such particles, as their concentration in the MBL is only about 200 $cm^{-3}$, well below their concentration in the LPL of about 700 $cm^{-3}$. In the MBL (with top at ~600 m asl), the total and

accumulation mode particle concentrations are somewhat lower than in the layers aloft ($N_{CN,20} = 420 \pm 160$ $cm^{-3}$ and $N_{acc} = 230 \pm 50$ $cm^{-3}$). The MBL appears to be less influenced by the African BB, with $M_{rBC} = 0.18 \pm 0.07$ µg m$^{-3}$ and $N_{rBC}$ accounting for only 10 % of the $N_{CN,20}$. Additionally, the aerosol population in the MBL appears less efficient as CCN, with only 20 % of particles being activated at $S = 0.5\%$.

   In between the UPL and LPL, the CL was found centered at ~3.2 km altitude (~200 m thick) with rela-

tively dry air as represented by a sharp decrease in $q$. Such clean layers have been previously observed in the dry season over the African continent and adjacent oceans, specifically in the southeastern Atlantic Ocean, with a few hundred (up to 1 km) meters thickness (Hobbs, 2003). Within the CL, the combustion tracer concentrations $M_{rBC}$, $N_{rBC}$, and $c_{CO}$ sharply decrease to $0.09 \pm 0.04$ µg m$^{-3}$, $30 \pm 12$ $cm^{-3}$ and $83 \pm 4$ ppb, respectively. We further found $N_{CN,20} = 500 \pm 60$ $cm^{-3}$, which is comparable to $N_{CN} = 500$ $cm^{-3}$ in an-

other CL as reported by Hobbs (2003). The $c_{O3}$ shows a smaller decrease to $48 \pm 2$ ppb. Hobbs (2003) proposed that the CL derived from the ultraclean upper tropospheric air.

   In the vicinity of the western African coast, similar tropospheric stratification has been observed in a recent aircraft campaign. Aerosol and trace gas profile measurements over the Gulf of Guinea in July 2016, measured under the influence of aged BB plumes originating in central Africa, revealed two distinct

aerosol layers, where the upper one, centered at 3.8 km altitude, was enriched in rBC (~0.3 µg m$^{-3}$), CO (~340 ppb), and OA (~65 µg m$^{-3}$) (Denjean et al., 2019; Flamant et al., 2018). Moreover, Weinzierl et al. (2011) reported $f_{vol} = 26$ % $(13 - 52 \%,\ p_3 - p_{97})$ for BB plumes near the western African coast in 2008 during SAMUM-2, which are larger than the $f_{vol}$ within the UPL in this study. This suggests that most of the volatile material normally present at higher proportions within BB plumes, is being removed during

atmospheric transport. Typically, organic aerosol becomes less volatile during atmospheric aging, concurrent with an increase in its O/C ratio (Grieshop et al., 2009; Isaacman-VanWertz et al., 2018; Slowik et al., 2012; Zhou et al., 2017). More recently, from comprehensive measurements on Ascension Island in the middle of the South Atlantic, Zuidema et al. (2018) described a smoke layer over a stratocumulus cloud deck where "shortwave-absorbing aerosol emanating from biomass burning in continental Africa advects

westwards over the southern Atlantic for approximately one third of the year, from June to October". These previously reported results suggest that the discrete rBC-enriched layer reported here has likely formed in the vicinity of the African coast and tapered as it approaches the South American continent. In addition, the atmospheric stability facilitated the transport across the South Atlantic towards the Amazon. Further, ongoing studies suggests the existence of a low level jet (LLJ) with a maximum around 800 hPa





(~ 3 km) induced by changes in boundary layer (BL) height during day (Anselmo et al., 2019 in prep.). The LLJ may be the main mechanism transporting the African pollution from the Atlantic Ocean into the Amazon Basin.

### 3.2.  Backward trajectories and potential source regions in Africa

The large-scale trade wind circulation over the tropical Atlantic region is defined by the position of the intertropical convergence zone (ITCZ), which oscillates north-southwards over the year. The ITCZ movement is a main factor of the pronounced aerosol seasonality in the central Amazon (Andreae et al., 2012; Martin et al., 2010; Moran-Zuloaga et al., 2018; Pöhlker et al., 2019, 2018; Saturno et al., 2018b). Based on BTs, which reflect the large-scale trade wind circulation patterns, we investigated the origin and age of

the UPL up to 10 days prior to its observation. Figure 5a compares BT ensembles – started in the relevant offshore area (i.e., 3° N to 3° S; 52° W to 44° W) and during the time period of the UPL observation by the aircraft (i.e., 2014-09-30 18:00 UTC) – for the starting heights at 500 m (representing near-surface conditions), 2500 m (peak of LPL), 3500 m (peak of UPL), and 5000 m above ground level (well above the UPL). The comparison of the different BT starting heights shows clear differences in air mass advec-

tion patterns: specifically, the UPL BTs indicate rather fast and directed air mass movements from easterly directions, whereas the LPL BTs indicated more curvilinear movement from east-southeasterly directions. The fact that the BT patterns diverge is consistent with the different trace gas and aerosol properties shown in Fig. 3, underlining that LPL and UPL represent air masses of different origin and/or atmospheric aging history. Furthermore, the wind fields at about 3500 m – which agree well with the BT patterns – illustrate

the large-scale meteorology at the UPL altitude. The transatlantic transport time of the UPL air masses was about 10 days according to the BT ensembles.

    The fire map in Fig. 5a helps to identify potential source regions of the UPL aerosol. It includes fires within a five-day window (15 to 20 September 2014) when the BB-laden air masses likely originated in the African source regions and then started their ~10-day journey across the Atlantic Ocean until the air-

craft observation on 30 Sep. Several hot spots of fire activity in Central and Southern Africa can be found in Fig. 5a$_1$. All of them are located in tropical and subtropical grasslands, savannas, and shrublands according to Olson et al. (2001). Particularly, the Miombo woodlands are well known as a region of frequent and intense fire activities, mostly driven by human activities (Andela and van der Werf, 2014; Barbosa et al., 1999; Earl et al., 2015). For the time frame of 15 to 20 Sep 2014, satellite-based natural color reflec-

tance images further underline the high fire activity through a larger number of clearly visible smoke plumes in the hot spot areas (one example is shown for 19 Sep 2014 in Fig. 5a$_2$). Note that at the same time, the fire activity in South America is still comparatively low (Fig. 5a$_1$). Overall, the fires, wind field, and BTs in Fig. 5a$_1$ show a coherent picture and suggest that the shown grassland, savanna and shrubland fires represent the sources for the UPL at the Brazilian coast.



It has been generally assumed – though not shown in detail and quantified yet – that African smoke accounts for a significant fraction of pollution input into the Amazon Basin (Saturno et al., 2018b). Therefore, we complemented the case-specific map in Fig. 5a₁, which focuses on the precise time window of flight AC19, by a seasonally average map in Fig. 5b, presenting the large-scale circulation patterns for multiyear September averages, during the peak in central and southern African fire activities (represented

as CO in Fig. S5). The map in Fig. 5b emphasizes the large extent of the African smoke plume (here represented by CO) over the Atlantic area. It further shows good agreement between the CO plume pattern and BT ensembles at the central Amazonian ATTO site, indicating that transatlantic smoke transport is a general and seasonally recurring phenomenon beyond the specific case of the analyzed layer in September 2014. This is supported by in-situ measurements made during the Atmospheric Tomography mission

(ATom) 2016-2018 flight missions, which showed elevated concentrations of BB aerosols (about 0.1 to 1 $\mu$g m$^{-3}$) over most the of the southern tropical Atlantic in July to October (Schill et al., 2019). Biomass smoke particles were the dominant aerosol fraction in this region between the surface and 4 km altitude. Further note, that precipitation rates over the southern Atlantic are comparatively low, as shown in Fig. 5b. Accordingly, rain-out and wash-out mechanisms likely do not substantially reduce the BB aerosol popula-

tion during transport.

### 3.3.  Geographic extent of pollution layers over the Atlantic and direct radiative effects

Satellite-based observations resolve the transatlantic transport of the pollution layers. Figure 6 illustrates the movement of the African aerosol particles across the Atlantic by means of CO, where individual

plumes during the dry season of 2014 can be identified. This includes the plume probed at the Brazilian coast during AC19 (marked by a dashed line), which appears to be coincidentally the strongest plume in 2014. Beyond this particularly event, several weaker plumes were also observed. Based on Fig. 6, a characteristic transport velocity of ~380 km d$^{-1}$ can be obtained, and therefore, an aging time of ~10 days. Similarly, satellite-based AOD observations can be used to resolve the plume movement as shown in Fig. S6.

The transatlantic transport of the particular BB plume that was probed during AC19 was temporally and geographically almost coincident with the transatlantic transport of a volcanogenic sulfate-rich plume, which we reported on previously in Saturno et al. (2018a). Here, a period of strong activity of the Nyamuragira volcano in Eastern Congo was associated with major SO₂ emissions, which were oxidized to sulfate during the transatlantic passage. The plume of sulfate-rich aerosols was observed airborne during

ACRIDICON-CHUVA flight AC14 on 21 Sep 2014 in the region from 200-400 km directly south of Manaus as well as by ground-based measurements at ATTO from 21 to 30 Sep 2014. Importantly, the BB plume probed during AC19 and the volcanogenic plume probed during AC14 were distinct events (i.e., did not occur in the same air masses) since the volcanogenic plume (i) occurred ~1 week earlier, (ii) was observed strongest between 4 and 5 km, in contrast to 3 to 4 km for the BB plume, and (iii) showed a very



low rBC mass concentration. However, the volcanogenic plume can be regarded as a "reference case of
        the dynamics and conditions of transatlantic aerosol transport from southern Africa to South America"
        (Saturno et al., 2018a). The temporal coincidence of the volcanogenic and BB plumes further suggests that
        transatlantic aerosol transport was particularly efficient in the second half of September 2014. Moreover,
        the observation that the single volcanogenic plume significantly influenced the aerosol particle chemical

composition, hygroscopicity, and optical properties at ATTO (Saturno et al., 2018a) suggests that the more
        frequent and presumably stronger African BB plumes likely have a similarly profound impact on the cen-
        tral Amazonian aerosol population.

        The African BB plumes (particularly the strong event in the end of September) were also observed
        by the vertically resolved aerosol extinction measurements by the CALIPSO satellite, revealing elevated

and vertically defined smoke layers over the southern Atlantic. Figure 7 shows two selected CALIPSO
        passages on 16 and 30 Sep 2014 (the day of AC19 flight). These passages show exemplary snapshots of
        the elevated smoke layers at different longitudinal locations: on 16 Sep 2014 a layer was probed relatively
        close to the southern African coast, whereas on 30 Sep 2014 a layer was probed halfway between Ascen-
        sion Island and the Amazon River delta. For the overpass on 30 Sep 2014, the layer's N-S extension was

about 1200 km and its altitude between 3 and 4 km, which agrees well with the altitude of the UPL obser-
        vation during flight AC19. For the passage on 16 Sep 2014, the layer's N-S extension was about 4º N to
        20º S (~2800 km) and its altitude between 2 and 5 km. In this context, a dedicated study of Adebiyi and
        Zuidema (2016) has showed that 45 % of the forward trajectories of satellite-detected smoke plumes in
        southern Africa exit the continent westwards between 5º S and 15º S and are transported westward by the

Southern African Easterly Jet (AEJ-S), overlying a semi-permanent marine stratocumulus deck. Moreover,
        Fig. 7 suggest that the layer's latitudinal extent decreases as it approaches the South American continent.

        In order to constrain the seasonal and vertical aspects of the transatlantic transport, we analyzed the
        satellite-retrieved aerosol profiles over the South Atlantic Ocean during the dry season of multiple years
        (2012 to 2018). Figure 8a-c shows the extinction coefficients of all CALIOP overpasses within the region

of interest (ROI, as defined in Fig. 5a) averaged over the months of August, September and October. High
        aerosol loadings (up to 5 km altitude) in the longitude band from 10º E to 20º E correspond to BB emis-
        sions over the African continent. Likewise, comparably high extinction coefficients (up to 3 km altitude)
        are observed due to BB fires in South America (60º W to 40º W). In the region of the South Atlantic
        (40º W to 10º E), the maximum extinction coefficient is observed at two different levels of the atmos-

phere, separated by a relatively clean layer in between. The lower level layer (altitude < 1 km) with pro-
        nounced extinction coefficient depicts the MBL, which is presumably dominated by the (coarse-mode)
        marine aerosols and is clearly visible throughout the three months in Fig. 8. On the other hand, the higher
        level layer (altitudes between 1 and 5 km) represents the African BB aerosol being transported westwards
        over the Atlantic all the way to South America. The transport pattern stands out in the months of August



and September, but is weakened in October, when the remaining BB plumes appears to be mostly/completely removed from the atmosphere half way before reaching South America. The injection height of BB aerosol in Africa is relatively high due to the AEJ-S, which induces an upward motion directly below the jet, enhancing updrafts over land that lift up BB aerosols to altitudes where it can be efficiently transported over the South Atlantic (Adebiyi and Zuidema, 2016). The vertical location of pollution plumes in the at-

mosphere is an important parameter, as it can considerably influence its atmospheric lifetime. The aerosol lifted up to higher altitudes tends to be advected over larger distances due to less efficient removal mechanisms (i.e., wet deposition). When leaving the African coast, the smoke layer is present at altitudes between 1.5 and 5 km, but becomes more restricted to higher altitudes ($3-5$ km) as it moves towards South America. Fig. 8 suggests a pronounced thinning of the layer during its movement westwards due to dilu-

tion.

        The transatlantic transport pattern of African BB, as presented here, is not well represented by the state-of-art atmospheric models. The simulations of the transatlantic transport of BB aerosol by several global aerosol models are able to capture the vertical distribution of aerosol over the African continent, but diverge from the satellite observations as it moves westward over the Atlantic ocean (Das et al., 2017). In

the models, BB aerosol plumes quickly descend to lower levels just off the western African coast, while our observations suggest that they are transported at high altitudes ($< 5$ km) well above the MBL all the way to the Amazon rain forest. After reaching the Brazilian coast, the smoke layer gradually subsides, likely being entrained into the cloud layer below, or more deeply mixing into the boundary layer. The effects of the aged pollution plume on radiative and cloud-nucleation properties over the Atlantic and upon

arrival in the Amazon Basin are still uncertain.

        In order to estimate the direct radiative effect that the African BB layer exerts along its transport over the Atlantic, we used the mean CALIPSO profiles to obtain information about the vertical extension of the plumes. Figure 8d shows the longitudinal profile of DRF-TOA calculated for different AOD values and atmospheric conditions as a daily mean value for three different months. We observed a decrease of the

warming effect of the aerosol towards South America. Near the African coast, a positive DRF-TOA was observed, reaching values as high as $+0.6$ W m$^{-2}$ in August, $+0.4$ W m$^{-2}$ in September and $+0.17$ W m$^{-2}$ in October. On the other hand, negative DRF-TOA, ranging from $-0.03$ to $-0.10$ W m$^{-2}$, was found through the three months for lower AOD values (longitudes $< 5°$ E). The change in sign of the DRF-TOA are mostly given by the high amount of absorption assumed for the aerosol layer in the simulations (single

scattering albedo of 0.84). For low AOD values, the absorption is not too strong and the back-scattered radiation makes some effect, as expressed by the negative aerosol radiative forcing at the TOA. For higher optical depths, the absorption in the layer becomes dominant, thus causing a positive forcing (warming) at TOA. Sensitivity cases for different assumptions on aerosol and surface properties are shown in Fig. S7. Our results suggest that the transport of BB smoke trough the Atlantic has strong direct effects on regional





radiative balance, which changes from warming to cooling in the way from South America.
       ica.

### 3.4. Transport of the pollution layer into the Amazon Basin

An important question for the aerosol cycling in the Amazonian troposphere is how the UPL evolves as it

moves from the Atlantic Ocean into the South American continent. With the data available from flight
       AC19 (six offshore and two onshore penetrations of the UPL, Fig. 1) at least some conclusion can be
       drawn for the first ~400 km of UPL (as well as CL and LPL) transport over land. Figure 9 overlays on- vs.
       offshore vertical profiles of selected meteorological, aerosol and trace gases parameters. Meteorologically,
       the $\theta$ profiles show some divergences between in- and offshore in the well-mixed continental boundary

layer (up to ~1.5 km altitude) and in the UPL altitude band. Even stronger differences were found in the $q$
       profiles, which show the broadening of the upper dry layer from off- to onshore profiles. Regarding the
       CL, the characteristic trace gas and aerosol concentration minima in the offshore profiles mostly disap-
       peared onshore (e.g., for CCN, $M_{rBC}$ and $c_{CO}$, Fig. 9c, d, e). This fading of the CL minimum appears to be
       linked to the evolution of UPL's vertical structure, which mostly remains intact, however shows a ten-

dency to broaden and subside into the altitude range of the CL (i.e., 3.1 to 3.3 km). We suppose that these
       changes represent the onset of a convective broadening of the UPL over the continent and a related en-
       trainment of its aerosol population through the underlying cloud layers into the convective boundary layer.
       In addition, the inshore profiles also suggest that some BB emissions have been added as the airmasses
       moved inland because the increased concentrations of CCN, rBC, and CO in the LPL cannot be explained

by downward mixing from the UPL alone. This is consistent with the presence of scattered fires in the
       coastal region (Fig. 1).

       In this context note that a pronounced UPL was exclusively observed during flight AC19 (the only
       flight over the NE basin), whereas no comparable stratification was found during the other 13 flights in the
       central and western parts of the basin (Wendisch et al, 2016). Also, upon ascent after take-off and descent

before landing of AC19 at Manaus International Airport, the altitude range (i.e., 3 to 4 km) was probed,
       however, no clear indications for an UPL were found. This suggests that the UPL started broadening over
       the first 400 km over the continent and faded away when the air masses reached the central basin in the
       region of Manaus (~1200 km from the coast). Accordingly, we conclude that convective mixing prevented
       that a well-defined UPL reached far into the central basin. This further suggests that the UPL aerosol is

presumably mixed downwards through convection and then transported further westwards with the bound-
       ary layer air masses. Similar downward mixing of BB emissions by convection over land has been re-
       ported from southern West Africa (Dajuma et al., 2019).





### 3.5. Estimated relevance of pollution layer for central Amazonian aerosol cycling

Considering the large horizontal extension of the pollution layer, it presumably accounts for an important

input of aerosols and trace gases into the Amazon Basin with potentially strong impacts for cloud micro-

physics as well as the atmospheric radiative budget. Here we put the experimental results outlined so far

into a broader context of multiple years of observations at the central Amazonian ATTO site to estimate

the relevance of the pollution layer input for the central Amazonian aerosol life cycle. Therefore, Fig. 10

combines the seasonal variability of relevant parameters to assess the interplay of African vs. South Amer-

ican BB emissions for the observed aerosol abundance and properties at ATTO. The influence of African

BB transport into the basin is represented by the fine-mode AOD at Ascension Island, $AOD_{fine,ASC}$ based

on ~20 years of AERONET observations. Note that Ascension Island is located in the main BT path of the

transatlantic pollution transport and therefore a well-located observational site *en route* (Fig. 5b). The in-

fluence of South American BB for the ATTO observations is represented by a BT data product, $CF_{BT}$,

based on pixel-wise accumulation of fire intensities along individual ATTO BTs over a multi-year period.

Generally, the Amazonian atmosphere is strongly influenced by BB aerosols during the dry season

and to a somewhat lower extent during the flanking transition periods causing significant increases in scat-

tering and absorption coefficients (Rizzo et al., 2013; Saturno et al., 2018b). In Fig. 10, the long-lasting

BB influence can be seen by means of the broad seasonal maxima in $M_{BCe}$, $N_{CCN}$ and $c_{CO}$. The BB-im-

pacted part of the year in the Amazon, including dry season and transition periods, will be called hereafter

the BB season. Remarkably, the African BB influence (represented by $AOD_{fine,ASC}$) vs. the South Ameri-

can BB influence (represented by $CF_{BT}$) shows contrasting and complementary seasonal cycles: $AOD_{fine,ASC}$ has its onset in July, peaks in September, and drops again in October (see also Fig. 6 and 8),

whereas $CF_{BT}$ has its onset in September, peaks in October/November, and drops towards the beginning of

December. The seasonal cycles in $M_{BCe}$, $N_{CCN}$, and $c_{CO}$ vs. $AOD_{fine,ASC}$ and $CF_{BT}$ in Fig. 10 suggests that

the Amazonian BB season can be regarded as consisting of an African smoke dominated period in the first

half and an South American smoke dominated period in the second half of this season. In fact, differences

between the first and second half of BB season have been found and are summarized in Table 2.

In the first half of the BB season, under predominant African influence, seasonally averaged BB tracer

concentrations of $M_{BCe} = 0.36 \pm 0.12$ µg m$^{-3}$ and $c_{CO} = 140 \pm 25$ ppb were observed, whereas in the second

half under predominant South American influence, $M_{BCe} = 0.41 \pm 0.17$ µg m$^{-3}$ and $c_{CO} = 190 \pm 70$ ppb

were somewhat higher. The CCN concentrations increase throughout the dry season, with $N_{CCN} = 1100 \pm 500$ cm$^{-3}$ in the African-BB and $N_{CCN} = 1800 \pm 900$ cm$^{-3}$ in the South American-BB dominated states. In

both periods, the BB-derived aerosol particles show high CCN efficiency, with 83 % and 87 % of particle

activation at S = 0.5 % during the first and second half of the BB season, respectively. Clearer differences

between the two BB properties in the dry season was observed, however, for the single scattering albedo

(SSA) and the BC enhancement ratio, $EnR_{BC}$ (Fig. 10 and Table 2). The aerosol population was strongly



absorbing in August, with a minimum SSA at about $0.85 \pm 0.02$. Subsequently, the absorbing properties decreased towards November with a relative maximum in SSA at about $0.90 \pm 0.03$. These results are consistent with the large $f_{rBC}$ of ~40 % in the African UPL vs. the rather low $f_{rBC}$ of ~5 % in an exemplary Amazonian BB plumes (Sect. 3.1). Moreover, Saturno et al. (2018b) have shown that the brown carbon (BrC) contribution to total absorption becomes increasingly important towards the end of the dry season. This is a further indication of the predominance of regional fires towards the later BB season, given that BrC is quickly photodegraded in the atmosphere after emission, with a typical lifetime of few days to weeks (Fleming et al., 2019; Wong et al., 2019), comparable to the transport times of African BB emissions across the Atlantic.

In the Amazonian dry season, the $EnR_{BC}$ values span from 3.1 to 8.9 ng m$^{-3}$ ppb$^{-1}$ (daily values), with the highest values occurring under the influence of African plumes, and associated with lower SSA (see also Saturno et al., 2018b). The $EnR_{BC}$ decreases from its peak of $7.4 \pm 3.1$ ng m$^{-3}$ ppb$^{-1}$ in July-September to a relative minimum of $4.4 \pm 2.0$ ng m$^{-3}$ ppb$^{-1}$ around November (Fig. 10g). The high $EnR_{BC}$ in African plumes can be attributed to more flaming combustion in the comparatively dry grassland, savanna, and shrubland vegetation in contrast to more smoldering combustion of Amazonian deforestation fires in the moist tropical forests. For comparison, Darbyshire et al. (2019) reported from the SAMBBA aircraft campaign over the southern Amazon basin, $EnR_{BC}$ of 3 ng m$^{-3}$ ppb$^{-1}$ in the west associated with more smoldering combustion of pasture and forested areas, in contrast to $EnR_{BC}$ of 12 ng m$^{-3}$ ppb$^{-1}$ in eastern part influenced by Cerrado fires. Note that differences between ground-based (this section) and aircraft $EnR_{BC}$ (section 3.1) presented in this paper are due to removal processes, combustion phase and, possibly, to the use of different measurement techniques. However, the independent ground-based observations do show a clear decrease in $EnR_{BC}$ from African- to Amazonian-BB dominated states, which is consistent with the aircraft measurements.

## 4. Summary and conclusions

In this study, we probed an event of African BB advection over the Atlantic Ocean off the Amazon Basin with instruments installed onboard the HALO aircraft during the ACRIDICON–CHUVA campaign in September 2014. Vertical profiles over the Atlantic Ocean and inshore over the northeastern Basin revealed a horizontally extended rBC-enriched layer (UPL) of about 300 m thickness at ~3.5 km altitude, which showed strongly elevated aerosol and trace gas concentrations. The plume was dominated by aerosol particles in the accumulation mode size range ($N_{acc} \sim N_{CN,20} = 970 \pm 260$ cm$^{-3}$), consisting mostly of non-volatile material. Remarkably, rBC particles appeared to be the dominant species, accounting for ~40 % of total aerosol number with mass concentrations ($M_{rBC}$) up to 2 μg m$^{-3}$. Along with rBC, high $c_{CO}$ ($150 \pm 30$ ppb) indicated that the layer originated from biomass burning. Moreover, the advanced photochemical aging of the plume was indicated by the elevated $c_{O3}$ ($56 \pm 9$ ppb). Despite the large fraction of rBC,



the aerosol in the UPL appeared to be very CCN efficient, with 60 % of particles being activated at $S = 0.5\%$.

Backward trajectory analysis and remote sensing observations showed that the layer originated from BB in African grasslands, savannas, and shrublands. Therefore, the aerosol within the pollution layer upon arrival in South America, as probed by aircraft, has experienced at least 10 days of atmospheric aging over the African continent and the Atlantic Ocean. Moreover, multi-year remote sensing observations showed that layered atmospheric structures, and also the westward advection of African BB plumes, are a rather

common phenomenon over the South Atlantic during the Amazonian dry season, peaking each year in August and September. Near the African coast, the vertical extent of the layer is a few kilometers and it narrows down to only a few hundred meters while transported to South America. This leads to a change in the direct radiative effect at the top of the atmosphere from positive (warming) to negative (cooling) during the transatlantic transport. The north-south extension of the layer also decreased significantly, as it moved

westwards. While the layered structure prevails all the way across the Atlantic for several days, it becomes quickly mixed out vertically just a few hundreds of kilometers after reaching the South American continent. The aerosol particles in the layer are entrained into the continental boundary layer due to convection and large-scale subsidence. We propose that long-range transport of such layers is the main pathway supplying African CCN and highly aged BC into the Amazonian atmosphere.

Long-term (2013-2018) ground-based aerosol measurements at the ATTO site in central Amazonia have demonstrated that long-range transported BB from Africa exerts an important impact on aerosol particle properties within the dry season. From July to December, the Amazonian atmosphere is strongly influenced by BB aerosols with corresponding increases in scattering and absorption coefficients (Saturno et al., 2018b, Rizzo et al., 2013), as well as the BB tracers BC and CO. The interplay of African versus South

American BB emissions at ATTO is expressed by defined seasonal cycles of single scattering albedo (SSA) that increased from 0.87 in August to 0.90 in November, while $EnR_{BC}$ decreased from $7.4 \pm 3.1$ ng m$^{-3}$ ppb$^{-1}$ to $4.4 \pm 1.9$ ng m$^{-3}$ ppb$^{-1}$. Closely related observations and processes for $O_3$ have been reported in a complementary study by Wolff et al. (2019).

    This study highlights the importance of the transatlantic transport as a source of highly aged rBC and

CCN-active particles to the Amazonian atmosphere, especially the northeastern regions, which are less impacted by anthropogenic pollution. This process clearly merits future modeling investigations to assess its effects on regional radiative forcing as well as cloud properties and lifecycle.



**Data availability**

The aircraft data from the ACRIDICON-CHUVA campaign are available through the HALO database un-

der https://www.halo.dlr.de/halo-db/. The ATTO data used in this study are available through the ATTO

data portal under https://www.attodata.org/. Satellite data used in this study are freely available through

https://giovanni.gsfc.nasa.gov/giovanni/ and https://www-calipso.larc.nasa.gov/products/lidar/browse_im-

ages/production/. For data requests beyond the available data, please refer to the corresponding authors.

**Author contribution**

BAH conducted most of the data analysis and wrote the paper. CP, MOA, and UP supervised the work. QW,

TD, JK, CK, NM, HP, DW, and MW contributed specific parts of the data analysis. MLP, MD, TK, DF,

LATM, DS, HS, BW, AW, MW, HZ, MZ, CP, UP, and MOA conducted the measurements during the

ACRIDICON-CHUVA campaign. BAH, MLP, JS, MS, FD, OK, JB, MAF, FM, JVL, and CP conducted

the measurements at the ATTO site. JD, PA, HB, LATM, RB, YC, JK, OK, JM, and HSu contributed to the

data analysis and interpretation through fruitful discussions and by providing valuable comments and ideas.

All authors contributed to the interpretation of the results and writing the paper.

**Competing interests**

The authors declare that they have no conflict of interest.





**Acknowledgements**

We acknowledge the Conselho Nacional de Desenvolvimento Científico e Tecnológico (CNPq, Brazil), process 200723/2015-4, the Max Planck Graduate Center with the Johannes Gutenberg University Mainz (MPGC) and the Max Planck Society, for the financial support. We thank the entire ACRIDICON-CHUVA team for collecting the data and for the fruitful scientific cooperation. Special thanks goes to the HALO pilots, Steffen Gemsa, Michael Grossrubatscher, and Stefan Grillenbeck. We thank Volker Dreiling, Sensor and Data Team of DLR Flight Experiments and the HALO team of the DLR for their cooperation. We acknowledge the generous support of the ACRIDICON-CHUVA campaign by the Max Planck Society, the German Aerospace Center (DLR), FAPESP (São Paulo Research Foundation), and the German Science Foundation (Deutsche Forschungsgemeinschaft, DFG) within the DFG Priority Program (SPP 1294) "Atmospheric and Earth System Research with the Research Aircraft HALO (High Altitude and Long Range Research Aircraft)". This study was also supported by EU Project HAIC under FP7-AAT-2012-3.5.1-1 and by the German Science Foundation within DFG SPP 1294 HALO by contract no VO1504/4-1 and contract no JU 3059/1-1. For the operation of the ATTO site, we acknowledge the support by the Max Planck Society, the German Federal Ministry of Education and Research (BMBF contracts 01LB1001A, 01LK1602A and 01LK1602B) and the Brazilian Ministério da Ciência, Tecnologia e Inovação (MCTI/FINEP contract 01.11.01248.00) as well as the Amazon State University (UEA), FAPEAM, LBA/INPA and SDS/CEUC/RDS-Uatumã. This paper contains results of research conducted under the Technical/Scientific Cooperation Agreement between the National Institute for Amazonian Research, the State University of Amazonas, and the Max-Planck-Gesellschaft e.V.; the opinions expressed are the entire responsibility of the authors and not of the participating institutions. Special thanks for all the people involved in ATTO project, in particular Reiner Ditz, Jürgen Kesselmeier, Andrew Crozier, Thomas Disper, Alcides Camargo Ribeiro, Hermes Braga Xavier, Nagib Alberto de Castro Souza, Adir Vasconcelos Brandão, Amauri Rodriguês Perreira, Antonio Huxley Melo Nascimento, Thiago de Lima Xavier, Josué Ferreira de Souza, Roberta Pereira de Souza, Bruno Takeshi and Wallace Rabelo Costa. Remote sensing analyses and visualizations used in this study were produced with the Giovanni online data system, developed and maintained by the NASA GES DISC. We acknowledge the National Oceanic and Atmospheric Administration (NOAA) Air Resources Laboratory (ARL) for the HYSPLIT transport and dispersion model. We thank Daniel Moran-Zuloaga, Maria Praß, Leslie Kremper, Tobias Könemann, Jan-David Förster, Björn Nillius, Stefan Wolff, Anywhere Tsokankunku, Oliver Lauer for their support and inspiring discussions.





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





**Appendix**

**Table A1.** List of frequently used symbols and acronyms.

| Description | Acronym | Symbol | Units |
|---|---|---|---|
| **Accumulation mode particle number concentration** | | $N_{acc}$ | cm⁻³ |
| **Aerosol Optical Depth** | AOD | | |
| **Aerosol Robotic Network** | AERONET | | |
| **Aerosol, Cloud, Precipitation, And Radiation Interactions And Dynamics Of Convective Cloud Systems Campaign** | ACRIDICON-CHUVA | | |
| **Amazon Tall Tower Observatory** | ATTO | | |
| **Atmospheric Infrared Sounder** | AIRS | | |
| **Avalanche Photo-Diode** | APD | | |
| **Backward Trajectory** | BT | | |
| **Basic HALO Measurement And Sensor System** | BAHAMAS | | |
| **Biomass Burning** | BB | | |
| **Black Carbon** | BC | | |
| **Clean Layer** | CL | | |
| **Cloud Condensation Nuclei** | CCN | | |
| **Cloud-Aerosol Lidar And Infrared Pathfinder Satellite Observations** | CALIPSO | | |
| **Condensation Number Concentration (> 20nm)** | | $N_{CN,20}$ | cm⁻³ |
| **Condensation Particle Counters** | CPC | | |
| **Continental Boundary Layer** | CBL | | |
| **Coordinated Universal Time** | UTC | | |
| **Dimethyl sulfide** | DMS | | |
| **Fire Radiative Power** | FRP | | W m⁻² |
| **Goddard Earth Observing Model Version 5** | GEOS-5 | | |
| **HALO Aerosol Submicrometer Inlet** | HASI | | |
| **High Altitude And Long Range Research Aircraft** | HALO | | |
| **Inter-Quartile Range** | IQR | | |
| **Intertropical Convergence Zone** | ITCZ | | |
| **Long-Range Transport** | LRT | | |
| **Low Level Jet** | LLJ | | |
| **Lower Pollution Layer** | LPL | | |
| **Marine Boundary Layer** | MBL | | |
| **Mass Absorption Aerosol Photometer** | MAAP | | |
| **Mass Absorption Cross Section** | MAC | | |
| **Mass-Equivalent Diameter** | | $D_{MEV}$ | µm |



| | | | |
|---|---|---|---|
| **Moderate Resolution Imaging Spectroradiometer** | MODIS | | |
| **Optical Diameter** | | $D_o$ | µm |
| **Organic Aerosol** | OA | | |
| **Potential Temperature** | | $\theta$ | K |
| **Refractory Black Carbon** | rBC | | |
| **Region Of Interest** | ROI | | |
| **Single Particle Soot Photometer** | SP2 | | |
| **Single Scattering Albedo** | SSA | | |
| **Southern African Easterly Jet** | AEJ-S | | |
| **Ultra High Sensitivity Aerosol Spectrometer** | UHSAS | | |
| **Upper Pollution Layer** | UPL | | |
| **Upper Troposphere** | UT | | |
| **Water Vapor Mass Mixing Ratio** | | $q$ | g kg$^{-1}$ |



**Table 1.** Characteristic aerosol and trace gas concentrations in the upper pollution layer (UPL), clean layer (CL), lower pollution layer (LPL) and marine boundary layer (MBL) observed during the flight section off the Brazilian coast (16:50 to 19:07 UTC; Fig. 1): arithmetic mean ± standard deviation (std), median and interquartile range (IQR).

| | UPL | | | | | CL | | | | | LPL | | | | | MBL | | | | |
|---|---|---|---|---|---|---|---|---|---|---|---|---|---|---|---|---|---|---|---|---|
| | mean | std | P50 | P25 | P75 | mean | std | P50 | P25 | P75 | mean | std | P50 | P25 | P75 | mean | std | P50 | P25 | P75 |
| $N_{CN,20}$ (cm$^{-3}$) | 970 | 260 | 940 | 780 | 1200 | 500 | 60 | 490 | 470 | 520 | 1300 | 200 | 1220 | 1140 | 1420 | 420 | 140 | 420 | 300 | 490 |
| $N_{acc}$ (cm$^{-3}$) | 850 | 330 | 800 | 580 | 1100 | 180 | 55 | 170 | 140 | 200 | 650 | 140 | 660 | 540 | 750 | 230 | 50 | 220 | 190 | 270 |
| $N_{CCN0.5}$ (cm$^{-3}$) | 560 | 180 | 540 | 430 | 700 | 230 | 40 | 230 | 200 | 260 | 510 | 90 | 500 | 430 | 580 | 95 | 30 | 94 | 76 | 116 |
| $N_{rBC}$ (cm$^{-3}$) | 280 | 110 | 270 | 180 | 370 | 30 | 12 | 27 | 20 | 35 | 110 | 20 | 105 | 94 | 119 | 50 | 16 | 49 | 35 | 62 |
| $M_{rBC}$ (µg m$^{-3}$) | 1.0 | 0.4 | 0.9 | 0.6 | 1.2 | 0.09 | 0.04 | 0.08 | 0.06 | 0.12 | 0.36 | 0.11 | 0.35 | 0.29 | 0.42 | 0.18 | 0.07 | 0.17 | 0.12 | 0.22 |
| $f_{fine}$ (%) | 15 | 14 | | | | 65 | 7 | | | | 48 | 11 | | | | 43 | 13 | | | |
| $f_{vol}$ (%) | 16 | 9 | | | | 43 | 6 | | | | 27 | 7 | | | | 39 | 9 | | | |
| $f_{rBC}$ (%)* | 40 | 1 | | | | 6 | 2 | | | | 9 | 2 | | | | 12 | 3 | | | |
| $f_{CCN}$ (%) | 60 | 6 | | | | 46 | 6 | | | | 41 | 8 | | | | 23 | 6 | | | |
| $c_{CO}$ (ppb) | 150 | 30 | 140 | 120 | 170 | 83 | 4 | 83 | 80 | 85 | 105 | 5 | 103 | 100 | 107 | 92 | 1 | 92 | 92 | 93 |
| $c_{O3}$ (ppb) | 56 | 9 | 57 | 49 | 63 | 48 | 2 | 48 | 47 | 49 | 45 | 2 | 45 | 43 | 46 | 21 | 1 | 20 | 20 | 21 |
| $c_{NOy}$ (ppb) | 2.5 | 0.8 | 2.4 | 2.0 | 3.0 | 0.9 | 0.4 | 0.8 | 0.7 | 0.9 | 2.1 | 0.8 | 2.0 | 1.1 | 3.0 | 1.7 | 0.1 | 1.8 | 1.6 | 1.8 |
| $c_{NO}$ (ppb) | 0.10 | 0.02 | 0.10 | 0.09 | 0.11 | 0.07 | 0.01 | 0.07 | 0.05 | 0.07 | 0.10 | 0.02 | 0.09 | 0.09 | 0.13 | 0.12 | 0.02 | 0.12 | 0.11 | 0.12 |

*Calculated by applying a bivariate fit to ΔrBC and ΔCO measurements.



**Table 2.** Characteristic aerosol and trace gas concentrations during the African vs. South American dominated periods of the BB season at the ATTO site: arithmetic mean ± std, median and IQR of daily averages from 2013-2018.

| | African dominated BB** | | | | | South American dominated BB*** | | | | |
|---|---|---|---|---|---|---|---|---|---|---|
| | mean | std | median | 25th perc | 75th perc | mean | std | median | 25th perc | 75th perc |
| $N_{CN,20}$ (cm$^{-3}$) | 1350 | 550 | 1300 | 900 | 1700 | 2000 | 1000 | 1800 | 1400 | 2300 |
| $N_{CCN0.5}$ (cm$^{-3}$) | 1100 | 500 | 1090 | 750 | 1400 | 1800 | 900 | 1600 | 1200 | 2000 |
| $M_{rBC}$ (µg m$^{-3}$) | 0.36 | 0.12 | 0.33 | 0.26 | 0.42 | 0.41 | 0.17 | 0.36 | 0.29 | 0.48 |
| $c_{CO}$ (ppb) | 137 | 27 | 131 | 120 | 150 | 190 | 70 | 170 | 150 | 200 |
| $f_{CCN}$ (%) | 83 | 6 | 84 | 80 | 87 | 87 | 04 | 88 | 84 | 90 |
| $EnR_{BC}$* (µg m$^{-3}$ ppb$^{-1}$) | 7.4 | 3.1 | 7.3 | 5.6 | 8.9 | 4.4 | 2.0 | 4.1 | 3.1 | 5.6 |
| SSA | 0.85 | 0.02 | 0.85 | 0.84 | 0.86 | 0.90 | 0.03 | 0.90 | 0.89 | 0.93 |

*Calculated in a daily basis by applying a bivariate fit to 30-min ΔrBC and ΔCO measurements.

** for calculating the African BB dominated state, possible influences of South American fires ($CF_{BT}$ < 50th percentile of $CF_{BT,dry}$) as well as periods with clean atmospheric conditions ($M_{rBC}$ < 50th percentile of $M_{rBC,dry}$) were excluded.

***for calculating the South American BB dominated state, we selected only cases at which ($CF_{BT}$ > 50th percentile of $CF_{BT,dry}$) as well as periods of polluted atmospheric conditions ($M_{rBC}$ > 50th percentile of $M_{rBC,dry}$).



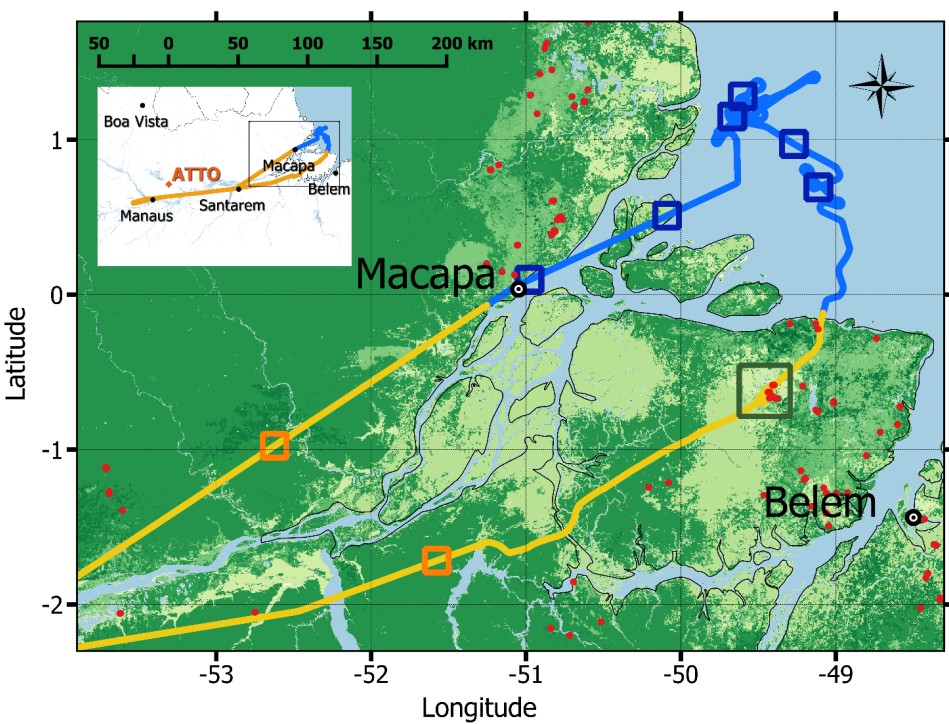

**Figure 1.** ACRIDICON-CHUVA flight AC19 on 30 September 2014. The squares represent the locations at which the aircraft ascended or descended through the upper pollution layer (UPL) (blue: offshore profiles, orange: inshore profiles). The yellow and blue segments of the flight track correspond to the in- and offshore sections that were averaged to obtain the profiles in Fig. 3 and 4. Red markers indicate fire spots on 30 September 2014 as obtained from INPE (http://www.inpe.br/queimadas/bdqueimadas/, last access on 17 April 2019), and the dark green square represents the location where fresh BB plumes were probed at ~1 km altitude.





**(a)**

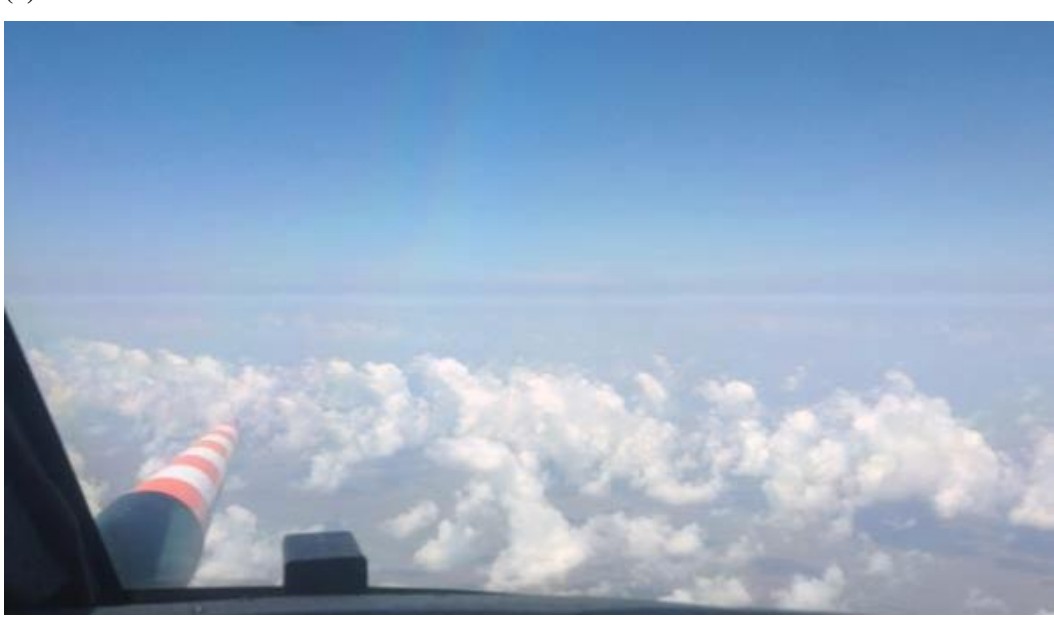

**(b)**

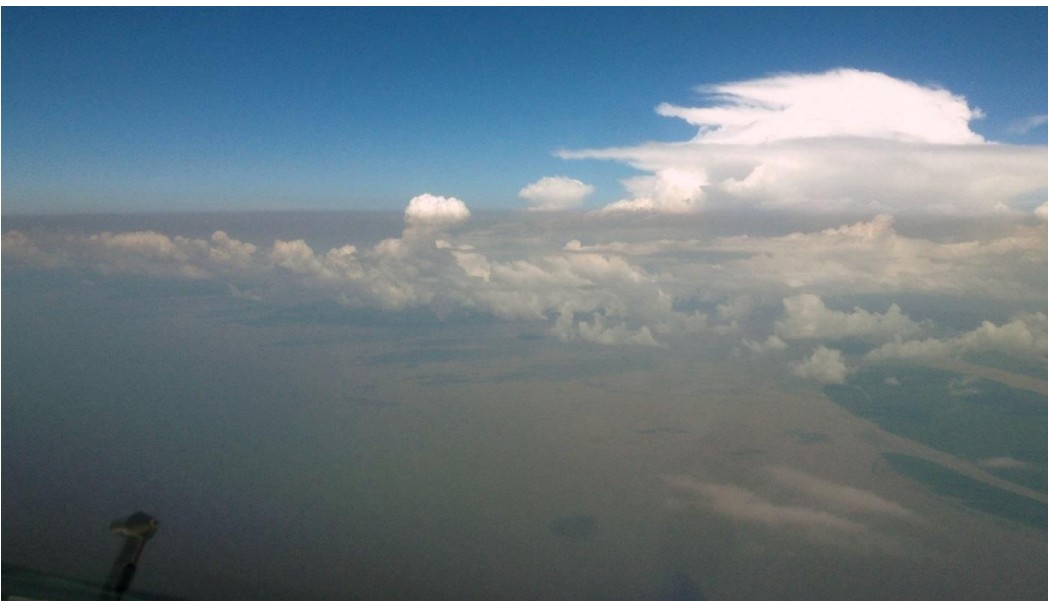

**Figure 2.** View from the HALO cockpit during flight AC19 on 30 September 2014, showing **(a)** the layering of the troposphere with clearly visible pollution layers as well as a clean layer in between at an offshore location (17:09 UTC) and **(b)** the brownish pollution layer arriving at the Brazilian coastline (16:55 UTC).



**Figure 3.** Vertical profiles of selected meteorological, aerosol and trace gas parameters measured off the Brazilian coast during flight AC19: (**a**) potential temperature, $\theta$, and water vapor mass mixing ratio, $q$; (**b**) total aerosol particle number concentration ($D_p > 20$ nm), $N_{CN,20}$, and ultrafine particle number fraction, $f_{fine}$; (**c**) accumulation mode particle number concentration, $N_{acc}$, and volatile particle number fraction, $f_{vol}$;





(**d**) CCN number concentration at $S = 0.5\%$, $N_{CCN0.5}$, and activated fraction at $S = 0.5\%$, $f_{CCN0.5}$; (**e**) rBC number concentration, $N_{rBC}$, and rBC number fraction, $f_{rBC}$; and (**f**) rBC mass concentration, $M_{rBC}$; (g) carbon monoxide, $c_{CO}$; (h) ozone, $c_{O3}$; (**i**) total reactive nitrogen, $c_{NOy}$; and (**j**) nitrogen monoxide, $c_{NO}$, mole fractions measured off the Brazilian coast during flight AC19. The black lines and shadings represent the median and IQR calculated for 150 m altitude bins during the flight section off the Brazilian coast (16:50 to 19:07 UTC, blue line in Fig. 1). The brown shaded area represents the approximate vertical location of the upper pollution layer (UPL); the altitudes of the lower pollution layer (LPL), the clean layer (CL), and the marine boundary layer (MBL) are indicated on the right side of the plot.





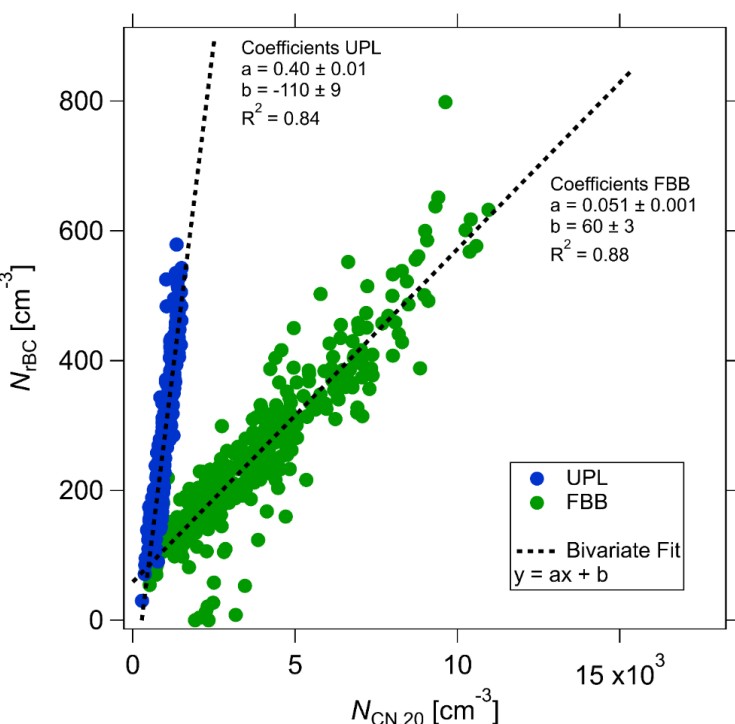

**Figure 4.** Correlation between total aerosol ($N_{CN,20}$) and black carbon ($N_{rBC}$) particle number concentrations in the UPL and in the fresh biomass burning (FBB) plume. The dotted lines are bivariate linear regressions with a slope corresponding to the average rBC number fraction, $f_{rBC}$.



**Figure 5.** Composite maps combining backward trajectories (BT) and satellite data products characterizing atmospheric conditions (**a₁**) during the flight AC19 on 30 September 2014 in comparison to (**b**) the averages of September observations during multiple years. Panel (**a₁**) shows HYSPLIT 10-days BT starting at different altitudes (500, 2500, 3500, 5000 m a.g.l.) at 18:00 UTC on 30 September 2014 (similar time and location as the UPL observations during flight AC19). Note that the altitudes where the BTs were initiated include the heights of the sampled UPL and LPL. The Fire Radiative Power (FRP) density (mW m$^{-2}$), retrieved by the Global Fire Assimilation System (GFAS v1.0) averaged from 15 to 20 of September 2014 is also shown with 0.1° x 0.1° grid resolution. The orbits of two CALIPSO passages on 30 and 16 September 2014 (see





Fig. 8) as well as the geographic location of the ATTO site and Ascension Island are also illustrated. Panel (**a₂**) shows multiple clearly visible fire plumes in the African sources region. Panel (**b**) shows multiyear averages of all Septembers for: (i) HYSPLIT BT ensembles starting at ATTO (1000 and 4000 m a.g.l.) from 2005 to 2018. Contour lines represent the fraction of occurrence of overpassing trajectories in a specific region as described in Pöhlker et al. (2019). (ii) AIRS-derived CO data products (400 to 600 hPa atmospheric levels) from 2005 to 2018 and (iii) TRMM precipitation from 2005 to 2018.

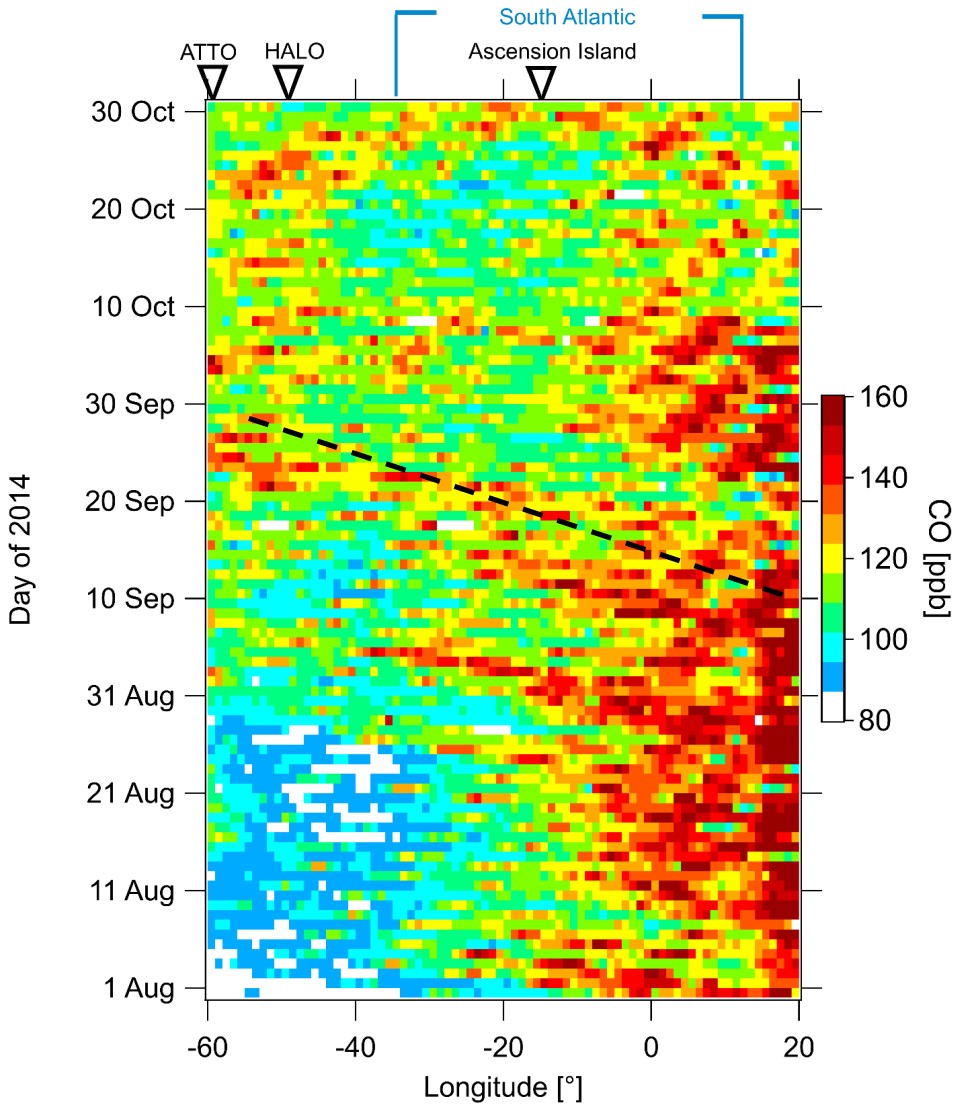

**Figure 6.** Hovmöller plot of the daily AIRS-derived carbon monoxide (400 to 600 hPa) distributed over the South Atlantic region (60 °W to 20 °E) from August to October 2014, averaged over the latitudinal band of 10 °S to 5 °N, corresponding to the region of interest (ROI) highlighted in Fig. 6a. Several events of transatlantic transport of aerosol from Africa towards South America can be identified. The black dashed line highlights a particularly strong plume originating around 10 September 2014 and arriving in the observational area of AC19 on 30 September.

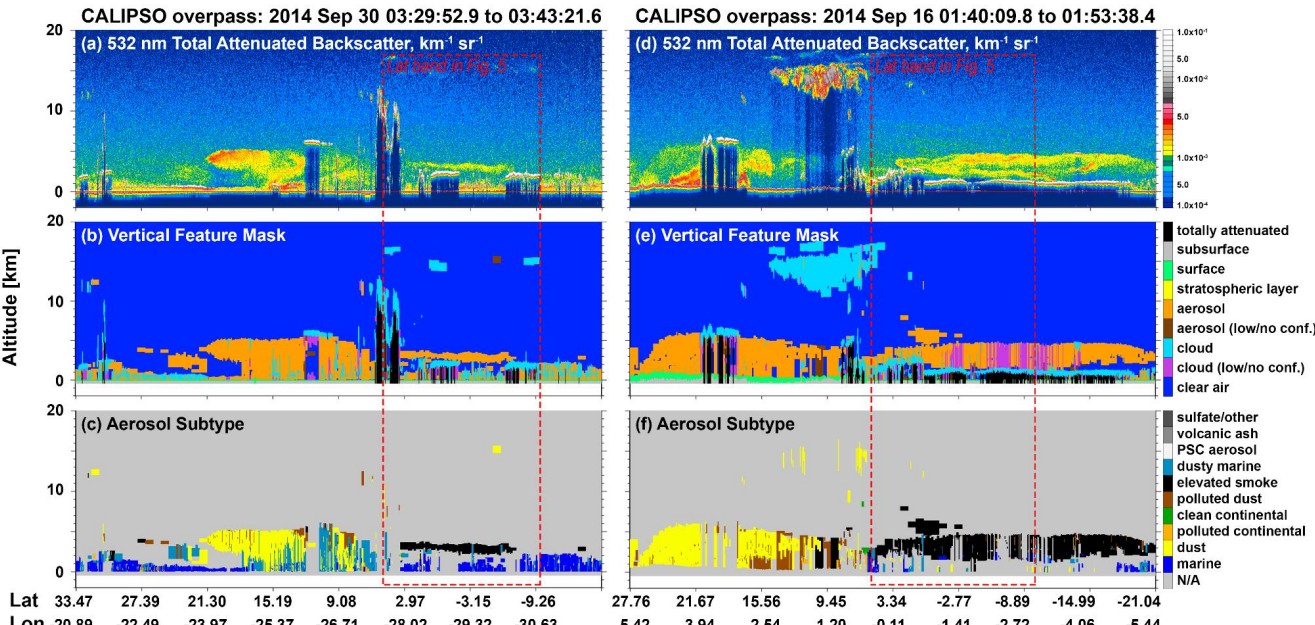

**Figure 7.** CALIPSO-derived lidar profiles for 16 and 30 September 2014, where African BB plumes were identified over the South Atlantic. The first profile near the South American coast shows the aerosol layer at similar altitudes as observed during flight AC19. Satellite orbits for both profiles are shown in Fig. 6a.



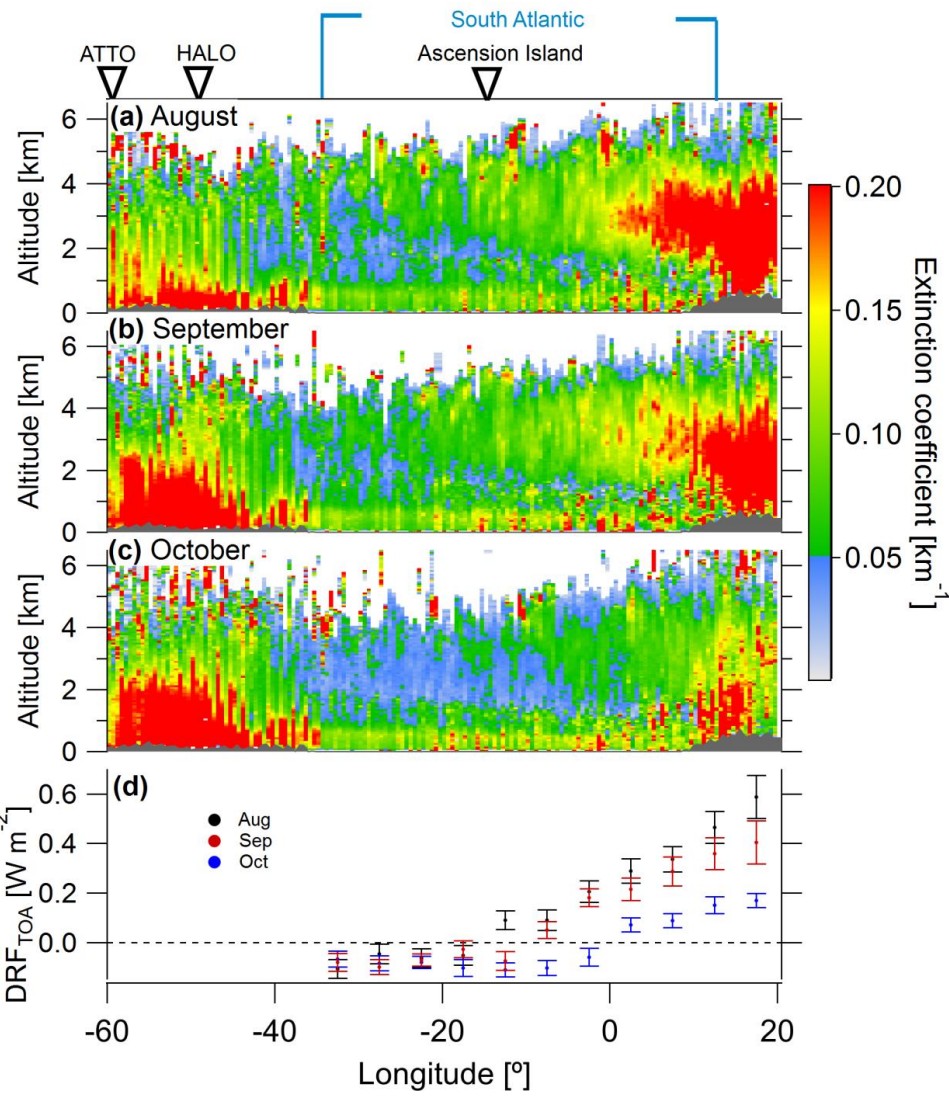

**Figure 8.** Curtain plot showing the columnar aerosol extinction coefficient at 532 nm, based on multiyear CALIOP data from 2012 to 2018 (only night time data). Panels represent monthly averages for the months of **(a)** August, **(b)** September, and **(c)** October within the latitude band from 10° S to 5° N, corresponding to the ROI indicated in Fig. 6a. The grey shaded area represents the mean surface elevation and depicts boundaries of the African and South American continents. Panel (d) shows the daily mean of the direct radiative forcing at the top of the atmosphere (DRF-TOA) exerted by the pollution layer over the South Atlantic Ocean, calculated using the LibRadTran radiative transfer model.



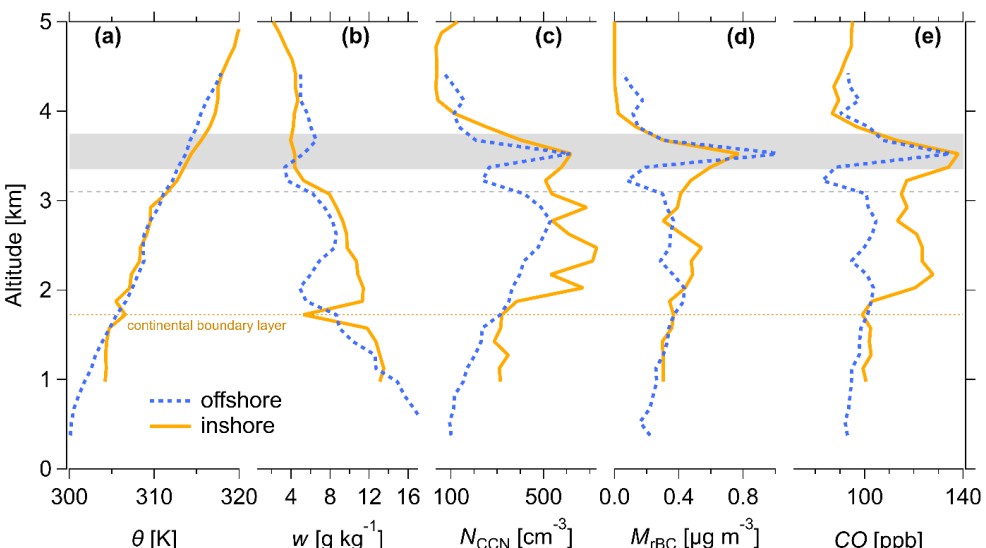

**Figure 9.** Vertical profiles of selected meteorological, aerosol and trace gas parameters measured inshore and offshore of the Brazilian coast during AC19: (**a**) potential temperature, $\theta$; (**b**) water vapor mass mixing ratio, $w$; (**c**) CCN ($S = 0.5$ %) number concentration, $N_{CCN0.5}$; (**d**) rBC mass concentration, $M_{rBC}$; and (**e**) carbon monoxide mole fraction, $c_{CO}$. The figure shows the medians calculated for 150-m altitude bins over the flight sections inshore (yellow) and offshore (blue) the Brazilian coast (as indicated in Fig.1). The grey shaded area represents the approximate vertical location of the upper pollution layer (UPL) and the grey dashed line, the lower limit of the clean layer (CL) observed exclusively during the offshore profiles.



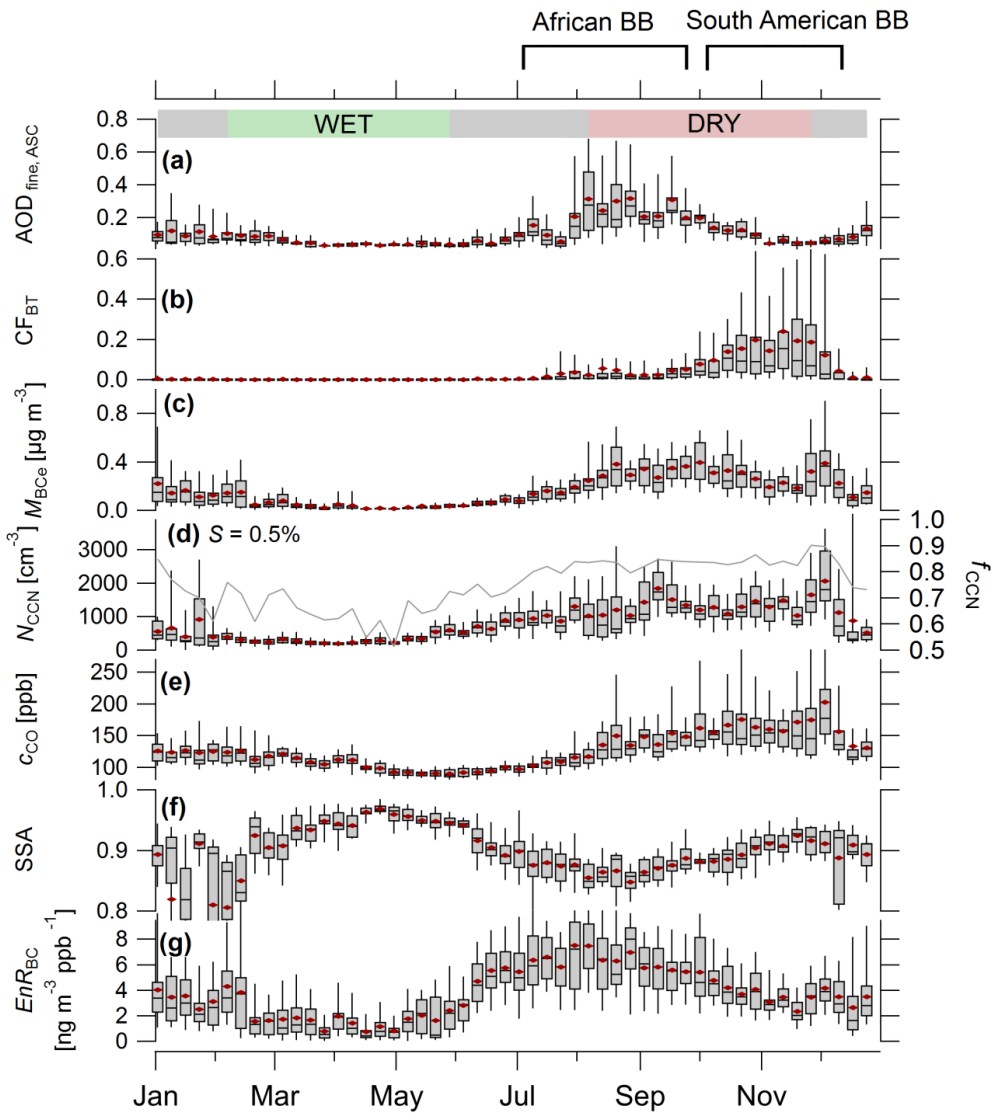

**Figure 10.** Seasonal variation of aerosol properties from long-term observations at Ascension Island and the ATTO site (2013-2018). From top to bottom, the plot shows multi-year weekly averages of **(a)** fine-mode aerosol optical depth ($AOD_{fine}$) at 500 nm retrieved by the AERONET sunphotometer at Ascension Island; **(b)** cumulative fire spots encountered along backward trajectories of air masses arriving at the ATTO site; **(c)** black carbon mass concentration $M_{BCe}$; **(d)** CO mole fraction; **(e)** single scattering albedo, SSA; and **(f)** BC enhancement ratio, $EnR_{BC}$. The boxplots present the mean (red markers), the median (segment) and the 9th and 91th percentiles (lower and upper box edges, respectively) of the long-term daily measurements. The green and red shaded areas represent the wet and dry season, respectively, as defined in Andreae et al. (2015). On the top of the figure, it is also indicated the periods of the year when the ATTO site is dominated by LRT African BB and the South American BB, respectively.