# Peer review of "Influx of African biomass burning aerosol during the Amazonian dry season"

_Atmospheric Chemistry and Physics, 2019_

## Referee Comment (RC1) · Anonymous Referee #1 · 21 Oct 2019

Manuscript: Influx of African biomass burning aerosol during the Amazonian dry season through layered transatlantic transport of black carbon-rich smoke (Holanda et al.,)

Referee Review of manuscript

The current manuscript discusses observations during a single research flight of two biomass burning layers off- and on-shore of Brazil, where the upper layer was characterized by very high black carbon (BC) number concentrations while the lower layer by a much lower BC concentration. Using back trajectory calculations, the high BC layer was identified as originating from the African coast. Using this finding, the authors examined the long term data collected at the ATTO site to identify that the early

part of the Amazonian dry season tends to be influenced by the long-range transport of the African biomass burning emissions while the late dry season is dominated by local burning. One of the core findings of this paper is that the SSA (at 637 nm) during the early portion of the dry season (i.e., influenced by the African plume) is nominally 0.85 while local BB emissions are found to be nominally 0.9. This finding suggests that optical and microphysical changes with the African BB plume occur during transport. The information and analysis contained in this manuscript is certainly worthy of publication, but, as discussed below, additional analysis that could be done with available data needs to be done to help explain underscore these findings. Therefore, it is recommended that the manuscript be revised and resubmitted.

I read this manuscript with great excitement as long-range transport of biomass burning emission is a subject area that has not received much study. However, upon reading this manuscript I felt cheated. Combining the aircraft data with the long-term ATTO dataset, the authors make a very compelling argument that Amazonia is indeed influenced by African fires, but do present any further data analysis that might provide the community with a deeper understanding of what processes might be present. In the abstract, the authors correctly point out that the "microphysical properties, spatiotemporal distribution and long-range transport" BC aerosols are "not well constrained". But as highlighted below, the authors seemingly have the data to help probe this, but did not analyze that data and, in turn, are missing the opportunity to further strengthen their manuscript. A quick perusal of the Wendisch et al., BAMS article reveals that the HALO payload included the two SP2s, one three-wavelength PSAP, a single wavelength PSAP, UHSAS, and a C-ToF-AMS. (This reviewer is quite surprised to learn that a nephelometer was not part of the HALO payload.) These available datasets open up additional analysis that would greatly strengthen and help elucidate properties and processes. For example, the authors could very easily combine the PSAP absorption coefficients with the SP2 to derive the mass absorption cross-sections for both layers. This would inform us about how different the optical properties are between the plumes. Furthermore, using the SP2 the authors could probe the black carbon mixing

state (e.g., coating thickness). This reviewer suspects that this analysis will reveal that the coating thickness for the African plume will be thinner than that derived for the local BB plume - which would further explain why the ATTO-site derived SSA is 0.85 for the African plume and 0.90 for the local BB plumes. Also, sticking with the SP2, why not examine the BC size distributions? What do the differences - should be they exist - tell us about the two plumes? Continuing with this theme, the authors have the UHSAS which could be used with the SP2 to examine (and compare) the two size distributions. As referenced above, the authors cite the paucity of BC microphysical properties data, yet seemingly do not take advantage of the readily available data.

Sticking with the theme of missing an opportunity, the authors chose to center their discussion on number concentration: higher number fraction of BC particles in the African layer versus the measured number fraction in the local layer. The availability of the C-ToF-AMS, along with the SP2, enables the authors to derive a more meaningful mass fraction of BC in both layers. Additionally, the AMS provides a measurement datastream that can provide aerosol composition, yet is not taken advantage of. (It is interesting to note that the authors make a passing reference to the AMS on Page 13, line 424 where they cite the sulfate content concentration.) A plume advecting for 10 days from the African shore to the South American shore should exhibit differences with that observed for a localized ("fresh") plume. It is because of the available datasets outlined in this paragraph and the one before it, that this reviewer felt cheated in that so much more could be derived from the aircraft dataset and thus provide a much more complete story. It is recommended that the authors mine their data for more nuggets of gold that are surly there.

Other specific issues:

Page 2, line 46. The authors state that the BC particle number fraction is $\sim$ 40%. I think this is wrong and that the value is closer to 30%. Using number concentrations reported on page 12 (lines 371, 372, and 383, N_CN,20 = 970, N_acc = 850, and N_rBC = 280 particles/cc) the fractions I derive are 0.29 (using N_CN,20) and 0.33 (using N_acc).

As a rough check, derived values from figure 4 support the 30% fraction vs 40%.

Page 3, Line 73 -75. The authors are reminded that semi-transparent coating also alter BC optical properties.

Page 5, line 138-139. The authors refer to the African latter as pollution, yet their discussion centers on biomass burning. Pollution tends to suggest an anthropogenic contribution. It is suggested that the authors that consider changes "pollution" to biomass burning or something that describes the layer as containing BB emissions.

Page 7, 209- 211. The authors might want to examine the paper by Adachi et al., (AST, 52, 46-56, 2018) where these authors present data from a mid-latitude wildfire that exhibit thermal stability well above the 250 C employed in the TD. How would increased thermal stability alter conclusions derived in the present analysis?

Page 8, line 250-252. In deriving the EnRbc ratio, is the CO background corrected? I assume so, but this should be explicitly stated.

Page 11, line 337. Please be consistent with respect to referencing black carbon. Within the SP2 community, the "black carbon" that is detected by this instrument is referred to as rBC (refractory black carbon). Indeed, in many places the authors refer to "rBC". To help minimize confusion, please be consistent and use only one term.

Page 12, 389- 390. The authors write "The high frBC further agrees with the pronounced brownish color of the visually observable layer in Fig. 2." This is a very misleading sentence. The brownish color could easily be due to brown carbon (BrC). Which, given the fact that the HALO payload included a three-wavelength PSAP, could be readily calculated via the Angstrom absorption exponent (AAE).

Page 21, line 678-681. As highlighted above, the HALO 3-wavelength PSAP would provide insight into the presence of BrC.

Page 37, Table 2. Please explicitly cite the wavelength used to derive the SSA (637 nm).

---

## Referee Comment (RC2) · Anonymous Referee #2 · 1 Nov 2019

General:

This paper describes the influx of African smoke to the Amazon during the dry season and discusses the implications of this transport for the Amazon. This work is novel and of interest to the aerosol community. I do recommend this paper for publication; however, I suggest that the authors revise the current manuscript, which is too long and detailed in its current form.

Major Comments:

1. In its current form, this paper is well-written but has so many details, that I found myself losing the main point of a paragraph or an entire section. I also found that a lot

of the major findings were either buried in the middle of a section or at the end of a very long discussion that could have been cut in half. I list below a few suggested areas to tighten up this paper but encourage the authors to re-evaluate their paper as a whole and determine areas that could go into the supporting information.

2. This paper has too many acronyms to keep track of. I found myself having to go back to the acronym table a lot, which made the paper difficult to read. I suggest that the authors try to reduce the number of terms and instruments discussed to focus only on the most important ones for this story and to place the rest of the measurements in the supporting information.

3. I felt that the abstract and introduction promise that the focus of the paper will be on radiative impacts but those impacts weren't as well-emphasized in the results. I suggest restructuring the conclusions section to include a brief overview of the findings then focus on the radiative impacts as an extension of their findings.

Specific Comments:

Abstract:

1. Lines 36-40: This sentence is far too long and dense. I suggest breaking it up.

2. The abstract should include a very brief clause or sentence describing the most relevant instruments used.

3. Lines 46-47: I had a hard time understanding this part of your sentence.

Introduction:

1. I suggest condensing the first two paragraphs down to half a paragraph. There is a lot of detail that is not necessary.

2. I also suggest adding a sentence or two regarding the impact of aging on the direct effect. It has been shown in several studies that coatings can greatly alter the radiative properties of black carbon [Moffet and Prather, 2009].

3. Lines 114-118 can be cut as they do not add much to the introduction.

4. The sentence on lines 121-122 is not clear to me.

Methods:

1. I strongly suggest reducing the text in this section. For example, the gas phase data does not play a major role compared to the aerosol data. I suggest mentioning what was measured in the gas phase then directing the reader to the SI for more details on the instrumentation. This will also reduce the number of acronyms that the reader must memorize.

2. I also suggest placing details about AIRS data and AERONET in the SI. The AIRS and AERONET data were not as integral to the study as other methods.

Results:

1. I suggest placing Figure 2 in the SI.

2. Sections 3 and 3.1 are too long and detailed. This information dilutes the major findings regarding the transport of biomass burning and its impact on the optical and cloud nucleating properties of Amazonian aerosol. I suggest condensing the material from 330-394 down to a paragraph if possible and placing much of this information in the SI.

3. Lines 442-443: It is not necessary to explain that O3 is a secondary pollutant produced photochemically in BB plumes.

4. Lines 447-467: This paragraph could go in a discussion section.

5. Lines 470-473 belong in the introduction.

6. The authors should note that the methodology shown in Figure 7 is similar to the methods used in [Barkley et al., 2019] to identify similar plumes.

7. Lines 591-606 are really important for understanding the implications of your work. I

strongly suggest either giving this paragraph its own section or placing it in a discussion section.

References:

Barkley, A. E., J. M. Prospero, N. Mahowald, D. S. Hamilton, N. Mahowald, K. J. Popendorf, A. M. Oehlert, A. Pourmand, A. Gatineau, K. Panechou-Pulcherie, P. Blackwelder, and C. J. Gaston (2019), African biomass burning is a substantial source of phosphorus deposition to the Amazon, Tropical Atlantic Ocean, and Southern Ocean, Proceedings of the National Academy of Sciences of the United States of America, DOI: 10.1073/pnas.1906091116.

Moffet, R. C., and K. A. Prather (2009), In-situ measurements of the mixing state and optical properties of soot with implications for radiative forcing estimates, Proceedings of the National Academy of Sciences of the United States of America, 106(29), 11872-11877.

---

## Author Response (AR1)

**Response format description:**

Black text shows the original referee comment, blue text shows the authors response, and red text shows
5   quoted manuscript text. Changes to the manuscript text are shown as *italicized and underlined*. We used
bracketed comment numbers for referee comments (e.g., [R2.1]) and author's responses (e.g., [A2.1]).
Line numbers refer to the discussion/review manuscript.

10  **Anonymous Referee #1**

Received: 21 October 2019

General comment:

The current manuscript discusses observations during a single research flight of two biomass burning
15  layers off- and on-shore of Brazil, where the upper layer was characterized by very high black carbon
(BC) number concentrations while the lower layer by a much lower BC concentration. Using back tra-
jectory calculations, the high BC layer was identified as originating from the African coast. Using this
finding, the authors examined the long term data collected at the ATTO site to identify that the early
part of the Amazonian dry season tends to be influenced by the long-range transport of the African bio-
20  mass burning emissions while the late dry season is dominated by local burning. One of the core find-
ings of this paper is that the SSA (at 637 nm) during the early portion of the dry season (i.e., influenced
by the African plume) is nominally 0.85 while local BB emissions are found to be nominally 0.9. This
finding suggests that optical and microphysical changes with the African BB plume occur during
transport. The information and analysis contained in this manuscript is certainly worthy of publication,
25  but, as discussed below, additional analysis that could be done with available data needs to be done to
help explain underscore these findings. Therefore, it is recommended that the manuscript be revised and
resubmitted.

Author response: We thank Referee #1 for the critical feedback and constructive suggestions. We ad-
30    dress the individual comments below.

[R1.1] I read this manuscript with great excitement as long-range transport of biomass burning emission is a subject area that has not received much study. However, upon reading this manuscript I felt cheated. Combining the aircraft data with the long-term ATTO dataset, the authors make a very compelling ar-
35    gument that Amazonia is indeed influenced by African fires, but do present any further data analysis that might provide the community with a deeper understanding of what processes might be present. In the abstract, the authors correctly point out that the "microphysical properties, spatiotemporal distribution and long-range transport" BC aerosols are "not well constrained". But as highlighted below, the authors seemingly have the data to help probe this, but did not analyze that data and, in turn, are missing
40    the opportunity to further strengthen their manuscript.

[A1.1] We understand your concern and considered it carefully. With this longer and more general response, we would like to address your comments on "missed opportunities" expressed in the referee comments R1.1 to R1.6. Subsequently, the comments R1.2 to R1.6 are further ad-
45    dressed individually and in more detail.

For a better understanding of the atmospheric and climatic relevance of BC, a detailed analysis of its spatiotemporal distribution in the atmosphere, on one hand, and its microphysical properties, on the other hand, is needed. Both aspects were probed during the ACRIDICON-CHUVA campaign, which yielded a large data set with manifold interesting findings (lots of "nuggets of
50    gold", quote from R1.6).

The problem we have dealt with quite intensively was how to portion the manifold BC-related results from the large ACRIDICON-CHUVA data set best into 'compact and coherent publishable units'. Let us clarify one key aspect here at the beginning already that may have contributed to the referee's concern(s): We did not 'forget' the BC microphysics, size distributions,
55    aerosol chemical information, etc. but rather decided to address all these results in another paper as a dedicated follow-up study (currently being prepared). In fact, in the first drafts of the present manuscript, we had an extended section on BC/aerosol microphysical properties included.

However, we noticed quite soon that the paper became way too long and hardly readable. In this context, Referee #2 seems actually very concerned that even the current form of this paper is too long (see R.2.1).

A few more thoughts on our rationale regarding 'compact and coherent publishable units': Our intention has been to split the results on BC and related aerosol aspects into several manuscripts, being (more or less) split along the large blocks of "BC spatiotemporal distribution in the atmosphere" and "BC microphysical properties". Evidently, the present study belongs to the first block and specifically deals with the layered transport of BB smoke from southern Africa to South America, which was observed during the only ACRIDICON fight (AC19) over the ocean. In order to systematically address the BC microphysical properties, also the other flights (probing several contrasting conditions) have to be included. So if we had addressed detailed BC/aerosol microphysics just within the scope of flight AC19, the study would be inherently incomplete, since some very interesting microphysical aspects would be missing that result from the comparison of contrasting flights and flight patterns. In contrast, if the microphysics from other flights (beyond AC19) would be included here, the manuscript would become way too long. So in the end, we decided to focus the present manuscript just on AC19 with one main focal point, which is the description of the layered transatlantic transport, its properties in terms of profiles and transport dynamics, as well as an estimate of its significance on the Amazon Basin in general.

As mentioned in the beginning, we considered the referee's comment(s) carefully. We came to the conclusion that certain aspects of the manuscript could/should indeed be changed and improved to account for the referee's concerns:

- We clarified several statements on the aim and scope of the study as outlined below. Hopefully this avoids any impression that the manuscript (i.e., the introduction) promises more than the results part actually covers.
- Second, we added several further results from the instruments UHSAS and C-ToF-AMS (as requested by the referee in R1.2 to R1.6) to discussed selected aspects of aerosol size distributions and chemical composition, which indeed helped to improve the quality of

the manuscript. We think that this addresses several of the referee's comments. For details, please refer to our responses to R1.2 to R1.6.

The following clarifications on statements have been included in the last paragraph of the introduction (P4-5, L130-144): This study focusses on the transatlantic transport of African BB smoke into the Amazon Basin by combining *in-situ* aircraft observations, modeling results, and remote sensing data. *The core of this work are aircraft observations made within a defined African pollution layer upon its arrival at the South American coast during the ACRIDICON-CHUVA campaign over Amazonia in September 2014 (Wendisch et al., 2016). We focus primarily on the spatiotemporal distribution and advection dynamics of the BB smoke layers by analyzing (i) aerosol and trace gas concentration profiles,* (ii) backward trajectories and African BB source regions, (iii) the seasonality of the pollution transport, (iv) the horizontal and vertical extent of the transported layers, and (v) the convective mixing and smoke entrainment from the layers into the planetary boundary layer as they are transported from the ocean into the South American continent. *Note that a detailed characterization of the microphysical aerosol properties within the BB smoke layers (e.g., the BC core diameters and mixing state) is beyond the scope of the present work and will be the subject of a separate and extended follow-up study (Holanda et al., in preparation). As a final step of the present study, we integrate its key results into the broader picture of the long-term aerosol observations at the central Amazonian ATTO site to estimate the relevance of African pollution for the aerosol life cycle in the dry season.*

[R1.2] A quick perusal of the Wendisch et al., BAMS article reveals that the HALO payload included the two SP2s, one three-wavelength PSAP, a single wavelength PSAP, UHSAS, and a C-ToF-AMS. (This reviewer is quite surprised to learn that a nephelometer was not part of the HALO payload.) These available datasets open up additional analysis that would greatly strengthen and help elucidate properties and processes. For example, the authors could very easily combine the PSAP absorption coefficients with the SP2 to derive the mass absorption cross-sections for both layers. This would inform us about how different the optical properties are between the plumes.

[A1.2] We agree with the referee. The focal point of this manuscript is to address the first observation of highly aged African BB plume over the Amazon basin using the SP2 instrument that measures the mass of individual rBC particles. Complementary aerosol measurements onboard HALO allowed us to estimate the contribution of black carbon particles in relation to other aerosol species in terms of number and mass concentrations. With respect to optical properties of the plume, the main reason why we did not include the PSAP (3-λ) data is because the atmospheric layers were detected primarily during ascents and descents of the aircraft. In such situations, the pressure fluctuations inside the inlet prevent reliable measurements with the PSAP because the instability of the filter leads to strong artefacts. Aside from that, integration times are too short to obtain good measurement statistics for the PSAP. The second PSAP instrument (1-λ) was operating behind the CVI, that means, measuring aerosol residuals, which is not addressed in this work. With respect to the UHSAS and C-ToF-AMS data sets, please refer to comments A1.4 and A1.5, respectively.

[R1.3] Furthermore, using the SP2 the authors could probe the black carbon mixing state (e.g., coating thickness). This reviewer suspects that this analysis will reveal that the coating thickness for the African plume will be thinner than that derived for the local BB plume - which would further explain why the ATTO-site derived SSA is 0.85 for the African plume and 0.90 for the local BB plumes. Also, sticking with the SP2, why not examine the BC size distributions? What do the differences - should be they exist – tell us about the two plumes?

[A1.3] That's true – the rBC size distributions and mixing state represent key properties for a detailed aerosol microphysical study. Please refer to our response A1.1, which specifies our rationale on why this is not included in the present manuscript.

[R1.4] Continuing with this theme, the authors have the UHSAS which could be used with the SP2 to examine (and compare) the two size distributions. As referenced above, the authors cite the paucity of BC microphysical properties data, yet seemingly do not take advantage of the readily available data.

[A1.4] As suggested by the referee, we included the aerosol number size distributions, based on UHSAS data for the different layers and BB plumes discussed, as a new figure in the main text (Figure 4). As expected, clear differences in the size distributions were observed between the two BB plumes/conditions: the aged African BB is characterized by a modal diameter ($d_o$) of 132 nm, while the fresh Amazonian BB show a smaller $d_o$ = 124 nm. The largest particle mean diameter was observed in the marine boundary layer (MBL) with $d_o$ 143 nm, and the smallest, in the clean layer (CL) and lower pollution layer (LPL) with $d_o$ = 90 nm and $d_o$ = 105 nm, respectively. The latter is consistent with the new particle formation in regions of cloud detrainment as discussed in the results section (L480 to L486). The UHSAS results are summarized in Table 2. Moreover, we found a good agreement between the SP2 and UHSAS measurements for $D_o$ > 200 nm, as shown in the individual particle number size distributions (PNSD) in Fig. S5. Below this threshold, the SP2 efficiency decreases significantly for the scattering signal.

[Figure]

*__Figure 4.__ Particle number size distributions (PNSD) measured by the UHSAS for UPL, CL, LPL and MBL, as defined in Fig. 3, and the fresh BB plume probed during AC19 (see Fig. 1). The data points (black dots) are fitted by lognormal functions between 90 and 500 nm (Heintzenberg, 1994).*

160

*__Table 2.__ Fit parameters of UHSAS size distributions (Fig. 4) for the different layers/plumes probed during AC19. A log-normal function (Heintzenberg, 1994) was used to fit a one-modal size distribution to the mean data points:* $\frac{dN}{d\ln d_p} = \frac{A}{\sqrt{2\pi}\ln\sigma_g} exp\left(-\frac{(\ln d_p - \ln d_0)^2}{2\ln(\sigma_g)^2}\right)$

|  | UPL | CL | LPL | MBL | BB |
|---|---|---|---|---|---|
| $A$ | 2920 | 970 | 2890 | 680 | 13930 |
| $d_0$ *(nm)* | 132 | 90 | 105 | 143 | 124 |
| $\sigma_g$ | 1.55 | 1.58 | 1.65 | 1.40 | 1.50 |
| $R^2$ | 1.00 | 0.99 | 1.00 | 1.00 | 1.00 |

[Figure]

165

*Figure S5. Particle number size distributions (median and interquartile range) derived from the UHSAS and SP2 (rBC + SC) for the (a) CL, (b) LPL, (c) MPL, (d) UPL and (e) fresh BB plume probed during flight AC19. Panel (f) shows the curve fits of the UHSAS data points.*

170    After including Fig.4 and Fig S5 in the present study, the following changes were necessary in the results section:

(L369-376): In the atmospheric column, $f_{vol}$ reaches its minimum of $16 \pm 9$ % within the UPL and generally shows a similar profile as $f_{fine}$, indicating a rather aged plume (Grieshop et al., 2009; Zhou et al., 2017). *The particle number size distributions of the UPL aerosol – in comparison to the LPL, CL, MBL, and fresh BB aerosols probed during AC19 – are shown in Fig. 4 and summarized in Table 2. A modal diameter of 132 nm was observed for the UPL aerosol, whereas the fresh BB aerosol showed a clearly smaller modal diameter of 124 nm. Further note that the modal diameter in the UPL is smaller than the 220 nm observed directly off the African coast (Weinzierl et al., 2006).*

(L446-447): One interesting aspect of the LPL is that the ultrafine fraction accounts for about half of the aerosol number concentration *($d_0$ = 105 nm, see PNSD in Fig. 4).*

(L454-456): In the MBL (with its top at ~600 m asl), the total and accumulation mode particle concentrations are somewhat lower than in the layers aloft ($N_{CN,20}$ = $420 \pm 160$ cm$^{-3}$ and $N_{acc}$ = $230 \pm 50$ cm$^{-3}$) *and present large diameters ($d_o$ = 143 nm)*.

(L465-467): We further found $N_{CN,20}$ = $500 \pm 60$ cm$^{-3}$, which is comparable to $N_{CN}$ = 500 cm$^{-3}$ in another CL as reported by Hobbs (2003). *Within the CL, the aerosol size distribution is substantially shifted towards the Aitken mode ($d_0$ = 90 nm, Fig. 4).*

NOTE: By including the UHSAS data into the analysis, we have to use a new filter for defining the fresh BB plume in order to get a more characteristic and well defined size distribution. Therefore, in order to keep the "BB" definition consistent in all the analyses, data points in Figs. 5 and S7 have also been updated.

[R1.5] Sticking with the theme of missing an opportunity, the authors chose to center their discussion on number concentration: higher number fraction of BC particles in the African layer versus the measured number fraction in the local layer. The availability of the C-ToF-AMS, along with the SP2, enables the authors to derive a more meaningful mass fraction of BC in both layers. Additionally, the AMS provides a measurement datastream that can provide aerosol composition, yet is not taken advantage of. (It is interesting to note that the authors make a passing reference to the AMS on Page 13, line 424 where they cite the sulfate content concentration.) A plume advecting for 10 days from the African shore to the South American shore should exhibit differences with that observed for a localized ("fresh") plume.

205

[A1.5] We agree (see also A1.1). Aerosol chemical components internally or externally mixed with the rBC cores are an important factor influencing the particles' hygroscopicity and radiative effects. As requested, we included composite plots with the rBC mass fractions (with respect to the total mass detected by the SP2+AMS) for each of the BB plumes as a new figure in the main

210
text (Fig. 6). We also included the LPL, CL, and MBL compositions for comparison. Note that the different layers could only be sampled with the C-ToF-AMS during the inshore intercepts. During the offshore section of AC19, the C-ToF-AMS was measuring aerosol residuals through the CVI inlet and could not be used in our analysis. Therefore, the statistics related to the C-ToF-AMS data are reduced due to the lower time resolution of the instrument (30 seconds) and to the

215
use of only-inshore profiles. Despite the limited statistics, we found clear differences in the aerosol mass fractions between the different air mass. These new results were included in the manuscript as outlined below.

[Figure]

**Figure 6.** *Cumulative mass concentrations of non-refractory submicrometer species (i.e., organic (Org), sulfate (SO$_4^{2-}$), nitrate (NO$_3^-$), ammonium (NH$_4^+$)) and rBC **(top)**; and mass fractions of the respective species to the total mass (M$_{total}$ = M$_{Org}$ + M$_{SO4}$ + M$_{NO3}$ + M$_{NH4}$ + M$_{rBC}$) in the UPL, CL, LPL, and MBL, as defined in Fig. 3, and the fresh BB probed during AC19 (see Fig. 1) **(bottom)**. Note that no C-ToF-AMS data were available from 17:27 to 19:05 UTC during the offshore section of the flight AC19 and, therefore, a reduced number of measurements points is included in the averages. The concentration of organics was below the detection limit in the MBL.*

230 Further, we analyzed the photochemical aging of the organic material, which is presented in the supplementary material (Figure S6).

[Figure]

**Figure S6.** *Scatterplot of the ratios $f_{43}$ (m/z 43 to total organic signal) against $f_{44}$ (m/z 44 to total organic signal) expressing the photochemical aging of the organic aerosol measured by the C-ToF-AMS. The blue and green markers correspond to measurements within the UPL and fresh*

235 *BB, respectively. The signal at m/z 44 relates mostly to $CO_2^+$ ions and the m/z 43 signal to $C_2H_3O^+$ ions. The triangular region (dashed lines) in the $f_{44}$ vs. $f_{43}$ space defines the boundaries within which most of the organic aerosol was found in previous studies and can be used as a guide to characterize oxidized organic components: data in the upper left represent more oxidized organics vs. the less oxidized organics in the lower right (Ng et al., 2010; Schulz et al.,*

240 *2018).*

By including the C-ToF-AMS data in the present study, the following changes were necessary:

1. We include a brief paragraph in the methodology describing the instrumentation used (L213-217): *A compact time-of-flight aerosol mass spectrometer (C-ToF-AMS, Aerodyne Research, Inc., Billerica, MA, USA) measured the mass concentration of four chemical species (i.e., organics, sulfate, nitrate, and ammonium) of the submicrometer aerosol with a time resolution of 30 seconds (Drewnick et al., 2005; Schulz et al., 2018). A complete description of the instrument and its operation during the ACRIDICON-CHUVA campaign is given in Schulz et al. (2018) and Andreae et al. (2018).*

2. In the results section, we include the following paragraph (L389-410): *In terms of absolute mass concentrations, rBC within the UPL, with $M_{rBC} = 1.0 \pm 0.4$ μg m$^{-3}$ (ranging from 0.5 to 2 μg m$^{-3}$), approaches the highest BC levels observed at ATTO ($M_{BCe}$ up to 2.5 μg m$^{-3}$; Pöhlker et al., 2018; Saturno et al., 2018b). Figure 6 shows the fractions of rBC mass relative to the other main constituents of the submicrometer aerosol ($M_{total}$ = non-refractory + rBC) in the UPL in comparison to the CL, LPL, MBL, and fresh BB values. Organic matter – comprising co-emitted primary as well as secondarily formed organics– accounts for the dominant mass fractions in all layers, with $f_{org,M} \approx 50$ % in the UPL, CL and, LPL, and as much as 73 % in the fresh BB plume. Generally, the dominance of organic matter is in agreement with previous studies performed at different locations and seasons in the Amazon region (e.g. Brito et al., 2014; Chen et al., 2015; Fuzzi et al., 2005; Martin et al., 2010, 2017; de Sá et al., 2019; Schneider et al., 2011; Schulz et al., 2018; Shrivastava et al., 2019; Talbot et al., 1990). For example, in the southwestern region of the Amazon, which is heavily impacted by BB, organics account for $f_{org,M} > 90$% in the dry season (Brito et al.2014). Note that the thermal stability of some organic species and tar balls in BB plumes can lead to an underestimation of the $f_{org,M}$ measured by the C-ToF-AMS (Adachi et al., 2018). Further, the organic matter in the UPL is significantly more oxidized than the fresh BB smoke, as shown in Fig. S7. This can be associated with the long aging times and the elevated $O_3$ mixing ratio in the UPL (Fig. 3h) (Martin et al., 2017). The rBC mass fractions account for $f_{rBC,M} = 15$ % in the UPL and $f_{rBC,M} = 12$% in the BB plume. A clear difference was observed for the mass fractions of the inorganic constituents sulfate ($SO_4^{2-}$), ammonium ($NH_4^+$), and nitrate ($NO_3^-$), which in sum account for $f_{inorg,M} = 35$ % in the UPL and $f_{inorg,M} = 16$ % in the BB*

*plume. The increased $f_{inorg,M}$ in the UPL can probably be explained by aging-related condensation of the secondarily formed species $SO_4^{2-}$, $NH_4^+$, and $NO_3^-$. On the other hand, the lower $f_{org,M}$ in the UPL compared to the fresh Amazonian BB is related to the evaporation of organics due to fragmentation during the aging over the Atlantic.*

3. We updated the values presented in the following sentence (L441-443): This possibility is supported by the relatively high sulfate content of the aerosol in this layer, which at an average value of *0.79* $\pm$ 0.02 µg m$^{-3}$ accounts for *23%* of total aerosol mass concentration *(Fig. 6)*.

4. Accordingly, a few modifications followed up in the conclusions section (L721-727):
*The plume was dominated by aerosol particles in the accumulation mode size range ($N_{acc}$ = 850 $\pm$ 330 cm$^{-3}$), peaking at ~130 nm diameter, and consisting mostly of particles containing non-volatile material. Remarkably, rBC particles appeared to be a dominant species, with mean number and mass concentrations of $N_{rBC}$ = 280 $\pm$ 110 cm$^{-3}$ and $M_{rBC}$ = 1.0 $\pm$ 0.4 µg m$^{-3}$, respectively. This accounts for ~40 % of total aerosol number and 15 % of the submicrometer aerosol mass concentrations. The UPL also shows high mass fractions of organics (50 %), sulfate (17 %), ammonium (8 %) and nitrate (10 %).*

[R1.6] It is because of the available datasets outlined in this paragraph and the one before it, that this reviewer felt cheated in that so much more could be derived from the aircraft dataset and thus provide a much more complete story. It is recommended that the authors mine their data for more nuggets of gold that are surly there.

[A1.6] We understand your criticism since the ACRIDICON-CHUVA data set is rich and unique, and therefore we tried to address most of your suggestions that fit the scope of this study. We tried to find the right balance in order to have a clear and comprehensive story as outlined in A1.1. For the discussion of other aspects of this data set, please look to our papers in preparation, which will be submitted to ACP.

Other specific issues:

[R1.7] Page 2, line 46. The authors state that the BC particle number fraction is $\sim$ 40%. I think this is wrong and that the value is closer to 30%. Using number concentrations reported on page 12 (lines 371, 372, and 383, N_CN,20 = 970, N_acc = 850, and N_rBC = 280 particles/cc) the fractions I derive are 0.29 (using N_CN,20) and 0.33 (using N_acc). As a rough check, derived values from figure 4 support the 30% fraction vs 40%.

[A1.7] Within individual plumes, we consider the enhancement ratio $\Delta N_{rBC}/\Delta N_{CN,20}$ to be more meaningful than the fraction $N_{rBC}/N_{CN,20}$. Moreover, $N_{rBC}$ equals 0 does not imply that $N_{CN,20}$ is also 0. For avoiding confusion between enhancement ratio and fraction, we clarified this in section 2.2 (L224-228):

The rBC enhancement ratio relative to CO ($En_{RBC,M} = \Delta M_{rBC}/\Delta c_{CO}$, where $\Delta$ is the difference between the concentration of the species in the plume and in the background atmosphere) was obtained by applying a bivariate fit to the rBC and CO correlation within individual pollution plumes. *Analogously, CCN and rBC enhancement ratios relative to total CPC particle counts ($\Delta N_{CCN,0.5}/\Delta N_{CN,20}$ and $\Delta N_{rBC}/\Delta N_{CN,20}$) were obtained by applying a bivariate fit between the respective quantities.*

For consistency, we include a panel in Fig. 5 for estimating the activated fraction $\Delta N_{CCN0.5}/\Delta N_{CN,20}$ for both the UPL and BB plumes using the regression method. Note that, from comment 1.4, we used a new flag for determining the BB interval, and therefore, the data points have been updated in Fig. 5.

[Figure]

***Figure 5.** Correlation between **(a)** rBC particle number concentrations ($N_{rBC}$) and total aerosol ($N_{CN,20}$); and between **(b)** CCN at S = 0.5 % ($N_{CCN,0.5}$) and total aerosol ($N_{CN,20}$) in the UPL (blue) and in the fresh biomass burning plume (green). The dashed lines are bivariate linear regressions applied to the data sets.*

Following the modifications in Fig. 5, we have changed the sentences:

(L377-379): *The CCN concentrations at S = 0.5 %, $N_{CCN,0.5}$, show a maximum within the UPL with $N_{CCN,0.5} = 560 \pm 180$ cm$^{-3}$ as well as a high CCN fraction, $f_{CCN,0.5} = 56 \pm 9$ % (Fig. 3d).*

(L384-386): *The ratio $\Delta N_{rBC}/\Delta N_{CN,20} \approx 40$ % in the UPL is much higher than $\Delta N_{rBC}/\Delta N_{CN,20} \approx 5$ % in the fresh BB plume (Fig. 5a).*

(L410-416): *Note that, despite the higher $\Delta N_{rBC}/\Delta N_{CN,20}$ in the UPL compared to the fresh BB (Fig. 5a), the UPL shows higher CCN activated fraction ($\Delta N_{CCN,0.5}/\Delta N_{CN,20} = 66$ %, Fig. 5b). The high CCN efficiency is likely due to internal mixing of rBC with sulfate, nitrate, and highly oxygenated organic aerosol. These findings, in combination with the UPL's large geographic*

*extent, suggests that it represents an aerosol and CCN reservoir of particular significance for the Amazonian cloud cycling and rainfall formation – i.e., cloud droplet formation and growth.*

(L725-727): Despite the large fraction of rBC, the aerosol in the UPL appeared to be very CCN efficient due to internal mixing of rBC with sulfate, nitrate, and oxygenated organic aerosol, with *~70 %* of particles activated at $S = 0.5\%$.

[R1.8] Page 3, Line 73 -75. The authors are reminded that semi-transparent coating also alter BC optical properties.

[A1.8] For clarity, we have added the word "semi-transparent" to the sentence (L85-89, see also comment A2.7): *The formation of non-absorbing or semi-transparent coatings on the BC cores changes the particles' optical properties (Fuller et al., 1999; Moffet and Prather, 2009; Pokhrel et al., 2017; Schnaiter, 2005; Zhang et al., 2015) as well as their ability to act as cloud condensation nuclei (CCN) (Laborde et al., 2013; Liu et al., 2017; Tritscher et al., 2011), which influences their atmospheric transport and lifetime.*

[R1.9] Page 5, line 138-139. The authors refer to the African latter as pollution, yet their discussion centers on biomass burning. Pollution tends to suggest an anthropogenic contribution. It is suggested that the authors that consider changes "pollution" to biomass burning or something that describes the layer as containing BB emissions.

[A1.9] Please refer to [A1.1] where we have also addressed this comment.

[R1.10] Page 7, 209- 211. The authors might want to examine the paper by Adachi et al., (AST, 52, 46-56, 2018) where these authors present data from a mid-latitude wildfire that exhibit thermal stability well above the 250 C employed in the TD. How would increased thermal stability alter conclusions derived in the present analysis?

[A1.10] The study by Adachi et al. (2018) discusses how aerosol particles react to high temperatures (up to 600°C) based on ambient samples collected from agricultural biomass burning. Specifically, they show that some organic species or tar balls do not completely vaporize after being heated to 600°C. From the data set presented in our study, we can expect some bias between the CPC counts after the thermodenuder (which heats our aerosol samples up to 250°C) and the AMS, which relies on vaporizing aerosol particles for measuring the mass spectra. The thermal stability of organics in BB plumes could be responsible for the minimum in the volatile fraction ($f_{vol}$) observed within the UPL, which coincides with a minimum in the ultrafine fraction ($f_{fine}$). Moreover, the AMS results can underestimate the concentration of organics. However, this won't interfere with the main conclusions of the present study. In order to make it clear for the reader, we have included the following sentence (L401-403): *Note that the thermal stability of some organic species and tar balls in BB plumes can lead to an underestimation of the $f_{org,M}$ measured by the C-ToF-AMS (Adachi et al., 2018).*

[R1.11] Page 8, line 250-252. In deriving the EnRbc ratio, is the CO background corrected? I assume so, but this should be explicitly stated.

[A1.11] Yes. For the long-term measurements at the ATTO site, we used weekly $5^{th}$ percentiles of CO and BCe measurements as background values, which were subsequently used for calculating the daily $EnR_{BCe}$ (Fig. 12g of the revised manuscript). In order to make it clear, we have added:
(L249): *The $5^{th}$ percentiles of the $BC_e$ and CO measurements of the corresponding week were used as background values.*

[R1.12] Page 11, line 337. Please be consistent with respect to referencing black carbon. Within the SP2 community, the "black carbon" that is detected by this instrument is referred to as rBC (refractory black carbon). Indeed, in many places the authors refer to "rBC". To help minimize confusion, please be consistent and use only one term.

395 [A1.12] Thanks for noting that. We have used the term rBC only when referring to SP2 results, and BC when generally speaking about black carbon. We have further modified this paragraph in response to the Referee #2, please also refer to comment R2.14.

[R1.13] Page 12, 389- 390. The authors write "The high frBC further agrees with the pronounced
400 brownish color of the visually observable layer in Fig. 2." This is a very misleading sentence. The brownish color could easily be due to brown carbon (BrC). Which, given the fact that the HALO payload included a three-wavelength PSAP, could be readily calculated via the Angstrom absorption exponent (AAE).

405 [A1.13] To avoid confusion, we changed the sentence to (L386): *Visually, the dark color of the layer observable in Fig. 2 corresponds with the high rBC fraction.*

[R1.14] Page 21, line 678-681. As highlighted above, the HALO 3-wavelength PSAP would provide insight into the presence of BrC.
410
[A1.14] Unfortunately, the PSAP data set could not be included on this study, as outlined in A1.2. But we will keep it in mind for further studies including the ACRIDICON-CHUVA data set. Please also refer to comment [A1.1].

415 [R1.15] Page 37, Table 2. Please explicitly cite the wavelength used to derive the SSA (637 nm).

[A1.15] Done.

**Response to referee comments and suggestions on acp-2019-775 by Holanda et al.**

**Response format description:**

Black text shows the original referee comment, blue text shows the authors response, and red text shows

5   quoted manuscript text. Changes to the manuscript text are shown as *italicized and underlined*. We used bracketed comment numbers for referee comments (e.g., [R2.1]) and author's responses (e.g., [A2.1]). Line numbers refer to the discussion/review manuscript.

10   **Anonymous Referee #2**

Received: 1 Nov 2019

General comment:

This paper describes the influx of African smoke to the Amazon during the dry season and discusses the

15   implications of this transport for the Amazon. This work is novel and of interest to the aerosol community. I do recommend this paper for publication; however, I suggest that the authors revise the current manuscript, which is too long and detailed in its current form.

We thank the referee for the critical evaluation of the manuscript and for the constructive suggestions.

Major Comments:

[R2.1] In its current form, this paper is well-written but has so many details, that I found myself losing the main point of a paragraph or an entire section. I also found that a lot of the major findings were ei-

25   ther buried in the middle of a section or at the end of a very long discussion that could have been cut in half. I list below a few suggested areas to tighten up this paper but encourage the authors to re-evaluate their paper as a whole and determine areas that could go into the supporting information.

[A2.1] Thanks for this critical feedback. As a response to the referee comment, we have revised the manuscript carefully in order to make the text and flow of arguments (wherever possible) shorter and more concise. We think that this helped to clarify certain aspects, such as the emphasizing the key findings of the study. Note however, that your request to generally shorten the manuscript and to reduce the discussions is somewhat in conflict to the overall recommendation of referee #1, who requested to rather include further data sets and analyses that (in her/his view) are missing (refer to comments R1.1 to R1.6 by referee #1). Ultimately, we tried to account for the general criticism of both referees by (i) generally streamlining and shortening the text wherever appropriate and (ii) adding some more information to the analysis that was explicitly requested in order to strengthen the manuscript.

[R2.2] This paper has too many acronyms to keep track of. I found myself having to go back to the acronym table a lot, which made the paper difficult to read. I suggest that the authors try to reduce the number of terms and instruments discussed to focus only on the most important ones for this story and to place the rest of the measurements in the supporting information.

[A2.2] We agree and revised the entire manuscript in order to reduce the number of acronyms wherever possible/appropriate.

[R2.3] I felt that the abstract and introduction promise that the focus of the paper will be on radiative impacts but those impacts weren't as well-emphasized in the results. I suggest restructuring the conclusions section to include a brief overview of the findings then focus on the radiative impacts as an extension of their findings.

[A2.3] Actually, the radiative influence of the observed pollution layer over the Atlantic is rather a side aspect of the study. We added it, envisioning that it may be of interest to a certain fraction of the readers. Moreover, many recent studies are dedicated to estimate the radiative impacts of the African BB layer over the South Atlantic (e.g. Deaconu et al., 2019; Denjean et al., 2019; Lu et al., 2018; Mallet et al., 2019; Meyer et al., 2013). While most of these studies focus on the

eastern Atlantic Ocean, here we want to bring the attention to the change on the radiative forcing towards the western Atlantic Ocean. Moreover, in order to calculate the direct radiative forcing of aerosol particles at the top of the atmosphere (DRF-TOA), we have used the multi-year and vertically-resolved aerosol extinction coefficients derived by CALIPSO. These results integrate the third large part of the paper, which shows the remote sensing results for different longitudinal bands over the Atlantic and the calculated radiative forcing. Accordingly, we have modified the *summary and conclusions* section in order to address the referee comment (L739):

*Based on the remote sensing data, we further calculate the DRF-TOA exerted by the pollution layer as a function of longitude. We found that the aging of the plume leads to a change in the DRF-TOA from a positive (warming) to a negative (cooling) effect as it moves westwards over the Atlantic.*

Specific Comments:

Abstract:

[R2.4] Lines 36-40: This sentence is far too long and dense. I suggest breaking it up.

[A2.4] Thanks for pointing this out. We modified the paragraph as follows (L38-42): Black carbon (BC) aerosols are influencing the Earth's atmosphere and climate, but their microphysical properties, spatiotemporal distribution, and long-range transport are not well constrained. *This study presents airborne observations of the transatlantic transport of BC-rich African biomass burning (BB) smoke into the Amazon Basin using a single particle soot photometer (SP2) as well as several complementary techniques. We base our results on observations of aerosols and trace gases off the Brazilian coast onboard the research aircraft HALO during the ACRIDICON-CHUVA campaign in September 2014.*

[R2.5] The abstract should include a very brief clause or sentence describing the most relevant instruments used.

[A2.5] Thanks – done. Please refer to comment [2.4], where we have also addressed this point in our response.

[R2.6] Lines 46-47: I had a hard time understanding this part of your sentence.

[A2.6] We modified this sentence as follows (L47-53): *The aged smoke is characterized by a dominant accumulation mode, centered at about 130 nm, with a particle concentration of $N_{acc} = 850 \pm 330$ cm$^{-3}$. The rBC particles account for ~15 % of the submicrometer aerosol mass and ~40 % of total aerosol number concentration. This corresponds to a mass concentration range from 0.5 to 2 µg m$^{-3}$ (1$^{st}$ to 99$^{th}$ percentiles) and a number concentration range from 90 to 530 cm$^{-3}$. Along with rBC, high $c_{CO}$ (150 ± 30 ppb) and $c_{O3}$ (56 ± 9 ppb) mixing ratios support the biomass burning origin and pronounced photochemical aging of this layer.*

Introduction:

[R2.7] I suggest condensing the first two paragraphs down to half a paragraph. There is a lot of detail that is not necessary.

[A2.7] We agree. Since the radiative effects of the pollution plumes are not the central focus of the manuscript, we condensed the two first paragraphs into the following one (L69-89):
*Biomass burning (BB) in the African and South American tropics and subtropics represents a globally significant source of atmospheric aerosol particles and trace gases (Andreae, 1991; Andreae et al., 1988; Barbosa et al., 1999; Ichoku and Ellison, 2014; Kaiser et al., 2012; Reddington et al., 2016; van der Werf et al., 2017). A major constituent of BB smoke is black carbon (BC), which is co-emitted along with organic aerosols and inorganic salts in proportions that depend on the fuel type and fire phase (Allen and Miguel, 1995; Andreae, 2019; Andreae and Merlet, 2001; Jen et al., 2019; Levin et al., 2010; Reid et al., 2005). The BC aerosol is a key component in the climate system as it significantly influences the Earth's radiative budget through the so-called direct, semi-direct, and indirect aerosol effects (Bond et al., 2013; Bou-*

*cher et al., 2016; Brioude et al., 2009; Koch and Del Genio, 2010; Stocker et al., 2013). Recent studies have classified BC as the second largest contributor to global warming and estimated its direct radiative forcing as high as +1.1 W m$^{-2}$, with 90 % uncertainty bounds spanning from +0.17 to +2.1 W m$^{-2}$ (Bond et al., 2013 and references therein). This large uncertainty arises from our poor understanding of the BC microphysical properties and its spatiotemporal distribution in the atmosphere (Boucher et al, 2013, Andreae and Ramanathan, 2013). During their typical atmospheric lifetime of several days, BC particles undergo photochemical aging, creating internally mixed BC aerosols via the condensation of low and semi-volatile compounds, coagulation, and cloud processing (Bond et al., 2013; Cubison et al., 2011; Konovalov et al., 2017, 2019; Schwarz et al., 2008; Willis et al., 2016). The formation of non-absorbing or semi-transparent coatings on the BC cores changes the particles' optical properties (Fuller et al., 1999; Moffet and Prather, 2009; Pokhrel et al., 2017; Schnaiter, 2005; Zhang et al., 2015) as well as their ability to act as cloud condensation nuclei (CCN) (Laborde et al., 2013; Liu et al., 2017; Tritscher et al., 2011), which influences their atmospheric transport and lifetime.*

120

125

130

[R2.8] I also suggest adding a sentence or two regarding the impact of aging on the direct effect. It has been shown in several studies that coatings can greatly alter the radiative properties of black carbon [Moffet and Prather, 2009].

135 [A2.8] This comment has been also addressed in the comment [2.7]. Please refer to L85 "*The formation of non-absorbing or semi-transparent coatings on the BC cores changes the particles' optical properties (Fuller et al., 1999; Moffet and Prather, 2009; Pokhrel et al., 2017; Schnaiter, 2005; Zhang et al., 2015) as well as their ability to act as cloud condensation nuclei (CCN) (Laborde et al., 2013; Liu et al., 2017; Tritscher et al., 2011), which influences their at-*
140 *mospheric transport and lifetime.*

[R2.9] Lines 114-118 can be cut as they do not add much to the introduction.

[A2.9] We agree with that. Therefore, we have removed the first sentence completely: "Moreover, the arrival of a plume of remarkably sulfur-rich aerosol particles in the central Amazon in September 2014 has been traced back to the LRT of volcanogenic emissions from eastern Congo (Saturno et al., 2018a)". Also, we have moved the second sentence into the caption of Fig. 7 of the revised manuscript: "For general illustration, animations  of the Goddard Earth Observing Model (Version 5, GEOS-5) show that aerosol particles are transported efficiently from Africa to South America and to a lesser extent from South America to Africa (Colarco et al., 2010; Yasunari et al., 2011)."

[R2.10] The sentence on lines 121-122 is not clear to me.

[A2.10] In order to make it clearer, we have substantially reduced and modified the paragraph (L106-119): *Several studies have found that the long-range transport (LRT) of long-lived species from Africa plays a major role for the Amazonian atmospheric composition. The transport of dust from distant sources into the heart of the Amazon Basin was first observed in 1977, although Africa was not identified as the source region at the time (Lawson and Winchester, 1979).* Subsequently, the plume-wise LRT of African dust and smoke during the Amazonian wet season has been well documented (Ansmann et al., 2009; Baars et al., 2011; Barkley et al., 2019; Moran-Zuloaga et al., 2018; Swap et al., 1992; Talbot et al., 1990; Wang et al., 2016). *The LRT of aerosols occurs also during the Amazonian dry season, when smoke from the intense African BB plays a substantial role.* The earliest observations of such pollution layers in the free troposphere over the Brazilian coast can be found in ozone ($O_3$) soundings made from Natal, on the east coast of Brazil (5.8º S, 35.2º W), where mixing ratios of ~70 ppb were measured with a maximum *in the month of* September (Kirchhoff et al., 1983; Logan and Kirchhoff, 1986). These measurements were continued over a ten-year period (1978-1988), confirming the climatological presence of a tropospheric $O_3$ maximum over the Brazilian coast, centered at the 500 hPa pressure level and peaking in the September-October period (Kirchhoff et al., 1991).

Methods:

[R2.11] I strongly suggest reducing the text in this section. For example, the gas phase data does not play a major role compared to the aerosol data. I suggest mentioning what was measured in the gas phase then directing the reader to the SI for more details on the instrumentation. This will also reduce the number of acronyms that the reader must memorize.

[A2.11] We understand your point, but have a somewhat opposing opinion here. In our view, every aspect that is prominently shown and discussed in the results or discussion part should also have an appropriate visibility in the methods section. We regard it as more confusing than helpful to distribute the information on methods and data analysis between main text and supplement. We critically checked the trace gas paragraph, did few improvements here and there, and finally came to the conclusion that this section is concise and rather easy to understand. Therefore, we prefer to keep the paragraph in the section 2.1, which is dedicated to describe the HALO data sets that we made use of.

[R2.12] I also suggest placing details about AIRS data and AERONET in the SI. The AIRS and AERONET data were not as integral to the study as other methods.

[A2.12] For the same reason stated in comment [2.11], we prefer to kept this information in section 2.4 of the methodology. However, we have substantially reduced the length and content of these paragraphs by removing (too) detailed information about remote sensing techniques, which are not the central focus of the study. Therefore, the following changes were made in the section (L254-268):

*2.4.   Satellite and ground-based remote sensing*

*In this study, we used the vertically-resolved extinction coefficients (LIDAR Level 2 Version 3 Aerosol Profile product with 5 km horizontal resolution) of the Cloud-Aerosol Lidar with Orthogonal Polarization (CALIOP) lidar system, onboard the Cloud-Aerosol Lidar and Infrared Pathfinder Satellite Observations (CALIPSO) satellite (Winker et al., 2009). The CALIPSO al-*

*gorithms detect and classify aerosol layers based on their observed physical and optical properties into the subclasses: polluted continental, biomass burning (smoke), desert dust, polluted dust, clean continental, and marine aerosol (Omar et al., 2009).*

205       *To obtain CO concentrations between the 400 and 600 hPa pressure levels, we used the Atmospheric Infrared Sounder (AIRS) onboard the NASA Aqua satellite available from the Giovanni online data system (https://giovanni.gsfc.nasa.gov/giovanni/, last access on 13 June 2019). Daily averages of aerosol optical depth (AOD) at 550 nm with original grid resolution of $1^o$ x $1^o$ was obtained from Moderate Resolution Imaging Spectroradiometer (MODIS) aerosol products*

210 *from the NASA Terra and Aqua satellites (Remer et al., 2005). Finally, AOD at 500 nm (level 2.0) was obtained by direct sun measurements in Ascension Island ($7.976^o$ S, $14.415^o$ W), using the CIMEL sunphotometer of the AErosol RObotic NETwork (AERONET, https://aeronet.gsfc.nasa.gov/, last access 12 Mar 2019) (Holben et al., 1998).*

215 We further streamlined the section 2.1 (L147-156):

**2.1. The ACRIDICON-CHUVA campaign**

*The data presented here were obtained during flight AC19 of the ACRIDICON-CHUVA aircraft campaign (Machado et al., 2017; Wendisch et al., 2016), which took place over the Atlantic*

220 *Ocean and the Amazon Basin on 30 September 2014. The main objective of ACRIDICON-CHUVA was to study the interactions between aerosol particles, deep convective clouds, and atmospheric radiation using a broad set of instruments for airborne observations of aerosol physical and chemical properties, trace gases, radiation, and cloud. The measurements were conducted onboard the German HALO (High Altitude and LOng range) research aircraft, oper-*

225 *ated by the German Aerospace Center (DLR), covering a wide geographic area of the Amazon Basin and probing different pollution states by means of highly resolved atmospheric profiles (altitudes up to 15 km).*

230 Results:

[R2.13] I suggest placing Figure 2 in the SI.

[A2.13] Evidently, this figure is not really essential for the key findings of the paper, however, it also does not really disturb the flow of reading as it is rather 'light'. We think that figure gives the reader a good visual impression of the pollution layer and, thus, aids the perception of the numbers presented in the manuscript. It further visually underlines how pronounced the layers actually are. Since the supplementary material typically receives significantly less attention, we prefer to keep Fig. 2 in the main text.

[R2.14] Sections 3 and 3.1 are too long and detailed. This information dilutes the major findings regarding the transport of biomass burning and its impact on the optical and cloud nucleating properties of Amazonian aerosol. I suggest condensing the material from 330-394 down to a paragraph if possible and placing much of this information in the SI.

[A2.14] The main aspect of the study is the *in situ* observation of the pollution layer at the South American coast. A unique aspect of our study is the broad spectrum of techniques in the aircraft that were available to probe the layer in great detail. Accordingly, a systematic and detailed summary of the key numbers, concentrations, and properties is the foundation for other aspects of the analysis. Accordingly, we strongly prefer to keep some aspects as detailed as it is. However, after revising the section, we performed some changes in order to make it more concise:

We moved the following sentence to the caption of Fig. S3 of the revised manuscript: The sounding shows similar tropospheric stratification as presented in Fig. 3. The first layer (top around 1000 m) is associated with the boundary layer, the second (top around 3200 m) is related to the shallow clouds top and the third one (around 5000 m) is the large scale inversion.

(L331-348): *The flight track of AC19 followed the direction of the Amazon River from Manaus towards the coast and included cloud-profiling maneuvers over the Atlantic Ocean (Fig. 1). A*

260   *remarkable observation during AC19 was the strong stratification of the troposphere over the ocean with vertically well-defined and horizontally extended layers, with varying degrees of pollution (Table 1). Based on contrasting aerosol concentrations, size ranges, and composition, we distinguished an upper and a lower pollution layer (UPL and LPL) with a horizontal clean air mass layer (CL) in between. The layers were discernible visually from the aircraft cockpit (Fig.*

265   *2).*

   *In this study, we present the tropospheric stratification for the lowest 5 km of the atmosphere, focusing primarily on aerosol and trace gas properties within the UPL and contrast them with the properties of the CL, LPL and the marine boundary layer (MBL). Aerosol properties in the upper troposphere during ACRIDICON-CHUVA have been characterized in previous studies*

270   *(Andreae et al., 2018; Schulz et al., 2018). Upon ascent and descent, the UPL was probed six times at offshore locations[1], right before it reached the South American continent, and two times onshore ~200-400 km from the coast line (blue squares in Fig. 1). The eight UPL penetrations were several hundred kilometers apart from each other, underlining the large horizontal extent of the layer. Later on the route back to Manaus airport, we observed an active fire plume north-*

275   *west of Belém (green square in Fig. 1, photo of plume in Fig. S3). This plume was probed at ~1 km above the fire and is expected to be only a few minutes old. Below, selected aerosol properties in this local, fresh BB plume are contrasted with the UPL aerosol properties.*

(L356-358): Generally, the profile of $\theta$ indicates rather stable conditions along the entire profile

280   with the UPL being centered at ~3.5 km altitude. *For comparison, radiosonde profiles at Belém airport for the same day as flight AC19 are shown Figure S4.*

[R2.15] Lines 442-443: It is not necessary to explain that O3 is a secondary pollutant produced photo-

285  chemically in BB plumes.
* * *
[1] Note that we count two passages through the layer over the Amazon River delta as offshore.

[A2.15] In response to your comment, we shortened and streamlined the entire paragraph substantially, as follows (L425-428): *The ozone as secondary pollutant also presents a maximum within the UPL ($c_{O3}$ = 56 ± 9 ppb) and appears to be anti-correlated with NO ($c_{NO}$ = 0.10 ± 0.02 ppb). Therefore, the fact that $O_3$ and $NO_y$ ($c_{NOy}$ = 2.5 ± 0.8 ppb) are strongly enhanced in the pollution layers, reflects the photochemical age of the plume.*

[R2.16] Lines 447-467: This paragraph could go in a discussion section.

[A2.16] All the discussions in this manuscript have been placed along with the results. For consistency, we would prefer to keep in this section the discussion on further airborne observations of BB layers near the African coast. This paragraph is also a link to the next section where we look more in detail the evolution of BB layers over the South Atlantic Ocean.

[R2.17] Lines 470-473 belong in the introduction.

[A2.17] We moved the sentence to the introduction section (L90) as suggested: The Amazonian atmosphere is strongly influenced by the yearly *north-south oscillation of the intertropical convergence zone (ITCZ) (Andreae et al., 2012; Martin et al., 2010; Pöhlker et al., 2019),* which causes a pronounced seasonality in aerosol concentrations (e.g., BC and CCN) and other aerosol properties (e.g., single scattering albedo) (Roberts et al., 2001; Roberts, 2003; Martin et al., 2010; Artaxo et al., 2013; Rizzo et al., 2013; Andreae et al., 2015; Pöhlker et al., 2016; Saturno et al., 2018b).

[R2.18] The authors should note that the methodology shown in Figure 7 is similar to the methods used in [Barkley et al., 2019] to identify similar plumes.

[A2.18] We included (L562): *More examples of similar layers over the Atlantic Ocean in September 2016 are shown and discussed in Barkley et al. (2019).*

[R2.19] Lines 591-606 are really important for understanding the implications of your work. I strongly suggest either giving this paragraph its own section or placing it in a discussion section.

[A2.19] Please refer to comment A2.3. The DRF-TOA is rather a side aspect and is an extension of what we can get from satellite derived aerosol extinction coefficients.

320

\* *Correspondence to:* M. Pöhlker (m.pohlker@mpic.de) and C. Pöhlker (c.pohlker@mpic.de)

**Abstract**

Black carbon (BC) aerosols are influencing the Earth's atmosphere and climate, but their microphysical properties, spatiotemporal distribution, and long-range transport are not well constrained. This study presents airborne observations of the transatlantic transport of BC-rich African biomass burning (BB) smoke into the Amazon Basin using a single particle soot photometer (SP2) as well as several complementary techniques. We base our results on observations of aerosols and trace gases off the Brazilian coast onboard the research aircraft HALO during the ACRIDICON-CHUVA campaign in September 2014.

During flight AC19 over land and ocean at the northeastern coastline of the Amazon Basin, we observed a BC-rich layer at ~3.5 km altitude with a vertical extension of ~0.3 km. Backward trajectories suggest that fires in African grasslands, savannas, and shrublands were the main source of this pollution layer, and that the observed BB smoke had undergone more than 10 days of atmospheric transport and aging over the South Atlantic before reaching the Amazon Basin. The aged smoke is characterized by a dominant accumulation mode, centered at about 130 nm, with a particle concentration of $N_{acc} = 850 \pm 330$ cm$^{-3}$. The rBC particles account for ~15 % of the submicrometer aerosol mass and ~40 % of total aerosol number concentration. This corresponds to a mass concentration range from 0.5 to 2 μg m$^{-3}$ (1$^{st}$ to 99$^{th}$ percentiles) and a number concentration range from 90 to 530 cm$^{-3}$. Along with rBC, high $c_{CO}$ (150 ± 30 ppb) and $c_{O3}$ (56 ± 9 ppb) mixing ratios support the biomass burning origin and pronounced photochemical aging of this layer. Upon reaching the Amazon Basin, it started to broaden and to subside, due to convective mixing and entrainment of the BB aerosol into the boundary layer. Satellite observations show that the transatlantic transport of pollution layers is a frequently occurring process, seasonally peaking in August/September.

By analyzing the aircraft observations together with the long-term data from the Amazon Tall Tower Observatory (ATTO), we found that the transatlantic transport of African BB smoke layers has a strong impact on the north and central Amazonian aerosol population during the BB-influenced season (July to December). In fact, the early BB season (July to September) in this part of the Amazon appears to be dominated by African smoke, whereas the later BB season (October to December) appears to be dominated by South American fires. This dichotomy is reflected in pronounced changes of aerosol optical properties such as the single scattering albedo (increasing from 0.85 in August to 0.90 in November) and the BC-to-CO enhancement ratio (decreasing from 11 to 6 ng m$^{-3}$ ppb$^{-1}$). Our results suggest that, despite the high fraction of BC particles, the African BB aerosol acts as efficient cloud condensation nuclei (CCN) with potentially important implications for aerosol-cloud interactions and the hydrological cycle in the Amazon.

**1. Introduction**

Biomass burning (BB) in the African and South American tropics and subtropics represents a globally significant source of atmospheric aerosol particles and trace gases (Andreae, 1991; Andreae et al., 1988; Barbosa et al., 1999; Ichoku and Ellison, 2014; Kaiser et al., 2012; Reddington et al., 2016; van der Werf et al., 2017). A major constituent of BB smoke is black carbon (BC), which is co-emitted along with organic aerosols and inorganic salts in proportions that depend on the fuel type and fire phase (Allen and Miguel, 1995; Andreae, 2019; Andreae and Merlet, 2001; Jen et al., 2019; Levin et al., 2010; Reid et al., 2005). The BC aerosol is a key component in the climate system as it significantly influences the Earth's radiative budget through the so-called direct, semi-direct, and indirect aerosol effects (Bond et al., 2013; Boucher et al., 2016; Brioude et al., 2009; Koch and Del Genio, 2010; Stocker et al., 2013). Recent studies have classified BC as the second largest contributor to global warming and estimated its direct radiative forcing as high as +1.1 W m$^{-2}$, with 90 % uncertainty bounds spanning from +0.17 to +2.1 W m$^{-2}$ (Bond et al., 2013 and references therein). This large uncertainty arises from our poor understanding of the BC microphysical properties and its spatiotemporal distribution in the atmosphere (Boucher et al, 2013, Andreae and Ramanathan, 2013). During their typical atmospheric lifetime of several days, BC particles undergo photochemical aging, creating internally mixed BC aerosols via the condensation of low and semi-volatile compounds, coagulation, and cloud processing (Bond et al., 2013; Cubison et al., 2011; Konovalov et al., 2017, 2019; Schwarz et al., 2008; Willis et al., 2016). The formation of non-absorbing or semi-transparent coatings on the BC cores changes the particles' optical properties (Fuller et al., 1999; Moffet and Prather, 2009; Pokhrel et al., 2017; Schnaiter, 2005; Zhang et al., 2015) as well as their ability to act as cloud condensation nuclei (CCN) (Laborde et al., 2013; Liu et al., 2017; Tritscher et al., 2011), which influences their atmospheric transport and lifetime.

The Amazonian atmosphere is strongly influenced by the yearly north-south oscillation of the intertropical convergence zone (ITCZ) (Andreae et al., 2012; Martin et al., 2010; Pöhlker et al., 2019), which causes a pronounced seasonality in aerosol concentrations (e.g., BC and CCN) and other aerosol properties (e.g., single scattering albedo) (Roberts et al., 2001; Roberts, 2003; Martin et al., 2010; Artaxo et al., 2013; Rizzo et al., 2013; Andreae et al., 2015; Pöhlker et al., 2016; Saturno et al., 2018b). This makes the central Amazon Basin an ideal environment to study atmospheric and biogeochemical processes as a function of the highly variable aerosol population. During the wet season (February to May), trace gas and aerosol emissions from the regional biosphere predominantly regulate atmospheric cycling, precipitation patterns, and regional climate (Pöhlker et al., 2012; Pöschl et al., 2010). Average wet season black carbon (BC) mass concentrations, $M_{BC}$, are ~0.07 µg m$^{-3}$ and $M_{BC}$ approaches zero during pristine episodes (Andreae and Gelencsér, 2006; Pöhlker et al., 2018). In contrast, the dry season (August to November) is characterized by intense and persistent BB emissions, changing substantially the atmospheric composition and cycling (Artaxo et al., 2013; Rizzo et al., 2013). Average dry season BC mass concentrations ($M_{BC}$) in central Amazonia are ~0.4 µg m$^{-3}$ with peaks reaching ~0.9 µg m$^{-3}$ (Pöhlker et al., 2018; Saturno et al., 2018b) while in the southern hot-spot regions of agriculture-related burning, the average $M_{BC}$ can be as

105   high as ~2.8 µg m$^{-3}$ (Artaxo et al., 2013).

Several studies have found that the long-range transport (LRT) of long-lived species from Africa plays a major role for the Amazonian atmospheric composition. The transport of dust from distant sources into the heart of the Amazon Basin was first observed in 1977, although Africa was not identified as the source region at the time (Lawson and Winchester, 1979). Subsequently, the plume-wise LRT of African dust and

110   smoke during the Amazonian wet season has been well documented (Ansmann et al., 2009; Baars et al., 2011; Barkley et al., 2019; Moran-Zuloaga et al., 2018; Swap et al., 1992; Talbot et al., 1990; Wang et al., 2016). The LRT of aerosols occurs also during the Amazonian dry season, when smoke from the intense African BB plays a substantial role. The earliest observations of such pollution layers in the free tropo-sphere over the Brazilian coast can be found in ozone (O$_3$) soundings made from Natal, on the east coast

115   of Brazil (5.8º S, 35.2º W), where mixing ratios of ~70 ppb were measured with a maximum in the month of September (Kirchhoff et al., 1983; Logan and Kirchhoff, 1986). These measurements were continued over a ten-year period (1978-1988), confirming the climatological presence of a tropospheric O$_3$ maximum over the Brazilian coast, centered at the 500 hPa pressure level and peaking in the September-October period (Kirchhoff et al., 1991).

120   The first comprehensive airborne measurements off the South American coast, made in 1989 near Na-tal, could also attribute these pollutions layers to LRT of African BB emissions (Andreae et al., 1994). Additional aircraft campaigns in southern Africa, the tropical South Atlantic, and the Amazon Basin have found pollution layers in the free troposphere with similar characteristics (e.g., Andreae et al., 1988; Diab et al., 1996; Thompson et al., 1996; Bozem et al., 2014; Marenco et al., 2016). Recent studies in the cen-

125   tral basin, at the Amazon Tall Tower Observatory (ATTO), and at the  northeastern edge of the Amazon found indications for a significant abundance of African smoke during the Amazonian dry season (Barkley et al., 2019; Pöhlker et al., 2019, 2018; Saturno et al., 2018b; Wang et al., 2016). However, robust quanti-tative data from observations and/or models (e.g., African BC and CCN fractions in the Amazon basin) have remained sparse.

130   This study focusses on the transatlantic transport of African BB smoke into the Amazon Basin by combining *in-situ* aircraft observations, modeling results, and remote sensing data. The core of this work are aircraft observations made within a defined African pollution layer upon its arrival at the South Amer-ican coast during the ACRIDICON-CHUVA campaign over Amazonia in September 2014 (Wendisch et al., 2016). We focus primarily on the spatiotemporal distribution and advection dynamics of the BB smoke

135   layers by analyzing (i) aerosol and trace gas concentration profiles, (ii) backward trajectories and African BB source regions, (iii) the seasonality of the pollution transport, (iv) the horizontal and vertical extent of the transported layers, and (v) the convective mixing and smoke entrainment from the layers into the planetary boundary layer as they are transported from the ocean into the South American continent. Note that a detailed characterization of the microphysical aerosol properties within the BB smoke layers (e.g., the BC core diameters and mixing state) is beyond the scope of the present work and will be the subject of a separate and extended follow-up study (Holanda et al., in preparation). As a final step of the present study, we integrate its key results into the broader picture of the long-term aerosol observations at the central Amazonian ATTO site to estimate the relevance of African pollution for the aerosol life cycle in the dry season.

**2. Materials and methods**

**2.1. The ACRIDICON-CHUVA campaign**

The data presented here were obtained during flight AC19 of the ACRIDICON-CHUVA aircraft campaign (Machado et al., 2017; Wendisch et al., 2016), which took place over the Atlantic Ocean and the Amazon Basin on 30 September 2014. The main objective of ACRIDICON-CHUVA was to study the interactions between aerosol particles, deep convective clouds, and atmospheric radiation using a broad set of instruments for airborne observations of aerosol physical and chemical properties, trace gases, radiation, and cloud. The measurements were conducted onboard the German HALO (High Altitude and LOng range) research aircraft, operated by the German Aerospace Center (DLR), covering a wide geographic area of the Amazon Basin and probing different pollution states by means of highly resolved atmospheric profiles (altitudes up to 15 km).

[revised manuscript text omitted]

Likewise, the volatile fraction ($f_{vol}$) is obtained from the difference between aerosol counts measured by the two CPCs (with and without a thermodenuder) divided by $N_{CN,20}$.

The CCN concentrations, $N_{CCN}$, were measured with a two-column CCN counter (CCNC, model CCN-200, DMT, Longmont, CO, USA) (Krüger et al., 2014; Roberts and Nenes, 2005; Rose et al., 2008). In this study, we used only the measurements at constant supersaturation ($S = 0.52 \pm 0.05$ %). The activated fraction, $f_{CCN,0.5}$, was calculated by dividing $N_{CCN,0.5}$ over $N_{CN,20}$.

A compact time-of-flight aerosol mass spectrometer (C-ToF-AMS, Aerodyne Research, Inc., Billerica, MA, USA) measured the mass concentration of four chemical species (i.e., organics, sulfate, nitrate, and ammonium) of the submicrometer aerosol with a time resolution of 30 seconds (Drewnick et al., 2005; Schulz et al., 2018). A complete description of the instrument and its operation during the ACRIDICON-CHUVA campaign is given in Schulz et al. (2018) and Andreae et al. (2018).

A dual-cell ultraviolet (UV) absorption detector (TE49C, Thermo Scientific) operating at a wavelength of $\lambda = 254$ nm was used to measure $O_3$ with precision of 2 % or 1 ppb. The CO mixing ratio was detected with a fast-response fluorescence instrument (AL5002, Aerolaser, Garmisch, Germany) (Gerbig et al., 1999). NO and total reactive nitrogen, $NO_y$ were measured by a modified dual-channel chemiluminescence detector (CLD-SR, Ecophysics) in connection with a gold converter (Baehr, 2003; Ziereis et al., 2000). More details on the measurement techniques can be found in Andreae et al. (2018).

The rBC enhancement ratio relative to CO ($EnR_{BC,M} = \Delta M_{rBC}/\Delta c_{CO}$, where $\Delta$ is the difference between the concentration of the species in the plume and in the background atmosphere) was obtained by applying a bivariate fit to the rBC and CO correlation within individual pollution plumes. Analogously, CCN and rBC enhancement ratios relative to total CPC particle counts ($\Delta N_{CCN,0.5}/\Delta N_{CN,20}$ and $\Delta N_{rBC}/\Delta N_{CN,20}$) were obtained by applying a bivariate fit between the respective quantities. Note that the best fit for the $c_{CO}$ vs. $M_{rBC}$ correlation was obtained after multiplying the $c_{CO}$ by a factor such that their means are equivalent, and then multiplying the resulting fit parameter by the same factor to obtain the $EnR_{BC,M}$.

**2.3. Ground-based aerosol and trace gas measurements at ATTO**

The ATTO site was established in 2010/2011 as a research platform for in-depth and long-term measurements of aerosol particles and trace gases as well as meteorological and ecological parameters in the central Amazon rain forest (Andreae et al., 2015). The research site is located 150 km northeast of Manaus, in a region characterized by periodic pristine atmospheric conditions during parts of the wet season versus strong BB pollution during the dry season (Pöhlker et al., 2016, 2018; Saturno et al., 2018b). The present study includes ATTO data of the aerosol absorption coefficient at $\lambda = 637$ nm, $\sigma_{ap}$, using the Multiangle Absorption Photometer (MAAP, model 5012, Thermo Electron Group, Waltham, USA) and the aerosol scattering coefficients, $\sigma_{sp}$, using a nephelometer (model Aurora 3000, Ecotech Pty Ltd., Knoxfield, Australia), respectively. The $M_{BCe}$ was calculated using a mass absorption cross section of 12.3 m$^2$ g$^{-1}$ for the dry season, as obtained by Saturno et al. (2018b). The single scattering albedo (SSA), which characterizes the absorption properties of an aerosol population, is defined as scattering divided by total extinction (absorption + scattering). All data were normalized to standard temperature and pressure (STP, $T_0$ = 273.15 K, $p_0$ =1013.25 hPa). The CCN concentrations at a supersaturation of 0.5 %, $N_{CCN}$ ($S$ = 0.5 %), were calculated using long-term scanning mobility particle sizer (SMPS) data and the κ-Köhler parametrization as described in Pöhlker et al. (2016). For more details about the aerosol optical properties characterization and CCN observations we refer to Saturno et al. (2018b) and Pöhlker et al. (2016, 2018), respectively. Further details on CO measurements conducted at ATTO site can be found in Winderlich et al. (2010) and Andreae et al. (2015). Daily $EnR_{BC}$ was calculated by applying a bivariate regression fit to 30-min averages of $\Delta BC_e$ and $\Delta CO$. The 5$^{th}$ percentiles of the BC$_e$ and CO measurements of the corresponding week were used as background values.

**2.4. Satellite and ground-based remote sensing**

In this study, we used the vertically-resolved extinction coefficients (LIDAR Level 2 Version 3 Aerosol Profile product with 5 km horizontal resolution) of the Cloud-Aerosol Lidar with Orthogonal Polarization (CALIOP) lidar system, onboard the Cloud-Aerosol Lidar and Infrared Pathfinder Satellite Observations (CALIPSO) satellite (Winker et al., 2009). The CALIPSO algorithms detect and classify aerosol layers based on their observed physical and optical properties into the subclasses: polluted continental, biomass burning (smoke), desert dust, polluted dust, clean continental, and marine aerosol (Omar et al., 2009).

To obtain CO concentrations between the 400 and 600 hPa pressure levels, we used the Atmospheric Infrared Sounder (AIRS) onboard the NASA Aqua satellite available from the Giovanni online data system (https://giovanni.gsfc.nasa.gov/giovanni/, last access on 13 June 2019). Daily averages of aerosol optical depth (AOD) at 550 nm with original grid resolution of 1$^o$ x 1$^o$ was obtained from Moderate Resolution Imaging Spectroradiometer (MODIS) aerosol products from the NASA Terra and Aqua satellites (Remer et al., 2005). Finally, AOD at 500 nm (level 2.0) was obtained by direct sun measurements in Ascension Island (7.976$^o$ S, 14.415$^o$ W), using the CIMEL sunphotometer of the AErosol RObotic NETwork (AERONET, https://aeronet.gsfc.nasa.gov/, last access 12 Mar 2019) (Holben et al., 1998).

**2.5. Direct radiative forcing at the top of the atmosphere**

In this study, we used the library for radiative transfer (LibRadtran) (Emde et al., 2016) with the uvspec tool to calculate the direct radiative forcing at the top of the atmosphere (DRF-TOA) by aerosol particles in the BB layer in the region of the South Atlantic Ocean. To solve the radiative transfer equation we chose the Discrete Ordinate Radiative Transfer solver (DISORT) 2 (Evans, 1998; Tsay et al., 2000). The

setup for the atmosphere was based on the standard tropical profile (Anderson et al., 1986), which was modified with measurement data. The vertical profiles of the mean aerosol extinction coefficient were calculated based on multiyear (2012-2018) CALIPSO retrievals. The extraterrestrial spectrum was used as described in Gueymard (2004). A wavelength range from 300 to 4000 nm was considered. The *ocean* was set as underlying surface. The AOD of the plume was calculated by integrating the mean extinction coefficient over the altitude band of the pollution layer (1 – 5 km). A SSA of 0.84 was assumed for the smoke layer based on Zuidema et al. (2016) and section 3.5. of the present study. An asymmetry parameter of 0.7 was used based on the typical BC value presented in Cheng et al. (2014). With the above parametrization, we obtained the mean daily value for the DRF-TOA along different longitudes.

**2.6. Backward trajectory modelling and fire intensities**

The hybrid single-particle Lagrangian integrated trajectory (HYSPLIT) model (Stein et al., 2015) was used to obtain systematic and multiyear sets of backward trajectories (BTs) for the ATTO site as outlined in detail in Pöhlker et al. (2019). The time series of cumulative fire intensity along the BTs ($CF_{BT}$) was calculated based on (i) an ensemble of filtered three-day HYSPLIT BTs, started every hour in the time frame between 01 January 2013 and 31 December 2018, at a starting height of 200 m, and (ii) daily georeferenced fire intensity maps, in W m$^{-2}$, from the Global Fire Assimilation System (GFAS). The GFAS fire intensity maps were obtained as NetCDF3 files with a spatial resolution of 0.1° latitude by 0.1° longitude (0.1° equals roughly 11 km). Only those segments of the individual BTs being in convective exchange with the surface/fires (i.e. BT segments with heights < 1000 m) or encountering *en-route* convection (i.e. BT segments with sun fluxes > 50 W m$^{-2}$) were included into the calculation of $CF_{BT}$. In addition, the individual BTs were terminated upon *en-route* occurrence of rain (i.e., for rainfall > 2 mm). Details on the BT data set and filtering can be found in Pöhlker et al. (2019).

We calculated the cumulative fire intensity for each trajectory as follows: each two consecutive points of the original trajectory (1-hour time step) build one linear segment of a trajectory with length ($L_i$) 1 < i < 72 h (see Fig. S1). For each trajectory segment, the collection of grid cells *(m,n)* that the trajectory passes through is computed: this is done by finding all locations along the trajectory for which either the latitude or longitude coordinate is an integer multiplied by 0.1°. To account for the residence time of air mass at each grid cell, the length ($l_{i,j}$) of the trajectory path within the cell *(m,n)* is calculated, divided by the length of the trajectory segment ($L_i$) and multiplied by the fire intensity ($F_{m,n}$) corresponding to the grid cell *(m,n)*. This results in the fire intensity weighted by the residence time of the air parcel along the segment. The cumulative fire intensity (*cumFire*) along every individual BT is calculated by summing up $F_{m,n}$ over the whole trajectory length. Note that as we used 3-day BTs, each trajectory was mapped to the raster of fire intensities of the three corresponding days (see example in Fig. S2). Finally, we summed up

the cumulative fire intensities over the 24 BTs for each day in order to obtain the $CF_{BT}$ time series with 1-day time resolution.

The method described above is summarized in the following equations. Let $F$ be a matrix containing fire intensities of size $M \ x \ N$. Let $T$ be a list of $K$ trajectories.

$$cumFire = \sum_j \sum_{i=1}^{72} F_{m,n} \cdot \frac{l_{i,j}}{L_i} \qquad (1)$$

and $l_{i,j}$ is the path of trajectory segment within cell $F_{m,n}$, calculated as follows

$$l_{i,j} = \sqrt{\left( (x_{m+1} - x_m) \cdot \cos\left(\frac{y_{n+1}+y_n}{2}\right) \right)^2 + (y_{n+1} - y_n)^2} \qquad (2)$$

Finally, for the ensemble of 24 trajectories at each day:

$$CF_{BT} = \sum_{k=0}^{24} cumFire(T_k) \qquad (3)$$

**2.7. GIS data products and analysis**

[revised manuscript text omitted]

In terms of aerosol properties, the UPL is characterized by a relative maximum in total number concentrations, $N_{CN,20} = 970 \pm 260$ cm$^{-3}$ (mean $\pm$ 1 std, Fig. 3b). Aerosol particles in the accumulation mode dominate the UPL aerosol, as $N_{acc} = 800 \pm 340$ cm$^{-3}$ accounts for most of $N_{CN,20}$ (~85 %). This corresponds to a significant drop in the ultrafine particle fraction with $f_{fine} \approx 15$ % within the UPL (Fig. 3b). The aerosols in the UPL are further characterized by a low fraction of volatile particles, $f_{vol}$, as shown in Fig. 3c. In the atmospheric column, $f_{vol}$ reaches its minimum of $16 \pm 9$ % within the UPL and generally shows a similar profile as $f_{fine}$, indicating a rather aged plume (Grieshop et al., 2009; Zhou et al., 2017). The particle number size distributions of the UPL aerosol – in comparison to the LPL, CL, MBL, and fresh BB aerosols probed during AC19 – are shown in Fig. 4 and summarized in Table 2. A modal diameter of 132 nm was observed for the UPL aerosol, whereas the fresh BB aerosol showed a clearly smaller modal diameter of 124 nm. Further note that the modal diameter in the UPL is smaller than the 220 nm observed directly off the African coast (Weinzierl et al., 2006). The CCN concentrations at S = 0.5 %, $N_{CCN,0.5}$, show a maximum within the UPL with $N_{CCN,0.5} = 560 \pm 180$ cm$^{-3}$ as well as a high CCN fraction, $f_{CCN,0.5} = 56 \pm 9$ % (Fig. 3d).

The $N_{acc}$ within the UPL is lower than at the ATTO site under strongly BB-influenced ($N_{acc,BB} \approx$ 3400 cm$^{-3}$) and average dry season conditions ($N_{acc,dry} \approx 1300$ cm$^{-3}$), yet still substantially higher than under average wet season at ATTO ($N_{acc,wet} \approx 150$ cm$^{-3}$) or pristine rain forest conditions ($N_{acc,PR} \approx 90$ cm$^{-3}$) (Pöhlker et al., 2016, 2018). Remarkably, rBC particles represent a dominant species of the UPL aerosol population in terms of number concentration with $N_{rBC} = 280 \pm 110$ cm$^{-3}$, corresponding to rBC number fraction of $f_{rBC,N} = 28 \pm 5$ % relative to $N_{CN,20}$ (Fig. 3e). The ratio $\Delta N_{rBC}/\Delta N_{CN,20} \approx 40$ % in the UPL is much higher than $\Delta N_{rBC}/\Delta N_{CN,20} \approx 5$ % in the fresh BB plume (Fig. 5a). Visually, the dark color of the layer observable in Fig. 2 corresponds with the high rBC fraction. For comparison, rBC number fractions of 0 – 15 % relative to $N_{CN,20}$ were observed in megacity pollution (Laborde et al., 2013) and $f_{rBC,N} \approx 6$ % in wildfire plumes injected into the lowermost stratosphere in the northern hemisphere (Ditas et al., 2018).

In terms of absolute mass concentrations, rBC within the UPL, with $M_{rBC} = 1.0 \pm 0.4$ μg m$^{-3}$ (ranging from 0.5 to 2 μg m$^{-3}$), approaches the highest BC levels observed at ATTO ($M_{BCe}$ up to 2.5 μg m$^{-3}$; Pöhlker et al., 2018; Saturno et al., 2018b). Figure 6 shows the fractions of rBC mass relative to the other main constituents of the submicrometer aerosol ($M_{total}$ = non-refractory + rBC) in the UPL in comparison to the CL, LPL, MBL, and fresh BB values. Organic matter – comprising co-emitted primary as well as secondarily formed organics– accounts for the dominant mass fractions in all layers, with $f_{org,M} \approx 50$ % in the UPL, CL and, LPL, and as much as 72 % in the fresh BB plume. Generally, the dominance of organic matter is in agreement with previous studies performed at different locations and seasons in the Amazon

region (e.g. Brito et al., 2014; Chen et al., 2015; Fuzzi et al., 2005; Martin et al., 2010, 2017; de Sá et al., 2019; Schneider et al., 2011; Schulz et al., 2018; Shrivastava et al., 2019; Talbot et al., 1990). For example, in the southwestern region of the Amazon, which is heavily impacted by BB, organics account for $f_{org,M} > 90\%$ in the dry season (Brito et al.2014). Note that the thermal stability of some organic species and tar balls in BB plumes can lead to an underestimation of the $f_{org,M}$ measured by the C-ToF-AMS (Adachi et al., 2018). Further, the organic matter in the UPL is significantly more oxidized than the fresh BB smoke, as shown in Fig. S6. This can be associated with the long aging times and the elevated $O_3$ mixing ratio in the UPL (Fig. 3h) (Martin et al., 2017). The rBC mass fractions account for $f_{rBC,M} = 15$ % in the UPL and $f_{rBC,M} = 12\%$ in the BB plume. A clear difference was observed for the mass fractions of the inorganic constituents sulfate ($SO_4^{2-}$), ammonium ($NH_4^+$), and nitrate ($NO_3^-$), which in sum account for $f_{inorg,M} = 35$ % in the UPL and $f_{inorg,M} = 16$ % in the BB plume. The increased $f_{inorg,M}$ in the UPL can probably be explained by aging-related condensation of the secondarily formed species $SO_4^{2-}$, $NH_4^+$, and $NO_3^-$. On the other hand, the lower $f_{org,M}$ in the UPL compared to the fresh Amazonian BB is related to the evaporation of organics due to fragmentation during the aging over the Atlantic. Note that, despite the higher $\Delta N_{rBC}/\Delta N_{CN,20}$ in the UPL compared to the fresh BB (Fig. 5a), the UPL shows higher CCN activated fraction ($\Delta N_{CCN,0.5}/\Delta N_{CN,20} = 66$ %, Fig. 5b). The high CCN efficiency is likely due to internal mixing of rBC with sulfate, nitrate, and highly oxygenated organic aerosol. These findings, in combination with the UPL's large geographic extent, suggests that it represents an aerosol and CCN reservoir of particular significance for the Amazonian cloud cycling and rainfall formation – i.e., cloud droplet formation and growth.

Regarding trace gases, Fig. 3g – j shows absolute maxima in the UPL for the mole fractions of carbon monoxide ($c_{CO}$), ozone ($c_{O3}$), and total reactive nitrogen ($c_{NOy}$) as well as a secondary maximum for nitrogen monoxide ($c_{NO}$). The elevated $c_{CO} = 150 \pm 30$ ppb along with the high $M_{rBC}$ indicates that the UPL air masses originated from BB emissions. Moreover, the ratio between these two co-emitted species can be used as tracer for the origin and age of BB plumes (Darbyshire et al., 2019; Guyon et al., 2005; Saturno et al., 2018b). The aged UPL is characterized by a higher rBC enhancement ratio, $EnR_{BC,M} = 14.7 \pm 0.6$ ng m$^{-3}$ ppb$^{-1}$, compared to fresh Amazonian BB with $EnR_{BC,M}$ of $6.3 \pm 0.2$ ng m$^{-3}$ ppb$^{-1}$ (Fig. S7). Recent aircraft measurements of African BB pollution over Ascension Island have found similar $EnR_{BC,M} = 11 - 17$ ng m$^{-3}$ ppb$^{-1}$ in the free troposphere (Wu et al., 2019). The ozone as secondary pollutant also presents a maximum within the UPL ($c_{O3} = 56 \pm 9$ ppb) and appears to be anti-correlated with NO ($c_{NO} = 0.10 \pm 0.02$ ppb). Therefore, the fact that $O_3$ and $NO_y$ ($c_{NOy} = 2.5 \pm 0.8$ ppb) are strongly enhanced in the pollution layers, reflects the high photochemical age of the plume. Overall, the trace gas mole fractions within the UPL are consistent with previous aircraft measurements. Over the Atlantic, off the city of Natal, Brazil, Andreae et al. (1994) found similar pollution layers with $c_{O3}$ and $c_{CO}$ up to 90 and 210 ppb, respectively. The mean

mole fraction of $NO_y$ in these plumes was extremely high: $4.4 \pm 3.1$ ppb, with enhancement ratios, $EnR_{NOy}$, in the range 0.018 to 0.108. The $EnR_{NOy}$ in the UPL (0.019) lies at the lower part of this range. Over Ascension Island, $c_{O3}$ can be as high as 80 ppb in the lower troposphere (Thompson et al., 1996).

Below the UPL, the atmospheric vertical profile off the Brazilian coast shows a second maximum in aerosol concentrations in the LPL ($N_{CN,20} = 1300 \pm 200$ cm$^{-3}$; $N_{acc} = 650 \pm 140$ cm$^{-3}$) at altitudes between ~2.3 to 3.0 km (Fig. 3). The properties of the UPL and LPL, however, are remarkably different. The LPL shows rather lower concentrations of rBC, ($M_{rBC} = 0.36 \pm 0.11$ µg m$^{-3}$ and $N_{rBC} = 110 \pm 20$ cm$^{-3}$), CO ($c_{CO}$ = $105 \pm 5$ ppb) and $O_3$ ($c_{O3} = 45 \pm 2$ ppb), which decreases with decreasing altitude. $NO_y$ actually reaches the highest concentrations in this layer, with values up to 3.0 ppb. We assume that the pyrogenic species found in the LPL are also advected from Africa, however, possible influences from urban emissions in Africa and/or South America, for example, should not be neglected. This possibility is supported by the relatively high sulfate content of the aerosol in this layer, which at an average value of $0.79 \pm 0.02$ µg m$^{-3}$ accounts for 23% of total aerosol mass concentration (Fig. 6). Sulfur-rich anthropogenic emissions from fossil-fuel combustion may have become mixed with BB emissions by cloud-venting over the Gulf of Guinea region (Dajuma et al., 2019).

One interesting aspect of the LPL is that the ultrafine fraction accounts for about half of the aerosol number concentration ($d_0 = 105$ nm, see PNSD in Fig. 4). Likewise, in the LPL the $f_{vol}$ is higher than in the other atmospheric levels. One possible explanation for this is that new particle formation occurs in the detrainment regions around the shallow cumulus, which brings air masses from the marine boundary layer (MBL), containing dimethyl sulfide and $SO_2$, into the LPL. This phenomenon has previously been reported by several authors (Hegg et al., 1990; Kerminen et al., 2018; Perry and Hobbs, 1994). Direct convective transport of ultrafine particles from the MBL into the LPL is unlikely to be an important source of such particles, as their concentration in the MBL is only about 200 cm$^{-3}$, well below their concentration in the LPL of about 700 cm$^{-3}$. In the MBL (with its top at ~600 m asl), the total and accumulation mode particle concentrations are somewhat lower than in the layers aloft ($N_{CN,20} = 420 \pm 160$ cm$^{-3}$ and $N_{acc} = 230 \pm 50$ cm$^{-3}$) and present larger diameters ($d_0 = 143$ nm). The MBL appears to be only weakly influenced by the African BB, with $M_{rBC} = 0.18 \pm 0.07$ µg m$^{-3}$ and $N_{rBC}$ accounting for only 10 % of the $N_{CN,20}$. Additionally, the aerosol population in the MBL appears less efficient as CCN, with only 20 % of particles being activated at $S = 0.5\%$ (Fig. 3d).

In between the UPL and LPL, the ~200 m thick CL was found centered at ~3.2 km altitude (~200 m thick) with relatively dry air as represented by a sharp decrease in $q$. Such clean layers have been previously observed in the dry season over the African continent and adjacent oceans, specifically in the southeastern Atlantic Ocean, with a few hundred (up to 1 km) meters thickness (Hobbs, 2003). Within the CL, the combustion tracer concentrations $M_{rBC}$, $N_{rBC}$, and $c_{CO}$ sharply decrease to $0.09 \pm 0.04$ µg m$^{-3}$, $30 \pm 12$

cm$^{-3}$ and 83 ± 4 ppb, respectively. We further found $N_{CN,20}$ = 500 ± 60 cm$^{-3}$, which is comparable to $N_{CN}$ = 500 cm$^{-3}$ in another CL as reported by Hobbs (2003). Within the CL, the aerosol size distribution is substantially shifted towards the Aitken mode ($d_0$ = 90 nm, Fig. 4). The $c_{O3}$ shows a slight decrease to 48 ± 2 
[revised manuscript text omitted]
). More examples of similar layers over the Atlantic Ocean in September 2016 are shown and discussed in Barkley et al. (2019). These passages show exemplary snapshots of the elevated smoke layers at different longitudinal locations: on 16 Sep 2014 a layer was probed relatively close to the southern African coast, whereas on 30 Sep 2014 a layer was observed halfway between Ascension Island and the Amazon River delta. For the overpass on 30 Sep 2014, the layer's N-S extension was about 1200 km and its altitude between 3 and 4 km, which agrees well with the altitude of the UPL observation during flight AC19. For the passage on 16 Sep 2014, the layer's N-S extension was about 4º N to 20º S (~2800 km) and its altitude between 2 and 5 km. In this context, a dedicat-
ed study of Adebiyi and Zuidema (2016) has showed that 45 % of the forward trajectories of satellite-
detected smoke plumes in southern Africa exit the continent westwards between 5º S and 15º S and are
transported westward by the Southern African Easterly Jet, overlying a semi-permanent marine stratocu-
mulus deck. Moreover, Fig. 9 suggest that the layer's latitudinal extent decreases as it approaches the
South American continent.

In order to constrain the seasonal and vertical aspects of the transatlantic transport, we analyzed the
satellite-retrieved aerosol profiles over the South Atlantic Ocean during the dry season of multiple years
(2012 to 2018). Figure 10a-c shows the extinction coefficients of all CALIPSO overpasses within the re-
gion of interest (ROI, as defined in Fig. 7a) averaged over the months of August, September and October.
High aerosol loadings (up to 5 km altitude) in the longitude band from 10º E to 20º E correspond to BB
emissions over the African continent. Likewise, comparably high extinction coefficients (up to 3 km alti-
tude) are observed due to BB fires in South America (60º W to 40º W). Over the South Atlantic from
40º W to 10º E, the maximum extinction coefficient is observed at two different levels of the atmosphere,
separated by a relatively clean layer in between. The lower layer (altitude < 1 km) with pronounced ex-
tinction coefficient represents the MBL, which is presumably dominated by the (coarse-mode) marine
aerosols and is clearly visible throughout the three months in Fig. 10. On the other hand, the higher layer
(altitudes between 1 and 5 km) represents the African BB aerosol being transported westwards over the
Atlantic all the way to South America. The transport pattern stands out in the months of August and Sep-
tember, but is weakened in October, when the remaining BB plumes appear to be mostly/completely re-
moved from the atmosphere half-way before reaching South America. The injection height of BB aerosol
in Africa is relatively high due to the AEJ-S, which induces an upward motion directly below the jet, en-
hancing updrafts over land that lift up BB aerosols to altitudes where they can be efficiently transported
over the South Atlantic (Adebiyi and Zuidema, 2016). The vertical location of pollution plumes in the
atmosphere is an important parameter, as it can considerably influence its atmospheric lifetime. Aerosol
lifted up to higher altitudes tends to be advected over larger distances due to less efficient removal mecha-
nisms (i.e., wet deposition). When leaving the African coast, the smoke layer is present at altitudes be-
tween 1.5 and 5 km, but becomes more restricted to higher altitudes (3 – 5 km) as it moves towards South
America. Fig. 10 suggests a pronounced thinning of the layer during its movement westwards due to dilu-
tion.

[revised manuscript text omitted]

**Competing interests**

The authors declare that they have no conflict of interest.

**Acknowledgements**

We acknowledge the Conselho Nacional de Desenvolvimento Científico e Tecnológico (CNPq, Brazil), process 200723/2015-4, the Max Planck Graduate Center with the Johannes Gutenberg University Mainz (MPGC) and the Max Planck Society, for the financial support. We thank the entire ACRIDICON-CHUVA team for collecting the data and for the fruitful scientific cooperation. Special thanks goes to the HALO pilots, Steffen Gemsa, Michael Grossrubatscher, and Stefan Grillenbeck. We thank Volker Dreiling, Sensor and Data Team of DLR Flight Experiments and the HALO team of the DLR for their cooperation. We acknowledge the generous support of the ACRIDICON-CHUVA campaign by the Max Planck Society, the German Aerospace Center (DLR), FAPESP (São Paulo Research Foundation), and the German Science Foundation (Deutsche Forschungsgemeinschaft, DFG) within the DFG Priority Program (SPP 1294, SCHN1138/1-2) "Atmospheric and Earth System Research with the Research Aircraft HALO (High Altitude and Long Range Research Aircraft)". This study was also supported by EU Project HAIC under FP7-AAT-2012-3.5.1-1 and by the German Science Foundation within DFG SPP 1294 HALO by contract no VO1504/4-1 and contract no JU 3059/1-1. For the operation of the ATTO site, we acknowledge the support by the Max Planck Society, the German Federal Ministry of Education and Research (BMBF contracts 01LB1001A, 01LK1602A, 01LK1602B and 01LG1205E) and the Brazilian Ministério da Ciência, Tecnologia e Inovação (MCTI/FINEP contract 01.11.01248.00) as well as the Amazon State University (UEA), FAPEAM, LBA/INPA and SDS/CEUC/RDS-Uatumã. This paper contains results of research conducted under the Technical/Scientific Cooperation Agreement between the National Institute for Amazonian Research, the State University of Amazonas, and the Max-Planck-Gesellschaft e.V.; the opinions expressed are the entire responsibility of the authors and not of the participating institutions. Special thanks for all the people involved in ATTO project, in particular Reiner Ditz, Jürgen Kesselmeier, Andrew Crozier, Thomas Disper, Alcides Camargo Ribeiro, Hermes Braga Xavier, Nagib Alberto de Castro Souza, Adir Vasconcelos Brandão, Amauri Rodriguês Perreira, Antonio Huxley Melo Nascimento, Thiago de Lima Xavier, Josué Ferreira de Souza, Roberta Pereira de Souza, Bruno Takeshi, Wallace Rabelo Costa, Uwe Schultz and Steffen Schmidt. Remote sensing analyses and visualizations used in this study were produced with the Giovanni online data system, developed and maintained by the NASA GES DISC. We acknowledge the National Oceanic and Atmospheric Administration (NOAA) Air Resources Laboratory (ARL) for the HYSPLIT transport and dispersion model. We thank Daniel Moran-Zuloaga, Maria Praß, Leslie Kremper, Tobias Könemann, Jan-David Förster, Björn Nillius, Stefan Wolff, Anywhere Tsokankunku, Oliver Lauer for their support and inspiring discussions. BW, MD and AW would like to acknowledge funding from the European Research Council under the European Community's Horizon 2020 research and innovation framework program/ERC Grant Agreement 640458 (A-LIFE).

[revised manuscript text omitted]

**Table 1.** Characteristic aerosol and trace gas concentrations in the upper pollution layer (UPL), clean layer (CL), lower pollution layer (LPL) and marine boundary layer (MBL) observed during the AC19 flight section off the Brazilian coast (16:50 to 19:07 UTC; Fig. 1). For comparison, corresponding data from a fresh biomass burning plume (BB, also observed during AC19, see Fig. 1) has been added. Data is summarized as arithmetic mean ± standard deviation (std) as well as 1$^{st}$ and 99$^{th}$ percentiles (P1 and P99).

| | UPL | | | | CL | | | | LPL | | | | MBL | | | | BB | | | |
|---|---|---|---|---|---|---|---|---|---|---|---|---|---|---|---|---|---|---|---|---|
| | mean | std | P1 | P99 | mean | std | P1 | P99 | mean | std | P1 | P99 | mean | std | P1 | P99 | mean | std | P1 | P99 |
| $N_{CN,20}$ (cm$^{-3}$) | 970 | 260 | 400 | 1500 | 500 | 60 | 450 | 760 | 1300 | 200 | 750 | 1750 | 420 | 140 | 220 | 800 | 4200 | 2100 | 1500 | 10500 |
| $N_{acc}$ (cm$^{-3}$) | 850 | 330 | 260 | 1550 | 180 | 60 | 90 | 360 | 650 | 140 | 320 | 940 | 230 | 50 | 140 | 330 | 2700 | 1400 | 660 | 6600 |
| $N_{CCN,0.5}$ (cm$^{-3}$) | 560 | 180 | 200 | 920 | 230 | 40 | 170 | 350 | 510 | 90 | 320 | 700 | 95 | 30 | 30 | 200 | 2000 | 1100 | 500 | 5100 |
| $N_{rBC}$ (cm$^{-3}$) | 280 | 110 | 90 | 530 | 30 | 12 | 9 | 60 | 110 | 20 | 60 | 190 | 50 | 16 | 24 | 84 | 280 | 110 | 120 | 630 |
| $f_{fine}$ (%) | 15 | 14 | 0 | 45 | 65 | 7 | 44 | 79 | 48 | 11 | 27 | 69 | 43 | 13 | 18 | 68 | 37 | 8 | 19 | 61 |
| $f_{vol}$ (%) | 16 | 9 | 3 | 40 | 43 | 6 | 25 | 55 | 27 | 7 | 13 | 44 | 39 | 9 | 22 | 62 | 17 | 7 | 2 | 38 |
| $f_{rBC,N}$ (%) | 28 | 5 | 17 | 40 | 6 | 2 | 2 | 11 | 9 | 2 | 5 | 14 | 12 | 3 | 5 | 19 | 7 | 2 | 4 | 11 |
| $f_{CCN,0.5}$ (%) | 60 | 6 | 44 | 71 | 46 | 6 | 34 | 62 | 41 | 8 | 26 | 59 | 23 | 6 | 11 | 38 | 45 | 5 | 31 | 54 |
| $c_{CO}$ (ppb) | 150 | 30 | 100 | 210 | 83 | 4 | 78 | 101 | 105 | 5 | 96 | 115 | 92 | 1 | 90 | 94 | 162 | 40 | 107 | 277 |
| $c_{O3}$ (ppb) | 56 | 9 | 36 | 71 | 48 | 2 | 46 | 53 | 45 | 2 | 41 | 51 | 21 | 1 | 19 | 23 | 34 | 2 | 31 | 38 |
| $c_{NOy}$ (ppb) | 2.5 | 0.8 | 1.0 | 4.5 | 0.9 | 0.4 | 0.6 | 2.0 | 2.1 | 0.8 | 0.8 | 3.1 | 1.7 | 0.1 | 1.4 | 1.8 | 2.1 | 0.3 | 1.7 | 2.9 |
| $c_{NO}$ (ppb) | 0.10 | 0.02 | 0.06 | 0.13 | 0.07 | 0.01 | 0.04 | 0.09 | 0.10 | 0.02 | 0.06 | 0.15 | 0.12 | 0.02 | 0.09 | 0.15 | 0.18 | 0.06 | 0.11 | 0.32 |
| $M_{rBC}$ (µg m$^{-3}$) | 1.0 | 0.4 | 0.3 | 2.0 | 0.09 | 0.05 | 0.02 | 0.25 | 0.36 | 0.11 | 0.17 | 0.73 | 0.17 | 0.07 | 0.07 | 0.37 | 0.70 | 0.24 | 0.35 | 1.36 |
| $M_{Org}$ (µg m$^{-3}$) | 2.5 | 1.2 | 1.2 | 4.4 | 0.91 | - | - | - | 1.72 | 0.15 | 1.57 | 1.86 | -0.02 | 0.04 | -0.08 | 0.04 | 4.6 | 1.6 | 2.2 | 7.4 |
| $M_{SO4}$ (µg m$^{-3}$) | 0.86 | 0.41 | 0.50 | 1.60 | 0.42 | - | - | - | 0.79 | 0.02 | 0.76 | 0.81 | 0.22 | 0.08 | 0.14 | 0.33 | 0.57 | 0.18 | 0.38 | 0.94 |
| $M_{NO3}$ (µg m$^{-3}$) | 0.33 | 0.26 | 0.08 | 0.70 | 0.04 | - | - | - | 0.05 | 0.02 | 0.03 | 0.07 | 0.013 | 0.012 | -0.002 | 0.025 | 0.15 | 0.06 | 0.05 | 0.25 |
| $M_{NH4}$ (µg m$^{-3}$) | 0.48 | 0.23 | 0.25 | 0.81 | 0.25 | - | - | - | 0.45 | 0.16 | 0.31 | 0.62 | 0.18 | 0.11 | 0.06 | 0.37 | 0.33 | 0.11 | 0.20 | 0.56 |

**Table 2.** Fit parameters of UHSAS-derived aerosol size distributions in Fig. 4, representing different conditions (i.e., layers, plumes) during AC19. A log-normal function (Heintzenberg, 1994) was used to fit a mono-modal size distribution to the mean data points:

$$\frac{dN}{d \ln d_p} = \frac{A}{\sqrt{2\pi} \ln \sigma_g} exp\left(-\frac{(\ln d_p - \ln d_0)^2}{2 \ln(\sigma_g)^2}\right)$$

|  | UPL | CL | LPL | MBL | BB |
|---|---|---|---|---|---|
| $A$ | 2920 | 970 | 2890 | 680 | 13930 |
| $d_0$ *(nm)* | 132 | 90 | 105 | 143 | 124 |
| $\sigma_g$ | 1.55 | 1.58 | 1.65 | 1.40 | 1.50 |
| $r^2$ | 1.00 | 0.99 | 1.00 | 1.00 | 1.00 |

1250

**Table 3.** Characteristic aerosol and trace gas concentrations during the African vs. South American dominated periods of the BB season at the ATTO site: arithmetic mean ± std, median and inter-quartile range of daily averages from 2013-2018.

| | African dominated BB** | | | | | South American dominated BB*** | | | | |
|---|---|---|---|---|---|---|---|---|---|---|
| | mean | std | median | 25th perc | 75th perc | mean | std | median | 25th perc | 75th perc |
| $N_{CN,20}$ (cm$^{-3}$) | 1350 | 550 | 1300 | 900 | 1700 | 2000 | 1000 | 1800 | 1400 | 2300 |
| $N_{CCN,0.5}$ (cm$^{-3}$) | 1100 | 500 | 1090 | 750 | 1400 | 1800 | 900 | 1600 | 1200 | 2000 |
| $M_{BCe}$ (µg m$^{-3}$) | 0.36 | 0.12 | 0.33 | 0.26 | 0.42 | 0.41 | 0.17 | 0.36 | 0.29 | 0.48 |
| $c_{CO}$ (ppb) | 140 | 30 | 131 | 120 | 150 | 190 | 70 | 170 | 150 | 200 |
| $f_{CCN,0.5}$ (%) | 83 | 6 | 84 | 80 | 87 | 87 | 04 | 88 | 84 | 90 |
| $EnR_{BC}$* (ng m$^{-3}$ ppb$^{-1}$) | 11 | 6 | 10 | 7 | 14 | 6 | 4 | 5 | 3 | 7 |
| $SSA$ (637 nm) | 0.85 | 0.02 | 0.85 | 0.84 | 0.86 | 0.90 | 0.03 | 0.90 | 0.89 | 0.93 |

* daily $EnR_{BC}$ (from 2013 – 2018) was calculated by applying a bivariate fit to 30-min averages of $M_{BCe}$ and $c_{CO}$ measurements (see Sect. 2.3).

** for calculating the African BB dominated state, periods with South American fires influences ($CF_{BT}$ > 50th percentile of $CF_{BT,dry}$) and with clean atmospheric conditions ($M_{BCe}$ < 50th percentile of $M_{BCe,dry}$) were excluded.

***for calculating the South American BB dominated state, we selected periods with South American fires influences ($CF_{BT}$ > 50th percentile of $CF_{BT,dry}$) and with polluted atmospheric conditions ($M_{BCe}$ > 50th percentile of $M_{BCe,dry}$).

[Figure]

**Figure 1.** ACRIDICON-CHUVA flight AC19 on 30 September 2014. The squares represent the locations at which the aircraft ascended or descended through the upper pollution layer (UPL) (blue: offshore profiles, orange: inshore profiles). The yellow and blue segments of the flight track correspond to the in- and offshore sections that were averaged to obtain the profiles in Fig. 3. Red markers indicate fire spots on 30 September 2014 as obtained from INPE (http://www.inpe.br/queimadas/bdqueimadas/, last access on 17 April 2019), and the dark green square represents the location where a fresh BB plume was probed at ~1 km altitude.

[Figure]

**Figure 2.** View from the HALO cockpit during flight AC19 on 30 September 2014, showing **(a)** the layering of the troposphere with clearly visible pollution layers as well as a clean layer in between at an offshore location (17:09 UTC) and **(b)** the brownish pollution layer arriving at the Brazilian coastline (16:55 UTC).

[Figure]

**Figure 3.** Vertical profiles of selected meteorological, aerosol, and trace gas parameters measured off the Brazilian coast during flight AC19: (**a**) potential temperature, $\theta$, and water vapor mass mixing ratio, $q$; (**b**) total aerosol particle number concentration, $N_{CN,20}$, and ultrafine particle number fraction, $f_{fine}$; (**c**) accumulation mode particle number concentration, $N_{acc}$, and volatile particle number fraction, $f_{vol}$; (**d**) CCN number concentration at $S = 0.5$ %, $N_{CCN,0.5}$, and activated fraction at $S = 0.5$ %, $f_{CCN,0.5}$; (**e**) rBC number concentration, $N_{rBC}$, and rBC number fraction, $f_{rBC,N}$; and (**f**) rBC mass concentration, $M_{rBC}$; (**g**) carbon monoxide, $c_{CO}$; (**h**) ozone, $c_{O3}$; (**i**) total reactive nitrogen, $c_{NOy}$; and (**j**) nitrogen monoxide, $c_{NO}$, mole fractions measured off the Brazilian coast during flight AC19. The black lines and shadings represent the median and inter-quartile range calculated for 150 m altitude bins during the flight section off the Brazilian coast (16:50 to 19:07 UTC, blue line in Fig. 1). The brown shaded area represents the approximate vertical location of the upper pollution layer (UPL). The altitudes of the lower pollution layer (LPL), the clean layer (CL), and the marine boundary layer (MBL) are indicated on the right side of the plot. The precise time windows when the UPL, CL, LPL and MBL were probed are shown in Table S1 of the SI.

[Figure]

**Figure 4.** Particle number size distributions (PNSD) measured by the UHSAS for UPL, CL, LPL, and MBL, as defined in Fig. 3, and the fresh BB plume probed during AC19 (see Fig. 1). The data points (black dots) are fitted by lognormal functions between 90 and 500 nm (Heintzenberg, 1994).

[Figure]

**Figure 5.** Correlation between **(a)** rBC particle number concentrations ($N_{rBC}$) and total aerosol ($N_{CN,20}$); and between **(b)** CCN at $S = 0.5$ % ($N_{CCN,0.5}$) and total aerosol ($N_{CN,20}$) in the UPL (blue) and in the fresh biomass burning plume (green). The dashed lines are bivariate linear regressions applied to the data sets.

[Figure]

**Figure 6.** Cumulative mass concentrations of non-refractory submicrometer species (i.e., organic (Org), sulfate ($SO_4^{2-}$), nitrate ($NO_3^-$), ammonium ($NH_4^+$)) and rBC **(top)**; and mass fractions of the respective species to the total mass ($M_{total} = M_{Org} + M_{SO4} + M_{NO3} + M_{NH4} + M_{rBC}$) in the UPL, CL, LPL, and MBL, as defined in Fig. 3, and the fresh BB plumes probed during AC19 (see Fig. 1) **(bottom)**. Note that no C-ToF-AMS data were available from 17:27 to 19:05 UTC during the off-shore section of the flight AC19 and, therefore, reduced number of measurements points are included in the averages. The concentration of organics was below the detection limit in the MBL.

[Figure]

**Figure 7.** Composite maps combining backward trajectories (BTs) and satellite data products characterizing atmospheric conditions **(a₁)** during flight AC19 on 30 September 2014 in comparison to **(b)** the averages of September observations during multiple years. Panel **(a₁)** shows HYSPLIT 10-days BT starting at different altitudes (500, 2500, 3500, 5000 m a.g.l.) at 18:00 UTC on 30 September 2014 (similar time and location as the UPL observations during flight AC19). Note that the altitudes where the BTs were initiated include the heights of the sampled UPL and LPL. The Fire Radiative Power (FRP) density (mW m$^{-2}$), retrieved by the Global Fire Assimilation System (GFAS v1.0) averaged from 15 to 20 of September 2014 is also shown as a fire map with 0.1$^{o}$ x 0.1$^{o}$ grid resolution. The orbits of two CALIPSO passages on 30

1275    and 16 September 2014 as shown in Fig. 9 as well as the geographic locations of the ATTO site and Ascension Island are also illustrated. Panel **(a₂)** shows multiple clearly visible fire plumes in the African

sources region. Panel **(b)** shows multi-year averages of all Septembers for: (i) HYSPLIT BT ensembles

starting at ATTO (1000 and 4000 m a.g.l.) from 2005 to 2018. Contour lines represent the fraction of occurrence of overpassing trajectories in a specific region as described in Pöhlker et al. (2019). (ii) AIRS-

1280    derived CO data products (400 to 600 hPa atmospheric levels) from 2005 to 2018 and (iii) TRMM precipitation from 2005 to 2018. For general illustration, animations[2] of the Goddard Earth Observing Model

(Version 5, GEOS-5) show that aerosol particles are transported efficiently from Africa to South America

and to a lesser extent from South America to Africa (Colarco et al., 2010; Yasunari et al., 2011).
* * *
[2] https://climate.nasa.gov/climate_resources/146/video-simulated-clouds-and-aerosols/ (last access 04 Jul 2019)
https://gmao.gsfc.nasa.gov/research/aerosol/modeling/nr1_movie/ (last access 04 Jul 2019)

[Figure]

**Figure 8.** Hovmöller plot of the daily AIRS-derived carbon monoxide (400 to 600 hPa) distributed over the South Atlantic region (60 ºW to 20 ºE) from August to October 2014, averaged over the latitudinal band of 10 ºS to 5 ºN, corresponding to the region of interest (ROI) highlighted in Fig. 7a$_1$. Several events of transatlantic transport of aerosol from Africa towards South America can be identified. The black dashed line highlights a particularly strong plume originating around 10 September 2014 and arriving in the observational area of AC19 on 30 September.

[Figure]

**Figure 9.** CALIPSO-derived lidar profiles for 16 and 30 September 2014, where African BB plumes were identified over the South Atlantic Ocean. The first profile near the South American coast shows the aerosol layer at similar altitudes as observed during flight AC19. Satellite orbits for both profiles are shown in Fig. 7a₁.

.

[Figure]

**Figure 10.** Curtain plot showing the columnar aerosol extinction coefficient at 532 nm, based on multi-year CALIOP data from 2012 to 2018 (only night time data). Panels represent monthly averages for the months of **(a)** August, **(b)** September, and **(c)** October within the latitude band from 10° S to 5° N, corresponding to the ROI indicated in Fig. 7a$_1$. The grey shaded area represents the mean surface elevation and depicts boundaries of the African and South American continents. Panel **(d)** shows the daily mean of the direct radiative forcing at the top of the atmosphere (DRF-TOA) exerted by the pollution layer over the South Atlantic Ocean, calculated using the LibRadTran radiative transfer model.

[Figure]

**Figure 11.** Vertical profiles of selected meteorological, aerosol and trace gas parameters measured inshore and offshore of the Brazilian coast during AC19: (**a**) potential temperature, $\theta$; (**b**) water vapor mass mixing ratio, $w$; (**c**) CCN ($S = 0.5$ %) number concentration, $N_{CCN,0.5}$; (**d**) rBC mass concentration, $M_{rBC}$; and (**e**) carbon monoxide mole fraction, $c_{CO}$. The figure shows the medians calculated for 150-m altitude bins over the flight sections inshore (yellow) and offshore (blue) the Brazilian coast (as indicated in Fig. 1). The grey shaded area represents the approximate vertical location of the upper pollution layer (UPL) and the grey dashed line, the lower limit of the clean layer (CL) observed exclusively during the offshore profiles.

[Figure]

**Figure 12.** Seasonality of aerosol and trace gas properties based on long-term observations (2013-2018) at Ascension Island **(a)** and at the ATTO site **(b-g)**. The plot shows **(a)** fine-mode aerosol optical depth (AOD-fine) at 500 nm retrieved by the AERONET sunphotometer at Ascension Island; **(b)** cumulative fire intensity along the backward trajectories (CF$_{BT}$) of air masses arriving at the ATTO site (for details see Sect. 2.6); **(c)** black carbon mass concentration $M_{BCe}$; **(e)** CCN concentrations and activated fraction at $S = 0.5$ %; **(e)** CO mole fraction; **(f)** single scattering albedo, SSA; and **(g)** BC enhancement ratio, $EnR_{BC}$. The boxplots represent weekly statistics with the mean (red markers), the median (segment), and the 5th and 75th percentiles (lower and upper box edges) of the long-term daily measurements. The green and red shaded areas represent the wet and dry season, with transition periods in between, as defined in Pöhlker et al. (2016). On the top of

the figure, markers indicate periods within the BB-influenced part of the year that ATTO is dominated either by the LRT of African BB or by the South American BB.

*Supplement to*

**Influx of African biomass burning aerosol during the Amazonian dry season through layered transatlantic transport of black carbon-rich smoke**

Bruna A. Holanda et al.

\* *Correspondence to:* M. Pöhlker (m.pohlker@mpic.de) and C. Pöhlker (c.pohlker@mpic.de)

**Table S1.** List of time sections in which the UPL, CL, LPL MBL and BB were sampled during flight AC19.

| | Time window | |
|---|---|---|
| *Offshore profile* | 16:50:00 | 19:07:00 |
| *UPL* | 16:21:50 | 16:22:40 |
| | 16:54:22 | 16:55:02 |
| | 17:08:22 | 17:09:00 |
| | 18:02:46 | 18:03:09 |
| | 18:03:47 | 18:04:18 |
| | 18:10:35 | 18:12:15 |
| | 18:34:32 | 18:35:47 |
| | 18:42:51 | 18:43:39 |
| | 18:47:50 | 18:48:04 |
| | 19:58:00 | 19:58:39 |
| *CL* | 17:09:08 | 17:09:30 |
| | 18:02:04 | 18:02:39 |
| | 18:36:08 | 18:38:42 |
| | 18:40:16 | 18:42:44 |
| | 18:43:43 | 18:44:24 |
| | 18:47:21 | 18:47:43 |
| *LPL* | 16:52:03 | 16:53:18 |
| | 17:10:00 | 17:11:37 |
| | 17:56:43 | 17:57:16 |
| | 17:57:33 | 17:58:15 |
| | 17:58:31 | 18:00:08 |
| | 18:54:33 | 18:55:51 |
| *MBL* | 17:23:14 | 17:27:28 |
| *Fresh BB* | 19:24:03 | 19:24:37 |
| | 19:24:49 | 19:25:08 |
| | 19:25:44 | 19:27:02 |
| | 19:27:11 | 19:28:16 |
| | 19:30:09 | 19:30:46 |
| | 19:32:24 | 19:34:03 |
| | 19:18:24 | 19:18:45 |

[Figure]

7    **Figure S1.** Sketch illustrating the initiation of a three-day BT starting at the ATTO site (red

8    circle) mapped to the raster of fire intensities.

[Figure]

**Figure S2.** Ensemble of three-day HYSPLIT BTs, starting at every hour at the ATTO site (1000

m a.s.l.) on 30 September 2014 and corresponding daily fire intensity maps (W m$^{-2}$) from the

Global Fire Assimilation System (GFAS).

[Figure]

14 **Figure S3.** View from the HALO cockpit, showing the active fire plumes (intersected at ~1 km

15 above ground) during flight AC19 at 19:20 UTC on 30 September 2014. See also fire map in Fig.

16 1.

[Figure]

**Figure S4.** Radiosonde sounding at Belem Airport (see Fig 1.) on 30 September 2014 at 12:00
UTC, provided Wyoming University (http://weather.uwyo.edu/upperair/sounding.html, last
access on 06 August 2019). The sounding shows similar tropospheric stratification as presented
in Fig. 3. The first layer (top around 1000 m) is associated with the boundary layer, the second
(top around 3200 m) is related to the shallow clouds top and the third one (around 5000 m) is the
large scale inversion.

[Figure]

25  **Figure S5.** Particle number size distributions (median and interquartile range) derived from the

26  UHSAS and SP2 (rBC + SC) for the **(a)** CL, **(b)** LPL, **(c)** MPL, **(d)** UPL and **(e)** fresh BB plume

27  probed during flight AC19. Panel **(f)** shows the curve fits of the UHSAS data points.

[Figure]

29     **Figure S6.** Scatterplot of the ratios $f_{43}$ (m/z 43 to total organic signal) against $f_{44}$ (m/z 44

30     to total organic signal) expressing the photochemical aging of the organic aerosol

31     measured by the C-ToF-AMS. The blue and green markers correspond to measurements

32     within the UPL and fresh BB, respectively. The signal at m/z 44 relates mostly to $CO_2^+$

33     ions and the m/z 43 signal to $C_2H_3O^+$ ions. The triangular region (dashed lines) in the $f_{44}$

34     vs. $f_{43}$ space defines the boundaries within which most of the organic aerosol was found in

35     previous studies and can be used as a guide to characterize oxidized organic components:

36     data in the upper left represent more oxidized organics vs. the less oxidized organics in the

37     lower right (Ng et al., 2010; Schulz et al., 2018).

[Figure]

40 **Figure S7.** Correlation between $\Delta c_{CO}$ and $\Delta M_{rBC}$ within the upper pollution layer (UPL) and fresh

biomass burning (BB) plume probed during AC19. A bivariate regression fit was applied to the

data set in order to obtain the BC enhancement ration, $EnR_{BC}$.

[Figure]

**Figure S8.** Monthly distribution of AIRS-derived carbon monoxide (400 to 600 hPa) over the Southern Hemisphere. The map shows averages over multiple years (2005-2018).

[Figure]

**Figure S9.** Hovmöller plot of the daily MODIS AOD (550 nm) distributed over the South Atlantic region (60 °W to 20 °E) from August to October 2014, averaged over the latitudinal band of 10° S to 5° N, corresponding to the region of interest (ROI) highlighted in Fig. 7a. Several events of transatlantic transport of aerosol from Africa towards South America can be easily identified, with the strongest plume starting approximately on 15 September 2014. A dashed line for this particular event is also shown in the picture, which arrives at a time close to our observations on AC19 on 30 September 2014. Westwards of 35 °W, the AOD levels are

54  increasingly influenced by the South American continent, which masks the AOD signals of the

55  transported African pollution as it approaches the South American continent.

[Figure]

57 **Figure S10.** Sensitivity tests showing the DRF-TOA changes due to different assumptions in the

58 aerosol and surface properties, comparing with the original DRF-TOA estimation. The

59 simulations show that if the aerosol layer is mostly scattering (SSA = 0.99), a general cooling

60 (back-scattering by the layer) is observed. On the other hand, if the absorbing ocean is replaced

61 by a higher surface albedo (0.6), a warming effect by the layer is observed due to the downward

62 radiation that is scattered back and forth between the aerosol layer and the surface.